# FROM FEW TO MANY: SELF-IMPROVING MANY-SHOT REASONERS THROUGH ITERATIVE OPTIMIZATION AND GENERATION

**Xingchen Wan**[1], **Han Zhou**[1,3*], **Ruoxi Sun**[2*], **Sercan Ö. Arık**[1]
[1]Google Cloud AI Research [2]Google DeepMind [3]University of Cambridge
[*]Work done at Google Cloud AI Research
{xingchenw, hzhouml, ruoxis, soarik}@google.com

## ABSTRACT

Recent advances in long-context large language models (LLMs) have led to the emerging paradigm of many-shot in-context learning (ICL), where it is observed that scaling many more demonstrating examples beyond the conventional few-shot setup in the context can lead to performance benefits. However, despite its promise, it is unclear what aspects dominate the benefits and whether simply scaling to more examples is the most effective way of improving many-shot ICL. In this work, we first provide an analysis on the factors driving many-shot ICL, and we find that 1) many-shot performance can still be attributed to often a few disproportionately influential examples and 2) identifying such influential examples ("optimize") and using them as demonstrations to regenerate new examples ("generate") can lead to further improvements. Inspired by the findings, we propose BRIDGE, an algorithm that alternates between the *optimize* step with Bayesian optimization to discover the influential sets of examples and the *generate* step to reuse this set to expand the reasoning paths of the examples back to the many-shot regime automatically. On Gemini, Claude, and Mistral LLMs of different sizes, we show BRIDGE led to significant improvements across a diverse set of tasks including symbolic reasoning, numerical reasoning and code generation.

## 1 INTRODUCTION

Recent advances in large language models (LLMs) have led to the emergence of in-context learning (ICL) as a promising new learning paradigm (Brown et al., 2020). ICL allows LLMs to learn tasks by simply being presented with a few examples within their context window. A key bottleneck for ICL has been the supported context length of LLMs, but with advancements in novel model architectures, computational infrastructures and efficient serving methods, state-of-the-art models such as Gemini (Reid et al., 2024; Anthropic, 2024) feature context windows of millions of tokens are overcoming this limitation. Such long-context LLMs open unprecedented avenues for the scaling of ICL – whereas previous LLMs were limited to processing only up to dozens of examples, current LLMs can now accommodate significantly more examples. More importantly, beyond merely *supporting* a longer context, it has also been shown that scaling more examples led to substantial performance improvements across tasks, creating a new promising paradigm known as *many-shot learning* (Agarwal et al., 2024; Bertsch et al., 2024).

Despite these advances, as a nascent paradigm, many-shot ICL still faces several challenges. Long context windows, while powerful, are computationally expensive and introduce significant latency and cost to serving, making it impractical or uneconomical to fully exploit the maximum context length and some kind of trade-off decisions have to be made under virtually any realistic settings. To leverage the expanded context while controlling the cost and latency under an acceptable limit, existing works typically investigate the experimental setting where as many examples as costs permit are simply randomly sub-sampled from the pool of all available examples and dumped into the context window. As observed both in prior works (Agarwal et al., 2024) and our investigations (Fig. 1), using the same *number* of examples but with different combinations of examples as demonstrations can lead to dramatically different performance for the *same* task. Across *different* tasks, it has also

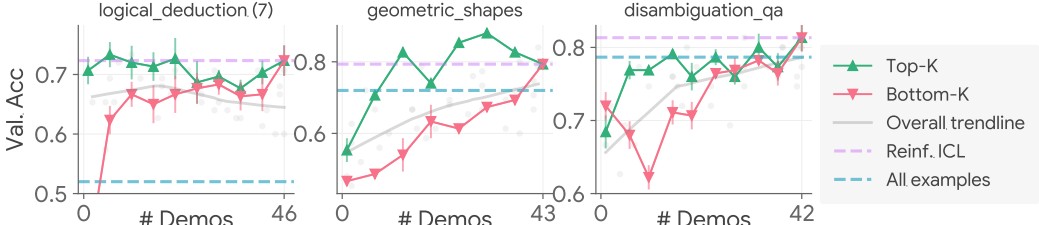

Figure 1: *It does not always take "many shots" to achieve many-shot performance – with judicious selection, it is possible to match or exceed many-shot performance achieved with using all available examples) with much fewer examples:* Accuracy on held-out splits against the number of examples on 3 BBH tasks of 1) overall trendline (fitted with locally weighted smoothing (LOWESS)), 2) using top-K most positive examples, or 3) using bottom-K least positive examples based on the ranking of the importance score described in Sec 2. Dotted lines refer to two many-shot baselines: reinforced ICL: using input, model-generated reasoning and output of all *correctly-predicted* inputs; All example: using all available input-output pairs from the train set. Lines and error bars show mean ± standard deviation across 3 runs with the ordering of the examples shuffled each trial.

been noted that the model behaves very differently when the number of examples is scaled up, with some showing a near-monotonic increase in performance as more examples are added, while others experience performance plateaus (e.g., gray line in the leftmost subfigure of Fig. 1) or even degradation (e.g., red line in the rightmost subfigure of Fig. 4). Understandably, such variability could pose challenges for practitioners and present obstacles to the application of many-shot learning as an effective paradigm in practice.

To address these, this paper aims to answer key research questions and proposes an effective novel approach. First, we analyze the factors driving the many-shot ICL in the *reinforced ICL* setup common in challenging reasoning tasks where we are provided with a labeled set of inputs and final labels, but the intermediate reasoning path has to be *model-generated*. We find that while ICL performance often increases with the number of shots, that improvement can often be at least partially attributed to a much smaller subset of examples that highly disproportionately contribute to the overall task performance – as we scale the number of examples, the probability of including these examples also increases. In many cases, if, however, we judiciously isolate these influential examples from the rest, the "many-shot" performance can be matched or even exceeded with this sometimes extremely small subset of well-chosen examples alone while adding more examples beyond this set often provides little benefit or even harms performance. We also argue that the findings explain some of the phenomena observed. For example, uneven influence can lead to high variance across different combinations of examples, whereas plateauing performance may occur when we run out of good examples with positive performance influences. One natural implication of these is the efficiency gains by reducing redundancy in many-shot ICL and identifying the optimized subsets. However, the natural next question to ask is whether scaling ICL examples in LLMs can still be beneficial after using up all beneficial examples identified in the previous step. We answer affirmatively to this: to still leverage LLMs' long context, these optimized, high-performing examples may serve as demonstrations to re-generate the more effective reasoning paths rationales on the train set back into the many-shot regime, which we find to often outperform both the original many-shot examples and using the optimized examples themselves. Building on these insights, we propose Bayesian Refinement and Iterative Demonstration Generation for Examples (BRIDGE), a search algorithm based on Bayesian optimization to improve many-shot ICL and *bridges* the few- and many-shot learning paradigms by automating the "optimize" and "generate" steps above iteratively. In the "optimize" step, it frames the problem as a combinatorial optimization task to discover the optimal set of demonstrations (i.e., *many-to-few*), and in the "generate" step, it uses the optimal set as seed examples to generate more examples for further performance enhancement (i.e., *few-to-many*). We demonstrate the effectiveness of BRIDGE on Gemini, Mistral and Claude models across a diverse range of tasks, including symbolic reasoning, numerical reasoning and text-to-SQL generation.

## 2 WHAT DRIVES MANY-SHOT IN-CONTEXT LEARNING PERFORMANCE?

Several previous studies on many-shot ICL (Agarwal et al., 2024; Bertsch et al., 2024) have investigated the *presence* of performance gains when we scale the number of examples. A key question that remains unanswered, though, is *what* exactly leads to this improvement. For example, it is

unknown whether the benefit is from *scaling examples itself* due to expanded knowledge in the context via more examples or because including more examples increases the probability of selecting a *small subset of disproportionately positive examples*, or a combination of the above with some task specificity. We argue that answering this question is critical – if the benefit comes from expanded knowledge from including more examples, it suggests that scaling and addressing long-context understanding challenges would dominate the end-to-end performance improvements, and future studies should aim to either include as many examples as practically possible or to imitate the behavior of the LLM as if many examples are included. If, on the other hand, the performance is dominated by a small effective subset of examples, more intelligent selection aiming to reduce redundancies and identify the high-performing subsets should outweigh näively scaling examples.

Prior work on *few*-shot setup have studied related problems such as the sensitivity to examples in the context (Zhao et al., 2021; Zhou et al., 2024b). However, it is presently unknown to what extent the findings still scale to the many-shot ICL setup because 1) in many-shot setup, the influence of each individual example would get much smaller, and 2) it is unknown whether careful example selection in the few-shot setup is still necessary if all examples can be included in the context, since by definition, any high-performing examples are subsets of *all* examples – if the long-context *LLM* is perfectly capable of identifying the most relevant pieces of information. If so, aside from other practical concerns like cost and latency, the need for users to manually curate examples may no longer be required.

**Setup.** We aim to shed insights on these important questions. We use the Gemini 1.5 Pro (Reid et al., 2024), the state-of-the-art long-context model, to focus on several representative tasks from the BBH tasks. All three tasks, as shown in by the gray lines in Fig. 1, benefit from increasing number of examples to varying degrees (in `logical_deduction`, the performance initially increases with the number of examples before plateauing and decreasing; in the other two tasks, there is a noisy but near monotonic improvement throughout) – we will test the key findings in a much more extensive collection of tasks in Sec. 4. Given the increased emphasis of modern LLMs on problem-solving and reasoning, we primarily focus on these tasks and adopt the *reinforced ICL* (Agarwal et al., 2024) setup, where we assume the availability of a labeled set of inputs and final labels to be used as many-shot demonstrations, whereas any intermediate outputs or rationales leading to the final answer are model-generated and modifiable (although we also conduct preliminary experiments in alternative setups such as low-resource machine translation in App. C.4). Lastly, we primarily focus on the tasks with the number of available labeled data up to 150-200 samples – while modern LLMs can often accommodate even more examples in the context, we focus on this range because **1)** we believe it is the most practically relevant and fills an important gap that neither few-shot ICL nor supervised (parameter-efficient) fine-tuning (which usually requires hundreds to thousands of examples) conventionally address, and **2)** while possible and of academic value, scaling beyond this range typically starts incurring significant latency and computational overhead, which scales quadratically w.r.t the input length for exact attention and is thus often practically less desired for most real-world use cases.

**Many-shot performance can still be driven by *few* high-performing examples.** A key test that would distinguish and disentangle the two possible sources of benefits from scaling mentioned at the beginning of this section is that whether we can attribute, at least to a large extent, the performance improvement from scaling examples back to a carefully selected, high-performing subset of examples with disproportionate influence. Formally, given a set of examples $\mathcal{E} = \{e_j\}_{j=1}^m$ and a performance metric to be maximized $g(\cdot) : \mathcal{P}(\mathcal{E}) \to \mathbb{R}$ (in this case, the accuracy on the validation set In this setup, the goal is to find whether we can construct a subset $\mathbf{e}^* = \{e_i^*\}_{i=1}^n \subset \mathcal{E}, \text{s.t.} n \ll m$ such that $g(\mathbf{e}^*)$ is much better than a randomly selected set of examples $\mathbf{e}$ of similar size and/or can even be comparable or better than using the full set of examples $g(\mathcal{E})$ in the context.

Whereas a conclusive test would involve enumerating and evaluating $g(\cdot)$ on the power set of $\mathcal{E}$ with $|\mathcal{P}(\mathcal{E})| = 2^{|\mathcal{E}|}$, it is clearly computationally intractable, and a natural simplification is whether we can rank the individual examples in $\mathcal{E}$ with some importance scoring function $M(e)$ to construct example subsets based on the example ranking. While many possible formulations of this are possible, here we define $M(e)$ based on imputed input gradient, which is a concept used in interpretable machine learning for importance attribution (Simonyan, 2013; Selvaraju et al., 2017; Sundararajan et al., 2017; Samek et al., 2021). In our context, directly computing input gradient is impossible as we only assume black-box LLMs without gradient backpropagation and $g(\cdot)$ is

not necessarily differentiable. To bypass these issues, we use use a sample-efficient Gaussian process regressor (GPR) (Williams & Rasmussen, 1995; 2006) to approximate $g(\cdot)$ with $\hat{g}(\cdot)$, whose input gradient $\nabla_{\mathbf{e}}\hat{g}(e)$ is analytically available: we first randomly sample $n$ subsets of $\mathcal{E}$ to give $\mathbf{e}_{1:n} = [\mathbf{e}_1, ..., \mathbf{e}_n]$, where each subset of examples is represented as a $m$-dimensional binary column vector $\mathbf{e}_i \in \{0,1\}^m$ with $\mathbf{e}_i^{(j)} = 1$ if the $j$-th example is present or $0$ otherwise; we then evaluate the performance metric of each $\mathbf{e}_i$ to obtain $\mathbf{g}_{1:n} = [g(\mathbf{e}_1), ..., g(\mathbf{e}_n)]$. We then compute and average the input gradient w.r.t. each possible $\{e_j\}_{j=1}^m \in \mathcal{E}$ to obtain an approximated marginalized importance of each example in $\mathcal{E}$(See App. A for detailed derivation of the input gradient-based score). Finally, we sort the examples based on $M(e)$ and construct subsets at regular interval from size 1 to $|\mathcal{E}|$ in both ascending and descending directions. Formally, we order $\{e_i\}_{i=1}^n$ such that $M(e_1) \leq M(e_2) \leq ... \leq M(e_n)$; the ascending and descending sets of size $t \in [1, |\mathcal{E}|]$ are given by $\mathbf{a}_t = \mathbf{e}_{1:t}$ and $\mathbf{d}_t = \mathbf{e}_{n-t:n}$ respectively. We then evaluate $g(\cdot)$ on these sets (Fig. 1).

As shown, while the gray lines (overall trend lines) often show positive correlation between performance and increasing number examples, we also observe often large gap between the green (top-$k$ examples) and the red (bottom-$k$ examples) lines, suggesting that *different sampling strategies can lead to performance differences that far outweigh the effect from naïve scaling* – e.g., if we establish an "exchange rate" between different example sets based on their imputed ordering, we can observe that including around top-10 examples (green lines) examples is as effective as or more effective than the set containing bottom-30 examples in `geometric_shapes`. More importantly, in both cases we observe that the green lines, which represent an intelligent selection strategy more sophisticated than random sampling, plateau far before the gray line, suggesting that it is possible to achieve comparable performance with much fewer number of examples: in `disambiguation_qa`, we find that using fewer than 20 top examples is almost already as good as using all 42 examples whereas subsequent additions only led to a few percent of gain, possibly within the margin of error with reshuffling (denoted by error bars on the figure). In the other tasks, we find the performance to peak much earlier and *adding more examples to the context actually led to performance deterioration*. The results suggest **1)** the fact that it is possible to match or outperform using *all* examples with *fewer*, carefully selected examples means that intelligent example selection is still relevant even with many-shot ICL, echoing findings from the recent works (Li et al., 2024b) that retrieval remains valuable for long-context models in the RAG setup; and **2)** naïvely including as many examples as possible can be suboptimal both in terms of computing cost and performance – while it is trivially true for the tasks whose performance does *not* improve monotonically with the number of examples, we show that

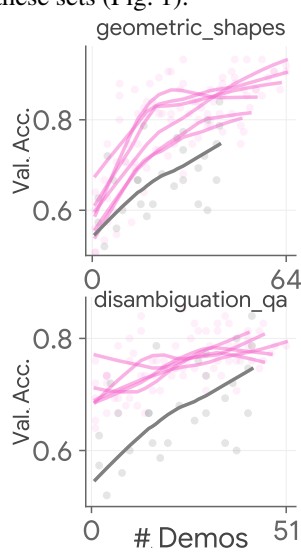

Figure 2: *Good demonstrations lead to better re-generated examples*: trendlines between accuracy and # examples; note that the re-generated examples by using top-5 examples sets as demonstrations outperform the original examples (gray line) by at all parts of the curve.

it can even be true when it apparently *does*: e.g., on `geometric_shapes`, the near monotonic improvement overall trend (gray line) may lead someone to conclude that it is beneficial to include as many examples as possible, even though the green line representing intelligent selection saturates and starts to decline earlier.

**Can we still benefit from scaling examples?** Experiments above demonstrated the presence of redundancy in many-shot ICL, revealing that using a smaller subset of examples can often reduce this redundancy without sacrificing performance. It is, however, a pruning operation that necessarily *reduce* the input tokens consumed. This leads to a natural question: can we still benefit from scaling through *expanding*? For this question, it is important to recognize that under the reinforced ICL setup, while the inputs and labels in many-shot setups are fixed, the model-generated intermediate outputs, which represent reasoning paths, are modifiable. Given that these intermediate roles are shown to play a critical role in steering model behaviors (Wan et al., 2024), it is possible that examples previously identified as non-important or non-beneficial may be again beneficial if the model-generated rationales can be improved.

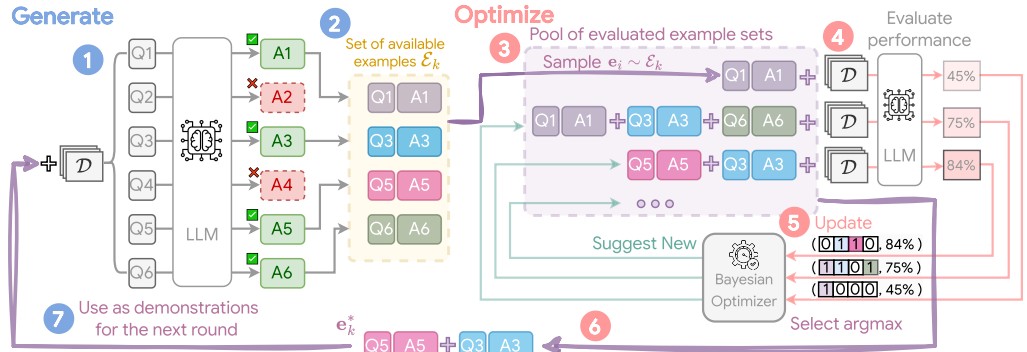

Figure 3: *Overview of* BRIDGE: With a labeled dataset $\mathcal{D}$, exemplified with 6 samples, at the Generation phase (left half), we generate initial examples by performing LLM inference on the inputs of $\mathcal{D}$ ("Q1-6") with zero-shot prompting to obtain the initial responses "A1-6", which include any intermediate outputs critical for ICL (**Step 1**). At **Step 2**, consistent with reinforced ICL in Agarwal et al. (2024), we filter the responses to retain the subset of $\mathcal{D}$ where the LLM predicted correctly to ensure the examples include correct reasoning steps to build $\mathcal{E}_k$, the pool of examples at round $k$ which form the *search space* for the subsequent Optimize step. At the Optimize step (right half), we initialize the proposed Bayesian optimizer by randomly sampling subsets $\mathbf{e}^{(0)} \subseteq \mathcal{E}_k$ as demonstrations to be **Step 3** evaluated on a held-out validation dataset ($\mathcal{D}$ can be reused for this purpose) to obtain a performance metric **Step 4**. The Bayesian optimizer (BO) is then updated with *binary vector representations* of $\mathbf{e}$ that led to this validation performance as input and the metric itself as output, and suggests a new subset of examples to be used as demonstrations for the next step **Step 5**; **Steps 4-5** are repeated (*inner loop*) until the BO budget is exhausted, after which the best evaluated set $\mathbf{e}_k^*$ is returned (**Step 6**). This set is then be used as demonstrations to generate the example pool for the next round $\mathcal{E}_{k+1}$ (**Step 7**).

To achieve so, we reuse the optimized example set from the previous steps as "seed" demonstrations for LLMs to re-generate the examples on the train set, the same set from which the optimized examples are generated. As shown by Fig. 2 where we use example set of different sizes as the seeds, the regeneration step not only increases the number of shots available but also results in better performance across the accuracy versus number-of-demonstrations trade-off.

## 3 METHODOLOGY

The findings presented above highlight a significant need for improvements that extend beyond simply increasing the number of examples straightforwardly. Instead, identifying the most useful example subset $\mathbf{e}^*$ is crucial both for effective cost-performance trade-off and for better reasoning path generation for more effective examples. Based on these insights, we propose *Bayesian Refinement and Iterative Demonstration Generation for Examples*, or BRIDGE in short (described in Algorithm 1 and depicted in Fig. 3, an optimization algorithm aiming to enhance many-shot ICL with intelligent example selection and iterative example generation. At a high level, the outer loop of BRIDGE is structured in two alternating steps of "optimize" and "generate". In the "optimize" step, the algorithm focuses on discovering the optimal subset of examples $\mathbf{e}^*$ via a carefully-designed (for low complexity, robustness to overfitting and budget control) *Bayesian optimization algorithm* that naturally leverages the GPR surrogate used in Sec. 2; in the "generate" step, BRIDGE utilizes the optimized subset as seed demonstrations to align the model with the best performing examples seen so far to re-generate new reasoning paths as an integral part of more effective examples back to the many-shot regime to leverage the long context. The two steps are iteratively repeated to progressively refine the examples.

**Optimize step.** While effective, directly using the importance scoring approach from Sec. 2 to identify the $\mathbf{e}^*$ would require us to set the optimal number of examples to select $||\mathbf{e}^*||$ as a hyperparameter, the optimal value of which is task specific. Furthermore, a key motivation for the importance-based ranking in Sec. 2 is to attribute performance to *individual* examples; this is, however, not required if we simply would like to find an optimal *subset* $\mathbf{e}^*$. To nevertheless use the GPR surrogate in Sec. 2 which has shown an impressive sample-efficient, modelling capability, we propose to use Bayesian optimization (BO) (Garnett, 2023; Frazier, 2018), a sample-efficient black-box optimization algorithm that has recently shown promise in combinatorial problems (Daulton et al.,

**Algorithm 1** BRIDGE.

1: **Input**: train set $\mathcal{D}_t$, validation set $\mathcal{D}_v$ (**can be the same as the train set**), number of iteration rounds $K \in \mathbb{N}$ (*outer-loop*), evaluation budget for BO per iteration $n_{eval}$ (*inner-loop*).
2: **Output**: Optimized set of examples $\mathcal{E}^*$.
3: [Generate] Generate the pool of initial examples $\mathcal{E}_0$ by predicting the LLM on the **train** set with zero-shot prompting or few-shot prompting (if handwritten few-shot demonstrations are available). Each instance in $\mathcal{E}_0$ is a concatenation of {input, model-generated reasoning, final outputs} for the subset of the train set where the model obtained the correct prediction.
4: **for** $k \in \{1, ..., K\}$ (**Outer** loop) **do**
5:     [Optimize] Run Bayesian optimization (calling subroutine Algorithm 2 on the **validation set** to obtain $\mathbf{e}_k^* \leftarrow$ BayesOpt($n_{eval}=n_{eval}, \mathcal{E}=\mathcal{E}_k$).
6:     [Generate] **Re-generate** examples $\mathcal{E}_k$ by re-predicting the LLM on the **train** set, but with the optimized examples $\mathbf{e}_k^*$ from the previous step as demonstrations; the {inputs, model-generated reasoning, output}-tuples are concatenated to form the new set of examples $\mathcal{E}_k$ for the next [Optimize] step.
7: **end for**
8: **return** Optimized example set $\mathcal{E}^*$ after $K$ rounds.

**Algorithm 2** Budget-controlled BO subroutine with random scalarization (BayesOpt).

1: **Input**: Evaluation budget for BO per iteration $n_{eval}$ (*inner-loop*), full set of available samples $\mathcal{E}$, number of random initializations $n_{init} = \min(16, n_{eval}/2)$.
2: **Output**: Optimized set of examples $\mathbf{e}^* \subseteq \mathcal{E}_t$.
3: Randomly generate $n_{init}$ subsets $\mathbf{e}_{1:n_{init}} := \{\mathbf{e_1}, ..., \mathbf{e}_{n_{init}}\}$ with each $\mathbf{e} \sim \{0, 1\}^{|\mathcal{E}_t|}$ s.t. $|\mathbf{e}| \sim$ Uniform$(1, |\mathcal{E}_t|)$.
4: Evaluate $\mathbf{g}_{1:n_{init}} = [g(\mathbf{e}_1, ..., \mathbf{e}_{n_{init}})]^\top$ and fit a $\mathcal{GP}$ on $\mathbf{e}_{1:n_{init}}$ as inputs and $\mathbf{g}_{1:n_{init}}$ as outputs. Set $\mathcal{D}_0 \leftarrow \{\mathbf{e}_{1:n_{init}}, \mathbf{g}_{1:n_{init}}\}$
5: **for** $t \in \{n_{init}, ..., n_{eval}\}$ (**Inner** loop) **do**
6:     Sample a random scalarization value $\beta_t \sim$ Uniform$(0, 1)$ and compute the scalarized objective *of this iteration* $h_t(\mathbf{e}) = \text{TCH}(\beta_t, [g(\mathbf{e}), |\mathbf{e}|])$.
7:     Compute $\mathbf{h}_{1:t}$ for all previously evaluated points $\mathcal{D}_{t-1}$, fit a GPR $\mathcal{GP}_t$ on $[\mathbf{e}_{1:t}, \mathbf{h}_{1:t}]$ and obtain the next configuration to evaluate by maximizing the *acquisition function* $\alpha(\cdot)$: $\mathbf{e}_t = \arg\max_{\mathbf{e} \subseteq \mathcal{E}} \alpha(\mathbf{e} \mid \mathcal{GP}_t)$.
8:     Evaluate $g(\cdot)$ with $\mathbf{e}_t$ and augment $\mathcal{D}_t \leftarrow \mathcal{D}_{t-1} \cup (\mathbf{e_t}, g(\mathbf{e}_t))$
9: **end for**
10: **return** $\mathbf{e}^* = \arg\max_{\mathbf{e} \in \mathcal{D}} g(\mathbf{e})$.

2022; Wan et al., 2021); it naturally synergizes with the GP surrogate yet automatically strikes a balance between exploration and exploitation to discover $\mathbf{e}^*$ without requiring us to set $||\mathbf{e}^*||$ beforehand, although BRIDGE is also compatible with alternative methods as drop-in replacement of the "Optimize" step, which we investigate in detail in App. C.1.

Instead of consuming the entire query budget by sampling randomly, as illustrated by Algorithm 2, BO only requires some initializing samples to warm-start (Step 3). Afterward, it guides exploration by iteratively (re)fitting a GPR with the previous observed inputs and outputs so far. Formally, at iteration $t \in [1, T]$, we have evaluated $g(\cdot)$ $t$ times at $\mathbf{e}_{1:t} = [\mathbf{e}_1, ..., \mathbf{e}_t]^\top$ with observed values $\mathbf{g}_{1:t}$. Whereas a straightforward application of BO would directly train a GP on $[\mathbf{e}_{1:t}, \mathbf{g}_{1:t}]$ as inputs-outputs and perform BO with $g(\cdot)$ as the objective function directly, a subtle but important distinction here is that our goal is to identify a subset $\mathbf{e}^*$ that, *when used as demonstrations on the train set*, generates to the most effective examples on the validation set, rather to simply find the highest-performing $\mathbf{e}^*$ on the validation set. While we expect the two objectives to be correlated (i.e., $\mathbf{e}$ that led to high validation performance is also likely to generate better samples on the train set), we also empirically find it is desirable to encourage $\mathbf{e}^*$ to have a smaller cardinality akin to a $\ell_0$ regularization to reduce overfitting on the validation set and to discourage memorization in subsequent generations from the previous example set $\mathcal{E}_{t-1}$ of which $\mathbf{e}^*$ is a subset. To achieve so, we augment the performance maximization $\max g(\mathbf{e})$ with a *sparsity objective* which counts the number of non-zero elements in $\mathbf{e}$: $\min \sum_j e^{(j)}$ – this transforms the problem into a *bi-objective* optimization problem, where instead of maximizing for the validation performance only, we also encourage sparsity as regularization. Practically, we solve the problem with *random scalarization* (Paria et al., 2020; Knowles, 2006). Specifically, as hinted in Step 7 of Algorithm 2, at each BO iteration, we first sample a random scalar $\beta_t \sim \text{Unif}(\beta_{LB}, \beta_{UB})$ that determines the weight of the performance objective $g(\cdot)$ of the $t$-th BO iteration (the weight of the sparsity objective is given by $1 - \beta_t$) and $\{\beta_{LB}, \beta_{UB}\}$ denote the lower and upper bounds of the weight for $g(\cdot)$ which are set to $\{0.25, 1\}$ by default. With this $\beta_t$, we then aggregate the vector objective $[g(\mathbf{e}), \sum_j e^{(j)}]$ back to a scalar $h_t(\mathbf{e})$ via *Tchebyshev scalarization* (TCH), a theoretically well-founded scalarization scheme common in multi-objective optimization (Chugh, 2020; Steuer & Choo, 1983; Bowman Jr, 1976) given by:

$$h_t(\mathbf{e}) = \max \left\{ \beta_t \big(g(\mathbf{e}) - \max\{g(\mathbf{e_1}), ..., g(\mathbf{e_t})\}\big), -(1 - \beta_t) \sum_j e^{(j)} \right\}, \quad (1)$$

where the minus sign before the last term is to cast the sparsity objective as maximization. We opt for random scalarization that differs step to step instead of a fixed scalarization weight or any hard constraint on $\sum_j e^{(j)}$ to retain the flexibility of exploring the entire Pareto front, since the

Table 1: Test accuracy of `gemini-1.5-pro-001` on selected BBH tasks with different prompting approaches. "All" refers to using the *entire* labeled set of 75 examples as demonstrations ("Direct": using all input-final answer pairs *without* any model-generated content; "CoT": using all input-rationale-final answer triplet, where the rationale is model-generated; "Infill": using all input-rationale-final answer triplet, where the rationale is *filled in* by prompting the model to generate the intermediate steps given the inputs *and* ground-truth answers); "Reinf. ICL" refers to reinforced many-shot ICL where we include the subset of train set that the LLM answered correctly under zero-shot as demonstrations; "Iterative Reinf." refers to the iterative variant of reinforced many-shot ICL where we directly use all the generated correct examples from the previous round as demonstrations for the next round without the optimize step, and the different columns of BRIDGE show the evolution of test accuracy at different milestones: e.g., 1O refers the results with optimized $\mathbf{e}_1^*$ from initial examples $\mathcal{E}_0$ as demonstrations (in general, we have $\mathbf{e}_k^* \subseteq \mathcal{E}_{k-1}$), and 1G refers to the results using $\mathcal{E}_1$ generated by re-evaluating the train set with $\mathbf{e}_1^*$ as demonstrations. All results shown are averaged across 4 random seeds with the standard deviation (stdev) denoted in the subscript. Best and second-best results along each row are **bolded** and underlined, respectively (ties are broken by favoring the result with lower stdev).

| Tasks | Direct | All CoT | Infill | Reinf. ICL | Iterative Reinf. | | BRIDGE (Ours) | | | | |
|---|---|---|---|---|---|---|---|---|---|---|---|
| # Iterations | - | 0 | 0 | 0 | 1 | 2 | 1O | 1G | 2O | 2G | 3O |
| causal_judgement | 61.0$_{4.7}$ | 62.7$_{2.1}$ | 68.0$_{2.8}$ | 66.3$_{4.8}$ | 68.7$_{1.9}$ | 69.3$_{2.7}$ | 68.3$_{1.5}$ | 62.7$_{1.6}$ | 59.7$_{1.5}$ | **72.0$_{0.0}$** | 70.0$_{2.0}$ |
| date_understanding | 87.2$_{2.0}$ | 86.0$_{2.3}$ | 94.8$_{1.8}$ | 88.8$_{2.5}$ | 93.0$_{1.0}$ | 94.9$_{1.3}$ | 92.2$_{1.5}$ | **97.0$_{0.7}$** | 94.8$_{1.9}$ | 95.0$_{1.2}$ | 95.5$_{1.8}$ |
| disambiguation_qa | 74.2$_{2.2}$ | 63.3$_{1.1}$ | 72.3$_{2.0}$ | 76.8$_{2.4}$ | 74.6$_{1.4}$ | 75.1$_{1.5}$ | 71.8$_{2.4}$ | 77.5$_{3.6}$ | 80.5$_{1.8}$ | **81.3$_{2.9}$** | 78.8$_{1.5}$ |
| dyck_languages | 16.8$_{2.9}$ | 39.0$_{3.7}$ | 24.5$_{2.9}$ | 55.5$_{3.6}$ | 64.4$_{5.3}$ | 74.4$_{3.6}$ | 49.2$_{2.7}$ | 76.2$_{3.8}$ | **80.0$_{2.7}$** | 77.5$_{1.1}$ | 76.8$_{3.8}$ |
| formal_fallacies | 82.8$_{3.7}$ | 86.8$_{1.3}$ | 84.3$_{2.8}$ | 86.2$_{1.1}$ | 88.1$_{0.9}$ | 89.4$_{1.4}$ | 86.0$_{2.1}$ | 85.0$_{2.5}$ | **90.8$_{2.3}$** | 90.8$_{2.8}$ | 88.2$_{2.3}$ |
| geometric_shapes | 69.0$_{4.1}$ | 61.8$_{4.2}$ | 73.5$_{2.3}$ | 80.2$_{2.8}$ | 81.0$_{2.5}$ | 82.3$_{1.7}$ | 78.5$_{2.1}$ | 82.5$_{3.6}$ | 89.2$_{3.8}$ | **92.3$_{1.1}$** | 89.2$_{0.8}$ |
| hyperbaton | 70.8$_{4.1}$ | 93.2$_{3.1}$ | 89.5$_{2.6}$ | 90.2$_{1.1}$ | 91.5$_{2.2}$ | 86.2$_{2.5}$ | 96.5$_{0.9}$ | 94.2$_{1.5}$ | 94.8$_{2.8}$ | 96.5$_{0.5}$ | **97.2$_{0.4}$** |
| logical_deduction (7) | 56.8$_{4.4}$ | 63.0$_{7.4}$ | 69.8$_{5.9}$ | 65.8$_{3.5}$ | 68.9$_{2.6}$ | 69.5$_{2.9}$ | 70.2$_{1.5}$ | 70.8$_{4.5}$ | **71.7$_{3.7}$** | 71.5$_{1.8}$ | 69.2$_{2.2}$ |
| movie_recommendation | **75.0$_{1.0}$** | 63.7$_{2.2}$ | 68.0$_{2.8}$ | 65.2$_{1.6}$ | 68.8$_{2.0}$ | 82.0$_{1.9}$ | 67.0$_{1.2}$ | 69.5$_{0.5}$ | 69.3$_{3.1}$ | 72.8$_{1.8}$ | 67.0$_{1.2}$ |
| multistep_arithmetic_two | 86.5$_{2.2}$ | 96.8$_{0.8}$ | 88.8$_{1.8}$ | 96.5$_{0.5}$ | 95.9$_{0.8}$ | 94.5$_{1.3}$ | 96.2$_{0.8}$ | 94.5$_{1.1}$ | 97.0$_{0.7}$ | **98.0$_{0.7}$** | 96.8$_{1.8}$ |
| object_counting | 92.5$_{2.3}$ | 84.8$_{4.3}$ | 95.3$_{1.1}$ | 95.5$_{0.9}$ | 95.8$_{2.2}$ | 95.1$_{1.6}$ | **96.2$_{0.4}$** | 96.0$_{1.9}$ | 94.5$_{1.1}$ | 94.2$_{0.4}$ | 95.0$_{0.7}$ |
| ruin_names | 85.2$_{3.1}$ | 85.5$_{2.1}$ | 89.8$_{1.6}$ | 89.8$_{1.9}$ | 88.6$_{1.5}$ | 90.5$_{0.9}$ | **90.8$_{1.1}$** | 88.8$_{1.7}$ | 89.2$_{1.5}$ | 88.8$_{2.4}$ | 90.3$_{0.8}$ |
| salient_translation_error_detection | 66.0$_{2.4}$ | 56.2$_{1.5}$ | 72.5$_{0.5}$ | 69.0$_{1.6}$ | 73.8$_{1.1}$ | 73.4$_{1.3}$ | 68.8$_{0.8}$ | 71.0$_{0.7}$ | 69.5$_{2.2}$ | 74.0$_{0.7}$ | **74.5$_{1.1}$** |
| snarks | 94.1$_{1.8}$ | 95.5$_{2.3}$ | 95.1$_{0.6}$ | 92.7$_{3.2}$ | 94.3$_{1.9}$ | 95.5$_{1.5}$ | 93.4$_{3.0}$ | 95.8$_{0.0}$ | 95.1$_{1.6}$ | 96.9$_{1.5}$ | **97.6$_{1.8}$** |
| sports_understanding | 93.8$_{1.3}$ | 94.2$_{1.3}$ | 95.0$_{0.7}$ | 93.0$_{1.4}$ | 94.1$_{0.9}$ | 95.4$_{1.2}$ | 92.8$_{1.9}$ | **97.0$_{1.2}$** | 96.2$_{0.8}$ | 95.8$_{0.4}$ | 95.8$_{0.8}$ |
| tracking_shuffled_objects (7) | 76.0$_{7.2}$ | 52.5$_{2.1}$ | 64.3$_{2.8}$ | 62.3$_{4.2}$ | 64.5$_{2.2}$ | 65.5$_{4.6}$ | 95.8$_{0.4}$ | 95.0$_{1.2}$ | **100.0$_{0.0}$** | 97.0$_{0.7}$ | 99.5$_{0.5}$ |
| Average | 74.22 | 74.06 | 78.70 | 79.61 | 81.61 | 82.37 | 82.11 | 84.61 | 85.77 | **87.13** | 86.33 |

exact relation between the number of samples and performance can differ across tasks. Since $\beta_t$ is in general different for each $t$, we then compute $\mathbf{h}_t = [h_t(\mathbf{e}_1), ..., h_t(\mathbf{e}_t)]$ on previously evaluated outputs and fit a GP on $\mathcal{H}_t := [\mathbf{e}_{1:t}, \mathbf{h}_t]$, which induces a Gaussian posterior predictive distribution with mean and variance at any $\mathbf{e} \subseteq \mathcal{E}$ (we use $\hat{h}_t$ to denote that it is the GP approximation of the actual function $h_t$):

$$\mathbb{E}_{\hat{h}_t(\mathbf{e})|\mathcal{H}_t}[\hat{h}_t(\mathbf{e})] = \mathbf{k}_t(\mathbf{K} + \eta^2\mathbf{I})^{-1}\mathbf{h}_t, \quad \mathbb{V}_{\hat{h}_t(\mathbf{e})|\mathcal{H}_t}[\hat{h}_t(\mathbf{e})] = k(\mathbf{e}, \mathbf{e}) - \mathbf{k}_t(\mathbf{K} + \eta^2\mathbf{I})^{-1}\mathbf{k}_t^\top, \quad (2)$$

where $\mathbf{k}_t = [k(\mathbf{e}, \mathbf{e}_1), ..., k(\mathbf{e}, \mathbf{e}_t)]$ and $k(\cdot, \cdot)$ is the covariance function of the GP (we use Matern 2.5 by default) which measures the similarity between two inputs – in our case, it is a function of the number of overlapping examples between two subsets of examples $\mathbf{e}, \mathbf{e}' \subseteq \mathcal{E}$. To select the next configuration to evaluate $\mathbf{e}_k$, the BO optimizes an *acquisition function*, another key component of BO that automatically trade off exploration and exploitation. At each inner-loop BO iteration, we choose the maximizer of the *expected improvement* (EI) (Zhan & Xing, 2020) for the next iteration $\mathbf{e}_t$: $\mathbf{e}_t = \arg\max_{\mathbf{e} \subseteq \mathcal{E}} \alpha(\mathbf{e}) = \arg\max_{\mathbf{e} \subseteq \mathcal{E}} \mathbb{E}_{\hat{h}_t(\mathbf{e})|\mathcal{H}_t}\big[\max\{0, \hat{h}_t(\mathbf{e}) - \max_{t' \in \{1, t\}} \hat{h}_t(\mathbf{e}_{t'})\}\big]$.

**Generate step.** At each *outer-loop* round $k \in \{1, ..., K\}$, given the optimized $\mathbf{e}_k^*$ as demonstrations, we regenerate and replace the example pool with the correct predictions and their generated rationales $\mathcal{E}_k \leftarrow f_{\text{LLM}}(\mathcal{D}_t, \mathbf{e}_k^* \subseteq \mathcal{E}_{k-1})$ for subsequent optimize step.

## 4 EXPERIMENTS

**Model and evaluation data.** We conduct experiments on an extensive collection of tasks requiring different set of skills task difficulty on two Gemini 1.5 models (`gemini-1.5-pro-001` and `gemini-1.5-flash-001`) while also testing key findings on Mistral family of models: Mistral NeMo (`mistral-nemo-12b`) and Mistral Large (`mistral-large-2407`), and Claude 3.5 Sonnet: **1)** BIG-Bench Hard (BBH) tasks encompassing a wide range of challenging numerical reasoning, commonsense problem-solving, logical deduction and tabular reasoning tasks – we particularly focus on the subset of 16 BBH tasks where the model performances have not saturated; **2)**

Table 2: Test accuracy of `gemini-1.5-pro-001` on MATH and GSM-Hard datasets. Refer to the captions of Table 1 for detailed explanations.

| Tasks | Reinf. ICL | Iterative Reinf. | | | | BRIDGE *(Ours)* | | |
|---|---|---|---|---|---|---|---|---|
| # Iterations | 0 | 1 | 2 | 1O | 1G | 2O | 2G | 3O |
| Hendryck's MATH | $63.75_{0.5}$ | $63.60_{0.9}$ | $63.60_{1.1}$ | $62.60_{1.3}$ | $63.00_{1.2}$ | $63.85_{1.1}$ | $\mathbf{64.65}_{0.3}$ | $\underline{64.40}_{0.9}$ |
| GSM-Hard | $69.88_{0.8}$ | $69.84_{0.4}$ | $69.33_{0.3}$ | $71.89_{0.4}$ | $71.31_{0.4}$ | $71.81_{0.4}$ | $\mathbf{73.32}_{0.4}$ | $\underline{72.50}_{0.6}$ |

Hendryck's MATH (Hendrycks et al., 2021), a challenging numerical reasoning dataset; **3)** GSM-Hard (Gao et al., 2022), a more challenging variant of the classical grade-school GSM-8K (Cobbe et al., 2021) with the numbers in the questions replaced with much larger and rarer ones. To further probe the utility of many-shot learning and BRIDGE in coding tasks, we also experiment on **4)** BIRD (Li et al., 2024a), a challenging large-scale text-to-SQL generation benchmark where the LLM has to generate sqlite programs from natural language instructions that are executed on real-world databases. For all datasets, when official train-test split is not available, we randomly split the data into train and test splits; unless stated otherwise, a single unified train split is used both for the generation of demonstrations and is reused for validation (i.e., the objective of the optimize step in Algorithm 1; the test splits are held-out and only used for evaluation of the algorithm. We refer the readers to App. B for detailed descriptions, prompt templates used and evaluation protocol.

**Experimental setup.** For all tasks, we run BRIDGE with $K = 3$ rounds (i.e., the number of *outer-loop* iterations in Algorithm 1) and within each round, we allow for $n_{\text{eval}} = 32$ evaluations on the validation set (i.e., the number of *inner-loop* iterations in Algorithm 2) and we report the results at the end of each "optimize" and "generate" steps to visualize the iteration process. For baselines, we consider **1)** using all provided examples and we consider three variants: a) using query-target *only* without any generated rationales (Direct), b) first prompt the LLM to generate rationales and answers, and use the concatenation of query-rationale-target as demonstrations, *regardless of whether the rationale led to the correct answer* (CoT), and c) prompting the LLM with both the query *and the final, ground-truth answer* to *fill in* the rationale – this technique has been variously referred to as, e.g., *infilling* (Hu et al., 2023), *rationalization* (Zelikman et al., 2022), or more generally, *teacher forcing* (Chen et al., 2025) due to its conceptual similarity to teacher forcing in recurrent neural network (RNN) training (Lamb et al., 2016) (Infill); **2)** reinforced ICL (Agarwal et al., 2024), where all available input-output pairs from the correct predictions on the train set with zero-shot prompting are used; and **3)** an iterative variant of reinforced ICL which can also be seen as BRIDGE without the optimize step: while we repeat the generation process on the train set $K = 3$ times, we do not first aim to select the optimized subset but instead use the entire generated examples from the previous step as demonstrations $\mathcal{E}_k \leftarrow f_{\text{LLM}}(\mathcal{D}_t, \mathcal{E}_{k-1})$.

**Results and discussions.** We show the test accuracy on the BBH tasks in Table 1 (`gemini-1.5-pro-001`), Table 3 (`gemini-1.5-flash`), Table 14 (Mistral) and Table 15 (Claude 3.5 Sonnet) (the latter two tables are in App. C.5). On MATH and GSM-Hard datasets, we show the Gemini 1.5 Pro results in Table 2. We observe that naïve many-shot scaling is in general ineffective and is outperformed by reinforced ICL; BRIDGE, however, outperforms the base reinforced many-shot ICL by more than 7% and 3% on Tables 1 and 3, respectively, and the extent of outperformance over the "Iterative reinforced ICL", which leads to moderate improvements on BBH with Gemini Pro but no significant performance gains on MATH, GSM-Hard and BBH with Gemini Flash. Both demonstrate that optimize is an integral component of BRIDGE and implicitly validates the findings in Sec. 3 that *many-shot performance can be driven by few disproportionately influential examples*, which constitutes a core motivation for our method. Barring some expected task-specific fluctuations, in both Tables 1 and 3, we also observe consistent and monotonic performance improvement as BRIDGE progresses over the successive optimize and generate steps, eventually peaking at 2G on Gemini Pro and 2O on Gemini Flash (although the performance difference between 2G and 2O on Gemini Flash is negligible and likely within margin of error) – based on the overall results, we recommend stopping BRIDGE at 2G or 2O. Interestingly, we observe that in both cases, an additional optimize step (i.e., the 3O column) somewhat degrades performance – our hypothesis is that as BRIDGE progresses, the generated examples become more aligned with the optimal behavior and the degree of redundancy as we observed in Sec. 2 reduces, and it becomes more difficult to squeeze the number of examples without harming task performance – indeed, from Fig.4 where we concretely analyze the behavior of the LLM in different tasks by evaluating the LLM under random subsets of $\mathcal{E}_0, ..., \mathcal{E}_2$ as demonstrations in held-out splits, we observe that the benefit from naïvely scaling examples under the base reinforced many-shot ICL (denoted by red lines) can be highly unstable across tasks: from the different subfigures of Fig. 4, we find the performance to consistently improve with more examples (leftmost), improve then plateau (middle

Table 3: Test accuracy of `gemini-1.5-flash-001` on BBH tasks.

| Tasks | Direct | All CoT | Infill | Reinf. ICL | Iterative Reinf. | | BRIDGE (Ours) | | | | |
|---|---|---|---|---|---|---|---|---|---|---|---|
| # Iterations | - | 0 | 0 | 0 | 1 | 2 | 1o | 1G | 2o | 2G | 3o |
| causal_judgement | $55.0_{5.0}$ | $57.7_{1.1}$ | $62.7_{2.7}$ | $66.0_{3.6}$ | **$67.7_{2.0}$** | $\underline{66.7_{1.6}}$ | $69.3_{2.7}$ | $66.0_{2.0}$ | $63.3_{1.5}$ | $65.0_{1.6}$ | $65.3_{1.5}$ |
| date_understanding | $84.8_{4.2}$ | $83.3_{1.3}$ | $89.3_{0.8}$ | $84.5_{2.3}$ | $86.8_{0.8}$ | $87.3_{0.8}$ | $85.0_{1.3}$ | $90.5_{0.5}$ | $\underline{91.5_{0.4}}$ | $90.8_{0.7}$ | **$92.5_{0.8}$** |
| disambiguation_qa | $68.8_{7.2}$ | $54.2_{1.5}$ | $69.0_{2.2}$ | $75.5_{0.5}$ | $77.8_{1.6}$ | $\underline{78.5_{3.5}}$ | $77.5_{1.3}$ | **$79.0_{1.1}$** | $77.5_{1.2}$ | $76.3_{0.8}$ | $74.3_{1.1}$ |
| dyck_languages | $46.0_{9.5}$ | $19.5_{7.0}$ | $31.3_{3.3}$ | **$66.8_{1.9}$** | $61.3_{2.6}$ | $60.0_{1.9}$ | $63.3_{2.0}$ | $62.0_{1.7}$ | $\underline{64.5_{1.8}}$ | $62.8_{2.4}$ | $61.8_{3.8}$ |
| formal_fallacies | $75.8_{1.9}$ | $74.0_{1.2}$ | $76.3_{1.1}$ | $77.3_{0.4}$ | $74.8_{1.9}$ | $72.5_{1.7}$ | **$78.3_{1.3}$** | $77.3_{1.5}$ | $75.5_{1.7}$ | $\underline{78.3_{1.8}}$ | $76.3_{0.8}$ |
| geometric_shapes | $45.8_{1.5}$ | $74.2_{4.1}$ | $71.3_{2.4}$ | $86.0_{1.9}$ | $93.8_{0.8}$ | $93.3_{1.5}$ | $93.8_{2.5}$ | $94.0_{4.2}$ | $95.5_{1.1}$ | $\underline{97.0_{0.0}}$ | **$98.0_{0.0}$** |
| hyperbaton | $87.0_{3.1}$ | $88.5_{1.5}$ | $93.5_{1.1}$ | $88.5_{1.5}$ | $\underline{95.5_{1.1}}$ | $93.3_{1.5}$ | $86.5_{7.6}$ | $\underline{95.5_{1.1}}$ | **$95.8_{0.8}$** | $94.8_{0.4}$ | $93.3_{1.5}$ |
| logical_deduction (7) | $37.5_{3.3}$ | $41.0_{1.9}$ | $57.0_{2.7}$ | $59.5_{3.4}$ | $61.9_{1.9}$ | $57.5_{4.7}$ | $61.8_{5.1}$ | $57.5_{1.1}$ | $\underline{70.5_{0.9}}$ | $66.5_{1.1}$ | **$75.0_{0.7}$** |
| movie_recommendation | $\underline{80.5_{3.3}}$ | $56.2_{0.8}$ | **$92.0_{1.9}$** | $67.0_{1.2}$ | $75.8_{1.3}$ | $75.8_{2.9}$ | $70.3_{2.3}$ | $73.3_{2.3}$ | $77.3_{1.5}$ | $78.8_{2.0}$ | $72.8_{3.2}$ |
| multistep_arithmetic_two | $55.0_{21.3}$ | $84.0_{2.9}$ | $89.0_{1.9}$ | $91.3_{0.8}$ | $94.0_{1.4}$ | $92.5_{1.8}$ | $96.3_{2.3}$ | $96.8_{0.4}$ | **$97.8_{0.4}$** | $94.8_{0.8}$ | $95.8_{0.4}$ |
| object_counting | $66.0_{2.7}$ | $91.3_{2.0}$ | $87.5_{2.3}$ | $93.3_{0.4}$ | $93.5_{1.5}$ | $92.5_{1.1}$ | $92.8_{1.9}$ | $93.8_{2.3}$ | **$95.5_{0.5}$** | $93.0_{1.2}$ | $\underline{93.8_{0.4}}$ |
| ruin_names | $83.2_{1.3}$ | $86.2_{1.3}$ | $88.0_{1.9}$ | $86.5_{1.8}$ | $89.5_{0.9}$ | $86.8_{0.8}$ | $89.3_{0.4}$ | $89.3_{0.8}$ | $87.0_{1.2}$ | **$90.3_{0.8}$** | $\underline{90.0_{1.2}}$ |
| salient_translation_error_detection | $62.0_{3.7}$ | $58.8_{2.0}$ | $65.3_{1.3}$ | $64.8_{1.5}$ | **$71.5_{2.2}$** | $64.0_{2.9}$ | $62.8_{0.8}$ | $\underline{71.0_{0.7}}$ | $69.8_{2.0}$ | $69.0_{0.7}$ | $67.3_{0.4}$ |
| snarks | $81.2_{0.7}$ | **$92.0_{1.2}$** | $80.9_{1.2}$ | $89.2_{1.8}$ | $88.9_{2.2}$ | $86.5_{1.5}$ | $88.9_{2.0}$ | $89.9_{1.8}$ | $89.6_{0.7}$ | $\underline{90.6_{0.6}}$ | $83.7_{3.5}$ |
| sports_understanding | $92.5_{1.5}$ | $91.5_{0.5}$ | $95.8_{0.4}$ | $\underline{95.8_{0.8}}$ | $95.5_{0.5}$ | **$96.3_{1.1}$** | $93.3_{1.1}$ | $95.3_{0.4}$ | $91.8_{0.4}$ | $95.0_{1.2}$ | $95.0_{0.0}$ |
| tracking_shuffled_objects (7) | $63.3_{5.4}$ | $72.3_{6.0}$ | $32.8_{1.9}$ | $92.2_{3.1}$ | $83.5_{1.1}$ | $80.0_{1.6}$ | $\underline{98.0_{0.7}}$ | $93.8_{2.2}$ | **$98.0_{0.0}$** | $97.8_{0.4}$ | $97.5_{0.5}$ |
| *Average* | 67.77 | 70.29 | 73.83 | 80.25 | 81.91 | 80.72 | 81.61 | 82.79 | **83.79** | $\underline{83.77}$ | 83.25 |

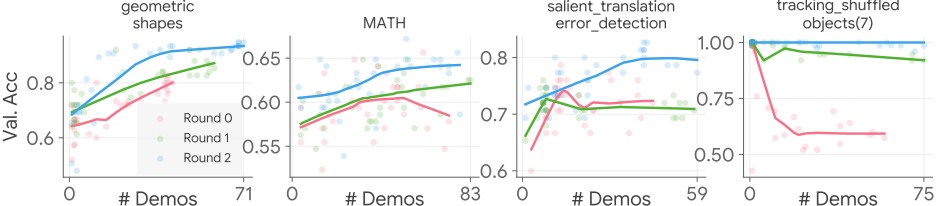

Figure 4: *Benefits from scaling examples naïvely (red lines) is very task specific, but each iteration of* BRIDGE *addresses it to a considerable degree by continually improving upon the previous round:* We randomly sample subsets of example pool $\mathcal{E}_k \, \forall \, k \in \{0 \text{ (i.e., original examples generated with handcraft few-shot or zero-shot)}, 1, 2\}$ and evaluate them on a held-out set in four representative tasks exhibiting different model behavior to example scaling. The trendlines are moving regressions fitted with LOWESS. Refers to additional figures in App. C.3.

two figures) and even simply deteriorate with more examples (rightmost) – whereas the latter two cases are direct manifestations that not all examples contribute positively to many-shot ICL and naïvely scaling examples is suboptimal, we note that it remains true even in the former case where there is an apparent strong, positive correlation between number of demos and performance, as we demonstrated in Sec. 2. Remarkably, BRIDGE alleviate the instability with each round of BRIDGE continually improving upon the previous round – in cases where scaling examples is already beneficial (`geometric_shapes`, leftmost figure), subsequent rounds of BRIDGE led to much better performance-cost trade-offs with the blue and green lines dominating over the red, whereas in other cases, BRIDGE often "delays" the saturation point (e.g., `salient_translation`) or at least ensure more examples does not lead to deterioration (e.g., `tracking_shuffled_objects`).

On BIRD dataset, we show the results in Table 4. Given the presence of a large training set (more than 9000 samples), we also compare against parameter-efficient supervised fine-tuning (PEFT) (Han et al., 2024), where we fine-tune the same target LLM with LoRA (Hu et al., 2021) on either the entire training set or using a number of train samples sub-sampled from the full training set. We observe that whereas the few-shot CHASE prompt effectively improves upon the baseline zero-shot direct prompting, additional rounds of BRIDGE led to further gains. The comparison against LoRA also demonstrates the potential of BRIDGE as an alternative to PEFT at least in certain scenarios. When provided with a similar number of labeled samples (i.e., $n_{\text{train}} = 256$), we observe that LoRA performs much worse, and it only outperforms BRIDGE when using up the *entire* train set for training.

Table 4: Execution accuracy on the BIRD dev set with `gemini-1.5-pro-001`. {S, M, C} refer to the accuracy aggregated across {Simple, Moderate, Challenging}-level problems based on assigned difficulty.

| Method | Exec. Acc. | Breakdown | | |
|---|---|---|---|---|
| | | S | M | C |
| Direct | 57.7 | 64.0 | 49.4 | 44.1 |
| CHASE prompt | 60.1 | 67.2 | 51.9 | 40.7 |
| CHASE + BRIDGE | | | | |
| *Round 0* | 59.1 | 65.7 | 51.3 | 42.1 |
| *Round 1* | 61.2 | 68.6 | 50.6 | 48.3 |
| *Round 2* | $\underline{62.0}$ | 68.5 | 53.0 | 49.0 |
| PEFT (LoRA) | | | | |
| $n_{\text{train}} = 256$ | 58.2 | 64.0 | 52.2 | 40.7 |
| $n_{\text{train}} = 1024$ | 60.2 | 66.6 | 53.0 | 42.1 |
| $n_{\text{train}} = 4096$ | 61.3 | 67.5 | 53.9 | 46.2 |
| $n_{\text{train}} = 9428$ (All) | **63.8** | 68.6 | 58.8 | 48.9 |

## 5 RELATED WORK

**Scaling ICL.** Before the advent of the long-context LLMs, early efforts in scaling ICL often study LLMs customized for long context (Li et al., 2023) or require architectural changes assuming white-box model access (Hao et al., 2022). However, the tasks considered are often limited, e.g., to conven-

tional, discriminative tasks like sentiment classification rather than generative tasks as considered in this work. Furthermore, these often study LLMs that are merely capable with handling many examples, but their behavior may differ significantly to modern, natively long-context LLMs that may actively *take advantage of* the context – indeed, both these works show mixed results, even significant performance deterioration when scaling up the number of examples, a phenomenon not seen in modern long-context LLMs like Gemini and Claude. Recent works like Agarwal et al. (2024) and Bertsch et al. (2024), on the other hand, reported significant gains in scaling ICL to hundreds or more examples and provided important motivation for our work. However, as mentioned in Sec. 2, these works primarily demonstrate the existence of the benefit from scaling but did not focus on investigate the sources of the gain or improving the cost-effectiveness of many-shot ICL. Additionally, there have also been works focusing on applications of many-shot ICL to multi-modalities (Jiang et al., 2024), LLM jail-breaking (Anil et al., 2024), detecting the risk of capturing incorrect skills (Lin & Lee, 2024), and analyzing memorization (Golchin et al., 2024).

**Example selection and expansion.** BRIDGE combines the "optimize" and "generate" steps, and there have been existing works sharing similar high-level ideas to each of the components. First, the "optimize" step can be seen as to improve the *data quality* with pruning and selection; in this regard, given that data quality is known to be one of the most influential factors for training LLMs (Xia et al., 2024), many previous works have utilized some flavor of pruning to remove redundant or harmful data samples at different stages of training, including pre-training (Marion et al., 2023) and instruction tuning (Xia et al., 2024). In ICL, as mentioned in Sec. 2, given the sensitivity of LLMs to examples, there have been works analyzing prompt sensitivity and proposing *example selection* techniques (Zhao et al., 2021; Lu et al., 2022; Zhou et al., 2024b; Wan et al., 2024). Recent work also explored heuristic-based prompt optimization based on similarity (Rubin et al., 2022; Liu et al., 2022), diversity (Levy et al., 2023; Xu et al., 2024), uncertainty (Wan et al., 2023a;b), fairness (Zhou et al., 2024a) etc. Our "generate" step, on the other hand, aims to acquire high-quality examples with the LLM itself. In this area, STaR (Zelikman et al., 2022) first proposes to bootstrap rationales from LLM with a small number of seed examples, followed by fine-tuning; Self-Instruct (Wang et al., 2023) bootstraps LLMs to instruction data. The "Reinforced ICL" technique introduced in Agarwal et al. (2024), upon which this work improves, and several recent works (Chen et al., 2023; Khattab et al., 2023; Opsahl-Ong et al., 2024) use similar technique to acquire and refine model-generated examples for ICL. Notwithstanding the similarities described, there are a few crucial differences: Almost all ICL works mentioned consider the *few*-shot setup, where selection is made necessary due to the constraint on the number of examples allowed in the context. However, we show that even in the many-shot setup where that constraint is relaxed and example selection is no longer a necessity, it can still be highly beneficial for performance and efficiency. Unlike the few-shot setup, BRIDGE is tailored for the many-shot setup with design decisions inspired by findings in Sec. 2, such as the implementation of sparsity regularization in the optimization objective to enable from scaling.

## 6 CONCLUSION

This paper focuses on understanding and enhancing the core factors underlying scaling ICL. We first provide an analysis on the nascent paradigm of many-shot ICL in LLMs and show that notwithstanding the long-context abilities of LLMs, the common practice of naïvely dumping as many examples as practically possible into the context can be both inefficient in cost and suboptimal in performance. Instead, the benefit from scaling examples can often be realized by identifying a subset of influential examples, and that subset can be used as demonstrations themselves to re-generate even more examples. Inspired by the findings, we propose BRIDGE by automatically executing the "optimize" and "generate" steps iteratively. We demonstrate that BRIDGE perform competitively on a wide range of tasks, significantly outperforming alternatives. We believe that this work builds the foundation for future research in many-shot ICL. First, we mainly focused on the restrictive *black-box* LLM setup, which is the most general and model-agnostic. However, for a more relaxed, white-box setup with access to LLM weights, it may be possible to perform optimization more efficiently – for example, it may be possible to take advantage of the internal representations of the model in reducing the cost of iterative optimization. Second, we currently focus on the "reinforced ICL" setup typical for reasoning-heavy tasks – while we have conducted experiments (e.g., low resource translation tasks) beyond this setup, further validations on other types of tasks would be valuable. Lastly, after optimization, the examples generated by BRIDGE are currently static at test time, and it would also be interesting to combine with a mechanism for sample-dependent ICL optimization to further enhance performance and reduce cost – we defer these important directions to future work.

ACKNOWLEDGMENTS

We thank all colleagues from Google Cloud AI Research for their valuable feedback. We also thank the anonymous ICLR reviewers and area chairs whose feedback has been instrumental in improving this work.

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

## A    Derivation of the Approximated Importance Score

In this section, we give detailed derivation of the importance score used in Sec. 2 to rank the examples. Recalling that we are given a pool of examples $\mathcal{E}$ with $|\mathcal{E}| = m$, a collection of $T$ subsets of $\mathbf{e}_i$, each represented as a binary vector $\mathbf{e}_i \in \{0,1\}^m$ and their corresponding scores on the validation set $g(\cdot) : \{0,1\}^m \rightarrow \mathbb{R}$, we first fit a GP regression with $\mathbf{e}_{1:T} = [\mathbf{e}_1, ..., \mathbf{e}_T]^\top$ and $\mathbf{g}_{1:T} = [g(\mathbf{e_1}, ..., g(\mathbf{e}_T)]^\top$, as presented in Eq. 2, the mean of the posterior GP $\hat{g}(\cdot)$ is given by:

$$\mathbb{E}_{\hat{g}(\mathbf{e})|\mathcal{G}_T}[\hat{g}(\mathbf{e})] = \mathbf{k}_{1:T}(\mathbf{K} + \eta^2 \mathbf{I})^{-1}\mathbf{g}_{1:T}, \tag{3}$$

where we define $\mathcal{G}_T$ as the shorthand of $[\mathbf{e}_{1:T}, \mathbf{g}_{1:T}]$ to denote that the fitted function $\hat{g}(\mathbf{e})$ is fitted on the observed input-output pairs; $\mathbf{k}_t = [k(\mathbf{e}, \mathbf{e}_1), ..., k(\mathbf{e}, \mathbf{e}_t)]$ and $k(\cdot, \cdot)$ is the covariance function of the GP (we use Matern 2.5 by default). As mentioned in Sec. 2, whereas we do not assume any differentiability property from $g(\cdot)$ on $\mathbf{e}$, since the approximated function $\hat{g}(\cdot)$ follows a posterior GP, its gradient w.r.t $\mathbf{e}$ is analytically available and is itself a GP, given by:

$$\nabla_{\mathbf{e}} g = \frac{\partial g(\mathbf{e})}{\partial \mathbf{e}} = \frac{\partial \mathbf{k}_{1:T}}{\partial \mathbf{e}}(\mathbf{K} + \eta^2 \mathbf{I})^{-1}\mathbf{g}_{1:T}, \tag{4}$$

noting that the expensive matrix inversion term, $(\mathbf{K} + \eta^2 \mathbf{I})^{-1}$ does not have a dependence on $\mathbf{e}$ and can be directly cached from Eq. 3 when we compute the posterior mean. The derivative term is essentially a differentiation operation of the covariance function to the input, and can be easily computed either analytically for common kernel choices or via automatic differentiation for popular GP or BO packages like `gpytorch` (Gardner et al., 2018) or `botorch` (Balandat et al., 2020).

With the computed $\nabla_{\mathbf{e}} g \in \mathbb{R}^m$, we can in principle compute the estimated derivative at any $\mathbf{e} \subseteq \mathcal{E}$. However, in practice, we find the derivative estimate to be more reliable at the *training points* of the GP (i.e., $[\mathbf{e}_1, ..., \mathbf{e}_T]$). We then evaluate the derivative at each of the training point, and the final importance score is marginalized by averaging across the training points:

$$M(e^{(j)}) = \frac{1}{T}\sum_{t=1}^{T}\nabla_{\mathbf{e}}\hat{g}|_{\mathbf{e}=\mathbf{e}_t}^{(j)}, \tag{5}$$

where we use the superscript $(j)$ to denote that the estimated importance of the $j$-th individual example (note the regular font $e \in \mathcal{E}$ denoting an *individual* example instead of the bold-face $\mathbf{e}$ denoting a *set of examples* in $\mathcal{E}$). We then compute the importance score of all examples in $\mathcal{E}$, which is then used to generate the assigned ranking in the analysis of Sec. 2 such as the Fig. 4.

## B    Implementation Details

### B.1    Datasets.

In the section below, we give detailed implementation details for the availability, data splitting protocol, input prompts and licensing information of the datasets used.

**BIG-Bench Hard (BBH).** BBH is a collection of 26 challenging reasoning tasks, and a task is selected if either 1) if it is studied in the seminal work on many-shot ICL (Agarwal et al., 2024) or 2) if the zero-shot performance of `gemini-1.5-pro-001` is below 90%, which indicates non-saturation of performance – these criteria led to a set of 16 tasks that we consider in Sec. 4. For all tasks, we randomize the data points and reserve 40% (usually 100 samples, but some sub-tasks of BBH benchmark have fewer data-points) as held-out sets for testing, whose inputs and labels are not revealed to the model except for final evaluation. For the rest of the dataset, in Sec. 2, we use 50% (30% of all available data points including the held-out test set) as the "train-set" from which the examples are generated and the other 50% for validation (i.e., the split where results in Fig. 4 is generated). In Sec. 4, we do not use the aforementioned validation set and use performance on the same set that generates the examples as the optimization objective. The BBH dataset is publicly available at `https://github.com/suzgunmirac/BIG-Bench-Hard` under an MIT license. For all BBH tasks, we use the prompt templates below:

```
1  You will be given a question. Think step by step before giving a final answer to this question
       . Show your final answer {{ TASK_SPECIFIC_CONSTRAINTS }} between <answer> and <\answer>
2
```

```
3 {{ EXAMPLES }}
4 ==
5
6 {{ QUESTION }}
7 {{ llm() }}
```

where we use a Jinja2-style syntax and the upper-cased blocks bracketed between double braces are *variables* that are replaced at inference time: TASK_SPECIFIC_CONSTRAINTS denote the constraint instruction specific to the type of the task. For example, for a multiple-choice task, this is replaced with "answer option letter only"; for a binary choice question, this is replaced with "Yes or No only" and for a free-form generation task, this is replaced by an empty string. EXAMPLES denote the concatenation of any examples e added to the input – for the initial generation step (i.e., Step 3 in Algorithm 1), we use zero-shot prompting and EXAMPLES is an empty string. For subsequent generation step, this is replaced with the concatenation of the examples selected by BRIDGE; finally, llm() denotes the place where an LLM response is solicited; the answer is then extracted and postprocessed to match with a ground-truth answer to measure accuracy.

**MATH and GSM-Hard.** In MATH and GSM-Hard, we similarly adopt the *unified* train set setup as mentioned in the previous paragraph. To minimize the chance of data contamination where the training inputs and outputs were leaked to the model during pre-training or instruction finetuning, we randomly sample 128 samples as the official test set as the train set from which the examples are generated and use the rest of the official test set for testing. The MATH dataset is available at `https://github.com/hendrycks/math` and GSM-Hard is available at `https://huggingface.co/datasets/reasoning-machines/gsm-hard`. Both datasets are license under an MIT license. On GSM-Hard, we use the same prompt as the BBH dataset mentioned above. On MATH, we use the *inner monologue* prompt consists of human-annotated few-shot prompt given by Agarwal et al. (2024) with an added preamble to ensure that the LLM generation follows the style of the examples given, noting that any model-generated examples will be added to the initial human-annotated examples:

```
1  You will be given several example math questions and their solutions. At the end, You will be
       given another question that you will need to solve. Make sure you follow the examples,
       and always finish your answer with 'Final Answer: The final answer is X. I hope it is
       correct.' where X is the correct answer.
2
3  Problem:
4  Find the domain of the expression $\frac{\sqrt{x-2}}{\sqrt{5-x}}$.
5
6  Solution:
7
8  I need to find the domain of this expression.
9
10 This expression has two square root expressions.
11
12 I know the values under each square root must be non-negative.
13
14 So let me start by ensuring that.
15
16 I set $x-2 \ge 0$, which simplifies to $x\ge2$.
17
18 Then, I set $5 - x \ge 0$, which simplifies to $x \le 5$.
19
20 But I need to make sure the denominator is not zero.
21
22 So, $5-x$ must also be greater than zero, which gives me $x<5$.
23
24 Therefore, the domain of the expression is $\boxed{[2,5)}$.
25
26 Answer: $[2,5)$.
27
28 Final Answer: The final answer is $[2,5)$. I hope it is correct.
29 ==
30
31 Problem:
32 If $\det \mathbf{A} = 2$ and $\det \mathbf{B} = 12,$ then find $\det (\mathbf{A} \mathbf{B}).$
33
34 Solution:
35
36 I need to find the determinant of the product of matrices A and B.
37
38 I remember that the determinant of the product of two matrices equals the product of their
       determinants.
39
```

```
40  So, $\det (\mathbf{A} \mathbf{B}) = (\det \mathbf{A})(\det \mathbf{B}) = (2)(12) = \boxed{24}$
        .
41
42  Answer: $24$.
43
44  Final Answer: The final answer is $24$. I hope it is correct."""",
45      r"""Problem:
46  Terrell usually lifts two 20-pound weights 12 times. If he uses two 15-pound weights instead,
        how many times must Terrell lift them in order to lift the same total weight?
47
48  Solution:
49
50
51  Okay, so Terrell lifts a total of $2\cdot 12\cdot20=480$ pounds of weight with the 20-pound
        weights.
52
53    Well, if he switches to 15-pound weights, the total weight lifted will be $2\cdot15\cdot n
        =30n$ pounds, where n is the number of lifts.
54
55  I want to find the number of lifts, n, for the total weight lifted to be the same.
56
57  I equate $30n$ to 480 pounds and solve for n.
58
59  \begin{align*}
60  30n&=480\\
61  \Rightarrow\qquad n&=480/30=\boxed{16}
62  \end{align*}
63
64  Answer: $16$.
65
66  Final Answer: The final answer is $16$. I hope it is correct.
67  ==
68
69  Problem:
70  If the system of equations
71
72  \begin{align*}
73  6x-4y&=a,\\
74  6y-9x &=b.
75  \end{align*}
76
77  has a solution $(x, y)$ where $x$ and $y$ are both nonzero, find $\frac{a}{b},$ assuming $b$
        is nonzero.
78
79  Solution:
80
81  I'm given a system of two equations.
82
83  I see that if I multiply the first equation by $-\frac{3}{2}$, I'll get another equation that
        has the same left-hand side as the second equation, $6y-9x$.
84
85  Let me try that $$6y-9x=-\frac{3}{2}a.$$
86
87  Ah, I also know that $6y-9x=b$, so I can equate these two equations.
88
89  So, $$-\frac{3}{2}a=b\Rightarrow\frac{a}{b}=\boxed{-\frac{2}{3}}.$$
90
91  Answer: $-\frac{2}{3}$.
92
93  Final Answer: The final answer is $-\frac{2}{3}$. I hope it is correct.
94  ==
95
96  {{ EXAMPLES }}
97
98  ==
99  Problem:
100 {{ QUESTION }}
101
102 Solution:
103
104 {{ llm() }}
```

**BIRD** On BIRD, we randomly sample 128 samples from the train split as the unified train and validation set and use the official test set (of 1534 data points) for testing. Since BIRD is a code generation task, the execution accuracy is computed not via a simple string match between the predicted and the ground-truth SQLs but by actually executing both SQLs on the database provided, and a score of 1 is only assigned when the predicted SQL is both executable and if whose results exactly match the execution results from the ground-truth SQL. All data, including the databases,

schemas and ground-truth gold SQL are available at the official repo: `https://bird-bench.github.io` under a `CC BY-SA 4.0` licence. With reference to Table 4, use two prompt versions for different rows. The *direct* prompt is a standard, zero-shot prompt to elicit the SQL prediction directly; it is used both for the "Direct" row to directly extract LLM answer and is also used as the prompt template for finetuning in the different "LoRA" rows:

```
1 You are a SQL expert tasked with answering user's questions about SQL tables by generating SQL
      queries in the SQLITE dialect.
2
3 Use only the following tables to answer the question:
4
5 {{ SCHEMA }}
6
7 Question: {{ QUESTION }}
8 Hint: {{ HINT }}
9 SQL: {{ llm()  }}
```

where `SCHEMA` refers to the *table schema*, which can be generated automatically by querying the database, `QUESITON` is the natural language question that we would like the LLM to convert to a SQL command and `HINT` is a hint which additionally explains the question provided by the BIRD dataset. For the `CHASE` and `CHASE + BRIDGE` rows, we use the prompt template proposed in Pourreza et al. (2025) to invoke reasoning and divide-and-conquer before the LLM gives the final answer:

```
1 You are an experienced database expert.
2 Now you need to generate a SQL query given the database information, a question and some
      additional information.
3 The database structure is defined by the following table schemas (comments after '--' provide
      additional column descriptions).
4 Note that the "Example Values" are actual values from the column. Some column might contain
      the values that are directly related to the question. Use it to help you justify which
      columns to use.
5
6 Given the table schema information description and the 'Question'. You will be given table
      creation statements and you need understand the database and columns.
7
8 You will be using a way called "recursive divide-and-conquer approach to SQL query generation
      from natural language".
9
10 Here is a high level description of the steps.
11 1. **Divide (Decompose Sub-question with Pseudo SQL):** The complex natural language question
      is recursively broken down into simpler sub-questions. Each sub-question targets a
      specific piece of information or logic required for the final SQL query.
12 2. **Conquer (Real SQL for sub-questions):**  For each sub-question (and the main question
      initially), a "pseudo-SQL" fragment is formulated. This pseudo-SQL represents the
      intended SQL logic but might have placeholders for answers to the decomposed sub-
      questions.
13 3. **Combine (Reassemble):** Once all sub-questions are resolved and their corresponding SQL
      fragments are generated, the process reverses. The SQL fragments are recursively combined
       by replacing the placeholders in the pseudo-SQL with the actual generated SQL from the
      lower levels.
14 4. **Final Output:** This bottom-up assembly culminates in the complete and correct SQL query
      that answers the original complex question.
15
16 Database admin instructions (violating any of the following is punishable to death!):
17 1. **SELECT Clause:**
18    - Only select columns mentioned in the user's question.
19    - Avoid unnecessary columns or values.
20 2. **Aggregation (MAX/MIN):**
21    - Always perform JOINs before using MAX() or MIN().
22 3. **ORDER BY with Distinct Values:**
23    - Use 'GROUP BY <column>' before 'ORDER BY <column> ASC|DESC' to ensure distinct values.
24 4. **Handling NULLs:**
25    - If a column may contain NULL values (indicated by "None" in value examples or explicitly
      ), use 'JOIN' or 'WHERE <column> IS NOT NULL'.
26 5. **FROM/JOIN Clauses:**
27    - Only include tables essential to answer the question.
28 6. **Strictly Follow Hints:**
29    - Adhere to all provided hints.
30 7. **Thorough Question Analysis:**
31    - Address all conditions mentioned in the question.
32 8. **DISTINCT Keyword:**
33    - Use 'SELECT DISTINCT' when the question requires unique values (e.g., IDs, URLs).
34    - Refer to column statistics ("Value Statics") to determine if 'DISTINCT' is necessary.
35 9. **Column Selection:**
36    - Carefully analyze column descriptions and hints to choose the correct column when
      similar columns exist across tables.
37 10. **String Concatenation:**
38    - Never use '|| ' ' ||' or any other method to concatenate strings in the 'SELECT' clause.
```

```
39  11. **JOIN Preference:**
40      - Prioritize `INNER JOIN` over nested `SELECT` statements.
41  12. **SQLite Functions Only:**
42      - Use only functions available in SQLite.
43  13. **Date Processing:**
44      - Utilize `STRFTIME()` for date manipulation (e.g., `STRFTIME('%Y', SOMETIME)` to extract
        the year).
45
46  When you get to the final query, output the query string ONLY inside the xml delimiter <
        FINAL_ANSWER></FINAL_ANSWER>.
47
48  Here are some examples:
49
50  {{ EXAMPLES }}
51
52  Now is the real question, following the instruction and examples, generate the SQL with
        Recursive Divide-and-Conquer approach.
53  **************************
54  [Table creation statements]
55  {{ SCHEMA }}
56
57  **************************
58  [Question]
59  Question: {{ QUESTION }}
60  Hint: {{ HINT }}
61
62  **************************
63  [Answer]
64  Repeating the question and hint, and generating the SQL with Recursive Divide-and-Conquer.
65  {{ llm() }}
```

Similar to the inner monologue prompt in the MATH dataset, the EXAMPLES in the template above also include 4 human annotated examples at initialization (an exemplary human-annotated example is shown in the code snippet below); any model-generated examples from BRIDGE are again concatenated to the human annotated examples at inference.

```
1   **************************
2   [Table creation statements]
3   CREATE TABLE generalinfo
4   (
5     id_restaurant INTEGER not null primary key,
6     food_type TEXT null, -- examples: `thai`| `food type` description: the food type
7     city TEXT null, -- description: the city where the restaurant is located in
8   );
9
10  CREATE TABLE location
11  (
12    id_restaurant INTEGER not null primary key,
13    street_name TEXT null, -- examples: `ave`, `san pablo ave`, `pablo ave`| `street name`
        description: the street name of the restaurant
14    city TEXT null, -- description: the city where the restaurant is located in
15    foreign key (id_restaurant) references generalinfo (id_restaurant) on update cascade on
        delete cascade,
16  );
17
18  **************************
19  [Question]
20  Question: How many Thai restaurants can be found in San Pablo Ave, Albany?
21  Hint: Thai restaurant refers to food_type = 'thai'; San Pablo Ave Albany refers to street_name
        = 'san pablo ave' AND T1.city = 'albany'
22
23  **************************
24  [Answer]
25  Repeating the question and hint, and generating the SQL with Recursive Divide-and-Conquer.
26  **Question**: How many Thai restaurants can be found in San Pablo Ave, Albany?
27  **Hint**: Thai restaurant refers to food_type = 'thai'; San Pablo Ave Albany refers to
        street_name = 'san pablo ave' AND T1.city = 'albany'
28
29  **1. Divide and Conquer:**
30
31  * **Main Question:** How many Thai restaurants can be found in San Pablo Ave, Albany?
32      * **Analysis:** The question asks for a count of restaurants, so we'll use `COUNT()` for
        that. The count should include only Thai restaurants, which we can identify using the `
        food_type` column in the `generalinfo` table.  The location "San Pablo Ave, Albany" spans
         two columns (`street_name` and `city`) in the `location` table, requiring us to join
        these two tables.
33      * **Pseudo SQL:** SELECT COUNT(`T1`.`id_restaurant`) FROM `generalinfo` AS `T1` INNER JOIN
         `location` AS `T2` ON `T1`.`id_restaurant` = `T2`.`id_restaurant` WHERE  <Thai
        restaurant> AND <in San Pablo Ave, Albany>
34
```

```
35      * **Sub-question 1:** Thai restaurant
36          * **Analysis:** This is a straightforward filter on the `generalinfo` table using the
        `food_type` column.
37          * **Pseudo SQL:** `T1`.`food_type` = 'thai'
38
39      * **Sub-question 2:** in San Pablo Ave, Albany
40          * **Analysis:** This location information is spread across two columns in the `
        location` table. We need to combine these conditions with an "AND" operator to ensure
        both are met.
41          * **Pseudo SQL:** `T2`.`street_name` = 'san pablo ave' AND `T2`.`city` = 'albany'
42
43   **2. Assembling SQL:**
44
45   * **Sub-question 1 (Thai restaurant):**
46      * **SQL:** `T1`.`food_type` = 'thai'
47
48   * **Sub-question 2 (in San Pablo Ave, Albany):**
49      * **SQL:** `T2`.`street_name` = 'san pablo ave' AND `T2`.`city` = 'albany'
50
51   * **Main Question (count of restaurants):**
52      * **SQL:** SELECT COUNT(`T1`.`id_restaurant`) FROM `generalinfo` AS `T1` INNER JOIN `
        location` AS `T2` ON `T1`.`id_restaurant` = `T2`.`id_restaurant` WHERE `T1`.`food_type` =
        'thai' AND `T2`.`street_name` = 'san pablo ave' AND `T2`.`city` = 'albany'
53
54   **3. Simplification and Optimization:**
55
56   * The SQL query from step 2 is already quite efficient. We've used `INNER JOIN` to combine the
        tables based on their relationship, and the `WHERE` clause clearly defines our filtering
        criteria. There's no need for nested queries or complex sub-selections in this case.
57
58   **Final Optimized SQL Query:**
59
60   <FINAL_ANSWER>
61   SELECT COUNT(T1.id_restaurant) FROM generalinfo AS T1 INNER JOIN location AS T2 ON T1.
        id_restaurant = T2.id_restaurant WHERE T1.food_type = 'thai' AND T1.city = 'albany' AND
        T2.street_name = 'san pablo ave'
62   </FINAL_ANSWER>
```

### B.2 IMPLEMENTATION DETAILS OF THE INFILLING BASELINE

*Infilling* is a technique of generating the intermediate outputs given both input queries *and* the ground-truth answer – this is used as a baseline in Tables 1 and 3 where we utilize all available labeled data in the context. Concretely, we use the following prompt adapted from Hu et al. (2023) to generate the intermediate rationales.

```
1   You will be given a question and its final, ground-truth correct answer.
2   Given the question and the answer, generate the step-by-step reasoning steps that led to the
        correct answer. Write your intermediate reasoning steps (but NOT the final answer)
        leading to the final answer between <answer> and </answer>.
3
4   Question: {{ question }}
5   Answer: {{ target }}
6   Steps: {{ llm() }}
```

## C ADDITIONAL EXPERIMENTS AND RESULTS

### C.1 ABLATION AND SENSITIVITY STUDIES

**Importance of Bayesian optimization.** To ablate BRIDGE, in Table 6 and Table 5, we compare against a simplified variant of BRIDGE with BO replaced with random search consuming the same evaluation budget (32 per stage) – we find that while random search is a remarkably strong baseline, BO nevertheless outperformed it consistently at all stages of the BRIDGE pipeline.

**Comparison to and combination with heuristic demonstration selection.** An alternative to it-eratively optimize the demonstrations in the "Optimize" step is using heuristics for demonstration selection which may incur a lower computational cost as we no longer have to repeatedly evaluate on the labeled validation set $m$ times. In this section, we study two representative demonstration selection techniques: *retrieval based on similarity in the embedding space* and *diversity*, and we both study them as standalone alternative to the full BRIDGE pipeline and, given that demonstration selection is not the only component of the BRIDGE framework, it is also straightforward to combine

Table 5: Comparison between BRIDGE with BO (BRIDGE-BO) and BRIDGE with random search (BRIDGE-RS) using `gemini-1.5-flash-001` on BBH tasks. The BRIDGE-BO results are lifted from Table 3, and the last row denotes the average improvement due to the use of BO over RS at the milestone in the progression of BRIDGE. Refers to captions of Table 1 for additional explanations.

| Tasks | BRIDGE-RS | | | | | BRIDGE-BO | | | | |
| --- | --- | --- | --- | --- | --- | --- | --- | --- | --- | --- |
| # Iterations | 1o | 1G | 2o | 2G | 3o | 1o | 1G | 2o | 2G | 3o |
| causal_judgement | $59.3_{2.0}$ | $\underline{66.7_{1.6}}$ | $\mathbf{67.7_{1.5}}$ | $63.0_{1.1}$ | $64.0_{1.6}$ | $61.3_{2.7}$ | $66.0_{2.0}$ | $63.3_{1.5}$ | $65.0_{1.6}$ | $65.3_{1.5}$ |
| date_understanding | $84.8_{1.3}$ | $90.5_{0.5}$ | $\underline{93.3_{0.4}}$ | $93.0_{0.7}$ | $\mathbf{94.5_{0.8}}$ | $85.0_{1.3}$ | $90.5_{0.5}$ | $91.5_{0.4}$ | $90.8_{0.7}$ | $92.5_{0.8}$ |
| disambiguation_qa | $73.8_{1.3}$ | $74.5_{1.1}$ | $74.0_{1.2}$ | $75.3_{0.8}$ | $70.5_{1.1}$ | $77.5_{1.3}$ | $\mathbf{79.0_{1.1}}$ | $\underline{77.5_{1.2}}$ | $76.3_{0.8}$ | $74.3_{1.1}$ |
| dyck_languages | $64.5_{1.5}$ | $62.5_{3.6}$ | $\mathbf{65.5_{4.2}}$ | $64.8_{1.1}$ | $68.0_{2.5}$ | $63.3_{2.0}$ | $62.0_{1.7}$ | $\underline{64.5_{1.8}}$ | $62.8_{2.4}$ | $61.8_{3.8}$ |
| formal_fallacies | $77.3_{1.1}$ | $75.0_{2.6}$ | $74.5_{1.7}$ | $77.5_{1.7}$ | $78.3_{2.5}$ | $\mathbf{78.3_{1.3}}$ | $77.3_{1.5}$ | $75.5_{1.7}$ | $\underline{78.3_{1.8}}$ | $76.3_{0.8}$ |
| geometric_shapes | $88.5_{3.8}$ | $93.3_{3.0}$ | $94.5_{2.1}$ | $\mathbf{98.0_{0.0}}$ | $95.3_{1.9}$ | $93.8_{2.5}$ | $94.0_{4.2}$ | $95.5_{1.1}$ | $97.0_{0.0}$ | $\mathbf{98.0_{0.0}}$ |
| hyperbaton | $94.0_{0.7}$ | $94.3_{0.4}$ | $95.0_{0.7}$ | $95.0_{0.7}$ | $88.8_{1.5}$ | $86.5_{7.6}$ | $\underline{95.5_{1.1}}$ | $\mathbf{95.8_{0.8}}$ | $94.8_{0.4}$ | $93.3_{1.5}$ |
| logical_deduction (7) | $62.8_{3.3}$ | $54.5_{2.2}$ | $67.8_{1.9}$ | $64.0_{2.6}$ | $66.8_{1.9}$ | $61.8_{5.1}$ | $57.5_{1.1}$ | $\underline{70.5_{0.9}}$ | $66.5_{1.1}$ | $\mathbf{75.0_{0.7}}$ |
| movie_recommendation | $68.5_{4.0}$ | $75.3_{2.6}$ | $72.5_{1.7}$ | $\underline{77.5_{1.3}}$ | $77.5_{1.8}$ | $70.3_{2.3}$ | $73.3_{2.3}$ | $77.3_{1.5}$ | $\mathbf{78.8_{2.0}}$ | $72.8_{3.2}$ |
| multistep_arithmetic_two | $82.5_{0.5}$ | $92.3_{1.3}$ | $95.0_{1.4}$ | $89.5_{1.5}$ | $92.5_{2.6}$ | $96.3_{2.3}$ | $\underline{96.8_{0.4}}$ | $\mathbf{97.8_{0.4}}$ | $94.8_{0.8}$ | $95.8_{0.4}$ |
| object_counting | $92.0_{1.2}$ | $92.5_{1.5}$ | $92.5_{1.1}$ | $93.0_{0.7}$ | $92.3_{1.1}$ | $92.8_{1.9}$ | $94.0_{2.3}$ | $\mathbf{95.5_{0.5}}$ | $93.0_{1.2}$ | $\underline{93.8_{0.4}}$ |
| ruin_names | $89.0_{1.2}$ | $88.0_{0.7}$ | $88.0_{2.4}$ | $87.0_{1.2}$ | $84.5_{1.1}$ | $89.3_{0.4}$ | $89.3_{0.8}$ | $87.0_{1.2}$ | $\mathbf{90.3_{0.8}}$ | $\underline{90.0_{1.2}}$ |
| salient_translation_error_detection | $66.3_{2.8}$ | $69.3_{2.5}$ | $67.0_{2.6}$ | $68.5_{1.8}$ | $68.8_{2.1}$ | $62.8_{0.8}$ | $\mathbf{71.0_{0.7}}$ | $\underline{69.8_{2.0}}$ | $69.0_{0.7}$ | $67.3_{0.4}$ |
| snarks | $87.2_{3.0}$ | $90.6_{1.2}$ | $88.9_{1.7}$ | $\mathbf{93.4_{1.5}}$ | $\underline{91.0_{1.6}}$ | $88.9_{2.0}$ | $89.9_{1.8}$ | $89.6_{0.7}$ | $90.6_{0.6}$ | $83.7_{3.5}$ |
| sports_understanding | $96.5_{1.1}$ | $96.3_{0.4}$ | $\mathbf{97.3_{0.4}}$ | $95.8_{0.4}$ | $\underline{96.8_{0.8}}$ | $93.3_{1.1}$ | $95.3_{0.4}$ | $91.8_{0.4}$ | $95.0_{1.2}$ | $95.0_{0.0}$ |
| tracking_shuffled_objects (7) | $98.3_{0.8}$ | $89.5_{0.9}$ | $96.5_{1.1}$ | $92.3_{2.4}$ | $\mathbf{98.5_{1.5}}$ | $98.0_{0.7}$ | $93.8_{2.2}$ | $\underline{98.0_{0.0}}$ | $97.8_{0.4}$ | $97.5_{0.5}$ |
| *Average* | 80.31 | 81.55 | 83.11 | 82.98 | 82.97 | 81.61 | 82.79 | **83.79** | 83.77 | 83.25 |
| $\Delta$(BO - RS) | - | - | - | - | - | +1.30 | +1.24 | +0.68 | +0.79 | +0.28 |

Table 6: Comparison between BRIDGE with BO (BRIDGE-BO) and BRIDGE with random search (BRIDGE-RS) using `gemini-1.5-pro-001` on BBH tasks. The BRIDGE-BO results are lifted from Table 1, and the last row denotes the average improvement due to the use of BO over RS at the milestone in the progression of BRIDGE. Refers to captions of Table 1 for additional explanations.

| Tasks | BRIDGE-RS | | | | | BRIDGE-BO | | | | |
| --- | --- | --- | --- | --- | --- | --- | --- | --- | --- | --- |
| # Iterations | 1o | 1G | 2o | 2G | 3o | 1o | 1G | 2o | 2G | 3o |
| causal_judgement | $66.2_{3.0}$ | $68.5_{2.0}$ | $70.2_{2.4}$ | $69.5_{2.4}$ | $\underline{70.8_{2.2}}$ | $68.3_{1.5}$ | $62.7_{1.6}$ | $59.7_{1.5}$ | $\mathbf{72.0_{0.0}}$ | $70.0_{2.0}$ |
| date_understanding | $88.4_{2.3}$ | $94.3_{1.0}$ | $94.1_{1.2}$ | $90.3_{3.3}$ | $94.3_{1.3}$ | $92.2_{1.5}$ | $\mathbf{97.0_{0.7}}$ | $94.8_{1.9}$ | $95.0_{1.2}$ | $\underline{95.5_{1.8}}$ |
| disambiguation_qa | $75.5_{2.1}$ | $79.0_{2.9}$ | $77.4_{1.2}$ | $\underline{80.6_{2.3}}$ | $78.4_{4.0}$ | $71.8_{2.4}$ | $77.5_{3.6}$ | $80.5_{1.8}$ | $\mathbf{81.3_{2.9}}$ | $78.8_{1.5}$ |
| dyck_languages | $56.9_{5.4}$ | $59.6_{4.9}$ | $67.5_{4.3}$ | $64.9_{4.0}$ | $70.4_{2.7}$ | $49.2_{2.7}$ | $76.2_{3.8}$ | $\mathbf{80.0_{2.7}}$ | $\underline{77.5_{1.1}}$ | $76.8_{3.8}$ |
| formal_fallacies | $87.4_{1.5}$ | $86.8_{2.3}$ | $\mathbf{90.8_{2.1}}$ | $88.5_{2.2}$ | $88.8_{2.2}$ | $86.0_{2.1}$ | $85.0_{2.5}$ | $\underline{90.8_{2.3}}$ | $90.8_{2.8}$ | $88.2_{2.3}$ |
| geometric_shapes | $77.8_{3.2}$ | $82.1_{4.0}$ | $81.8_{2.5}$ | $86.5_{3.8}$ | $85.5_{2.4}$ | $78.5_{2.1}$ | $82.5_{3.6}$ | $89.2_{3.8}$ | $\mathbf{92.3_{1.1}}$ | $\underline{89.2_{0.8}}$ |
| hyperbaton | $94.3_{1.6}$ | $93.1_{2.4}$ | $94.2_{1.3}$ | $94.9_{1.5}$ | $94.0_{1.2}$ | $96.5_{0.9}$ | $94.2_{1.5}$ | $94.8_{2.8}$ | $\underline{96.5_{0.5}}$ | $\mathbf{97.2_{0.4}}$ |
| logical_deduction (7) | $70.9_{3.3}$ | $68.3_{2.7}$ | $66.6_{2.5}$ | $\mathbf{71.9_{3.3}}$ | $68.9_{2.1}$ | $70.2_{1.5}$ | $70.8_{4.5}$ | $\underline{71.7_{3.7}}$ | $71.5_{1.8}$ | $69.2_{2.2}$ |
| movie_recommendation | $63.5_{3.2}$ | $67.4_{1.8}$ | $67.4_{2.1}$ | $64.6_{2.3}$ | $63.4_{2.9}$ | $67.0_{1.2}$ | $\underline{69.5_{0.5}}$ | $69.3_{3.1}$ | $\mathbf{72.8_{1.6}}$ | $67.0_{1.2}$ |
| multistep_arithmetic_two | $97.3_{1.1}$ | $97.5_{0.7}$ | $96.9_{0.8}$ | $96.1_{1.5}$ | $\underline{97.9_{0.3}}$ | $96.2_{0.8}$ | $94.5_{1.1}$ | $97.0_{0.7}$ | $\mathbf{98.0_{0.7}}$ | $96.8_{1.8}$ |
| object_counting | $95.3_{2.4}$ | $\mathbf{98.1_{1.1}}$ | $97.3_{1.7}$ | $97.3_{1.9}$ | $95.4_{2.3}$ | $96.2_{0.4}$ | $96.0_{1.9}$ | $94.5_{1.1}$ | $94.2_{0.4}$ | $95.0_{0.7}$ |
| ruin_names | $86.6_{1.7}$ | $86.5_{1.9}$ | $88.9_{1.8}$ | $89.9_{1.2}$ | $87.1_{1.7}$ | $\mathbf{90.8_{1.1}}$ | $88.8_{1.7}$ | $89.2_{1.5}$ | $88.8_{2.4}$ | $\underline{90.3_{0.8}}$ |
| salient_translation_error_detection | $71.1_{3.2}$ | $73.4_{1.6}$ | $73.9_{2.2}$ | $71.9_{1.5}$ | $70.8_{1.6}$ | $68.8_{0.8}$ | $71.0_{0.7}$ | $69.5_{2.2}$ | $\underline{74.0_{0.7}}$ | $\mathbf{74.5_{1.1}}$ |
| snarks | $93.8_{1.6}$ | $95.3_{1.4}$ | $96.0_{1.6}$ | $96.0_{1.1}$ | $95.6_{1.8}$ | $93.4_{3.0}$ | $95.8_{0.0}$ | $95.1_{1.6}$ | $\underline{96.9_{1.5}}$ | $\mathbf{97.6_{1.8}}$ |
| sports_understanding | $93.5_{1.7}$ | $94.1_{0.6}$ | $95.1_{0.9}$ | $95.9_{0.9}$ | $96.0_{1.7}$ | $92.8_{1.9}$ | $\mathbf{97.0_{1.2}}$ | $96.2_{0.8}$ | $95.8_{0.4}$ | $95.8_{0.8}$ |
| tracking_shuffled_objects (7) | $92.4_{3.8}$ | $94.4_{1.2}$ | $99.0_{0.3}$ | $98.4_{0.9}$ | $\mathbf{100.0_{0.0}}$ | $95.8_{0.4}$ | $95.8_{0.0}$ | $\mathbf{100.0_{0.0}}$ | $97.0_{0.7}$ | $99.5_{0.5}$ |
| *Average* | 81.86 | 83.64 | 84.86 | 84.81 | 84.82 | 82.11 | 84.61 | 85.77 | **87.13** | 86.33 |
| $\Delta$(BO - RS) | - | - | - | - | - | +0.25 | +0.97 | +0.91 | +2.32 | +1.51 |

them with BRIDGE by swapping the BO/random search component in the "Optimize" step with these heuristics. Below we describe the implementation details of both techniques:

- **Retrieval**: One popular demonstration selection method is via *retrieval* (Rubin et al., 2022; Das et al., 2021). Concretely, we may either use an off-the-shelf pretrained embedding model (we use the latest Gecko embedding (Lee et al., 2024) for this purpose) or tune a customized retriever to obtain the *nearest* examples from an example store, typically by computing the vector embedding for each of the test queries and each of the cached demonstrations followed by a maximum inner product search (MIPS) to retrieve the top-$k$ demonstrations based on cosine similarity. Unlike the optimization-based approach where the number of examples in the context can be determined automatically, $k$ here is a key hyperparameter that needs to be set by the user. In this case, consider 3 different $k$ values: $k = \{10, 25\}$ where the number of examples is fixed, or $k =$ All, where we use all available, correctly predicted examples – this essentially uses the same set of examples as Reinforced ICL but in a specific, input-dependent order: the examples are sorted in an ascending order based on the cosine similarity between the embedding of the test input and

the example store and the most similar examples appears as the final demonstration that is directly concatenated to the test input.

- **Diversity**: Another popular learning-free demonstration selection method is by selecting diverse examples. While multiple ways to measure diversity exist, here we use the technique similar to the one used in Zhang et al. (2023) by 1) computing the embedding of all the available demonstrations and 2) run the $k$-means clustering algorithm and select the $k$ examples whose vector embeddings are nearest to each of the $k$ centroids. Unlike *retrieval*, there is no input dependency as the clustering algorithm does not depend on the input query but similar to *retrieval*, $k$ here is also a hyperparameter to be set and we again use $k = \{10, 25\}$. Note that we omit $k = $ All, as otherwise the number of clusters would be equal to the number of examples and we would be essentially be running Reinforced ICL with all available examples as demonstrations.

Since these demonstration selection baselines purely perform *selection* (i.e., the "optimize" step of BRIDGE) but neither the subsequent generations nor the iterative process, we first compare the BO demonstration selection (i.e., BRIDGE at Step 10) against these baselines and we show the results in Table 8. Overall, we find that "Diversity" and "Retrieval", regardless of their hyperparameters, perform on par or slightly worse than Reinforced ICL. While the hyperparameter choice can sometimes lead to significant differences on a per-task level, we also observe that when aggregated across the tasks, it does not lead to significant differences. On the other hand, the BO selection in BRIDGE outperform all these baselines. We believe there are two possible explanations leading to this out-performance. Firstly, while the heuristic-based methods have lower computational cost, key hyperparameters, such as the number of demonstrations to retrieve, need to be determined a-priori. However, as we have shown in the main text at, for example, Fig. 4, the optimal number of demonstrations can be highly task-specific, and while iterative optimization-based selection incurs a higher cost, it is also capable of optimizing the *number* of demonstrations. Secondly, a key finding we have in Sec. 2 is that not all examples are equally helpful and *removing* some examples as in-context demonstrations can sometimes lead to performance improvement during the "Optimize" stage. Again, while the heuristic-based approaches do not necessarily use *all* demonstrations, it makes the selection choice purely from heuristic metric (e.g., similarity to test query) rather than from a validation metric, and hence is incapable of removing these potentially "harmful" demonstrations from the pool of candidate examples.

However, beyond a simple comparison between a single stage of BRIDGE against these methods, it is also worth noting that BRIDGE *is more than a demonstration selection* method. As such, it is also possible to *combine* these methods with BRIDGE by using them as a drop-in replacement of the BO-based demonstration selection, effectively changing the implementation of the "Optimize" step *only*. To test this, we test two other variants of BRIDGE, named BRIDGE-RETRIEVAL and BRIDGE-DIVERSITY, where we replace the "Optimize" step in each round with the heuristic-driven demonstration selection mentioned above and the aggregated results are shown in Table 9 whereas the task-specific breakdown of the best method in Table 10 – for conciseness, we only show the per-task breakdown for the best BRIDGE variant (BRIDGE-RETRIEVAL using all examples), which show that BRIDGE also works well with alternative demonstration selection method, although the advantage of optimization-based selection as shown in Table 8 carries over when we use the selection as a component in the overall BRIDGE pipeline.

**Additional comparisons against iterative reinforced ICL in a restricted setup.** To provide further evidence emphasizing the need for the "Optimize" step and to make sure that the additional gain of BRIDGE does not simply comes from the fact that BRIDGE may take advantage of more correctly predicted demonstrations in the validation set due to repeated sampling in the later iterations, we conduct a further experiment comparing BRIDGE and iterative reinforced ICL, but in a restricted setup with the support set **restricted to the subset of the train set where the model predicted correctly initially**, instead of the entire train set. In other words, in subsequent iterations of BRIDGE and iterative reinforced ICL, both methods are restricted to make use of the subset of train set initially predicted correctly only as examples; we term these approaches "iterative reinforced ICL (restricted)" and "BRIDGE (restricted)" respectively, and we show the results in Table 7. On a high level, we found the result provides further evidence on the importance of selection: Iterative Reinf ICL (restricted) without the "optimize" step actually did not meaningfully improve over standard Reinf ICL (average accuracy: 79.6%); BRIDGE (restricted), however, still meaningfully improves

Table 7: Comparison of BRIDGE and iterative reinforced ICL **in the restricted setup** where the methods may only use the *subset* of the train set that the model initially predicted correctly. Experiments performed on `gemini-1.5-pro-001`.

| Tasks | *Restricted* Iterative Reinf. | | *Restricted* BRIDGE (Ours) | | | | |
|---|---|---|---|---|---|---|---|
| # Iterations | 1 | 2 | 1O | 1G | 2O | 2G | 3O |
| causal_judgement | $\mathbf{69.7_{1.1}}$ | $65.0_{1.5}$ | $67.7_{2.7}$ | $65.0_{1.1}$ | $66.0_{1.5}$ | $67.0_{1.1}$ | $65.0_{2.0}$ |
| date_understanding | $92.5_{1.5}$ | $\mathbf{94.0_{1.0}}$ | $92.5_{1.7}$ | $93.0_{1.6}$ | $92.5_{1.1}$ | $93.5_{1.7}$ | $89.3_{1.5}$ |
| disambiguation_qa | $74.0_{0.7}$ | $75.8_{3.0}$ | $70.5_{2.7}$ | $72.3_{1.5}$ | $75.5_{3.2}$ | $71.8_{2.4}$ | $\mathbf{76.5_{3.8}}$ |
| dyck_languages | $59.0_{6.5}$ | $53.0_{2.9}$ | $55.0_{5.2}$ | $52.3_{5.4}$ | $56.5_{3.4}$ | $57.5_{1.7}$ | $\mathbf{60.3_{3.8}}$ |
| formal_fallacies | $86.8_{3.3}$ | $90.5_{2.2}$ | $85.3_{1.5}$ | $\mathbf{90.5_{1.1}}$ | $83.0_{3.2}$ | $83.5_{0.9}$ | $85.5_{2.3}$ |
| geometric_shapes | $75.3_{1.8}$ | $78.3_{3.1}$ | $75.0_{2.6}$ | $80.5_{4.5}$ | $81.3_{4.5}$ | $\mathbf{85.0_{2.6}}$ | $80.0_{2.6}$ |
| hyperbaton | $85.3_{4.0}$ | $84.5_{3.4}$ | $94.0_{1.6}$ | $95.8_{0.8}$ | $91.8_{1.3}$ | $93.0_{2.6}$ | $\mathbf{97.0_{0.7}}$ |
| logical_deduction (7) | $67.5_{1.8}$ | $69.0_{2.5}$ | $69.5_{2.7}$ | $\mathbf{71.5_{3.2}}$ | $66.8_{2.1}$ | $70.5_{2.3}$ | $70.8_{2.2}$ |
| movie_recommendation | $64.8_{2.7}$ | $63.3_{2.2}$ | $\mathbf{68.3_{3.3}}$ | $62.0_{4.1}$ | $63.0_{1.2}$ | $63.3_{1.3}$ | $61.8_{2.2}$ |
| multistep_arithmetic_two | $95.0_{1.2}$ | $95.5_{0.5}$ | $\mathbf{97.8_{0.8}}$ | $95.3_{0.8}$ | $95.5_{1.8}$ | $95.8_{1.3}$ | $95.8_{1.5}$ |
| object_counting | $94.5_{2.9}$ | $94.5_{2.1}$ | $96.0_{1.6}$ | $94.8_{0.8}$ | $94.3_{1.3}$ | $96.0_{0.7}$ | $\mathbf{96.3_{1.1}}$ |
| ruin_names | $87.3_{1.3}$ | $88.8_{0.8}$ | $\mathbf{91.5_{0.9}}$ | $89.5_{1.8}$ | $88.5_{2.1}$ | $90.0_{0.0}$ | $89.8_{1.8}$ |
| salient_translation | $68.0_{2.1}$ | $67.3_{1.5}$ | $70.0_{2.2}$ | $69.8_{3.3}$ | $72.8_{2.4}$ | $73.5_{2.7}$ | $\mathbf{75.8_{2.2}}$ |
| snarks | $93.8_{1.2}$ | $93.8_{2.1}$ | $93.8_{2.3}$ | $95.1_{1.2}$ | $95.1_{0.7}$ | $\mathbf{95.5_{0.6}}$ | $94.8_{2.1}$ |
| sports_understanding | $94.0_{1.4}$ | $95.0_{1.0}$ | $92.8_{1.1}$ | $95.5_{1.5}$ | $\mathbf{97.0_{0.0}}$ | $96.0_{0.7}$ | $94.8_{0.8}$ |
| tracking_shuffled_objects (7) | $66.8_{3.1}$ | $67.3_{1.9}$ | $99.0_{0.0}$ | $96.3_{1.5}$ | $98.5_{0.5}$ | $97.5_{1.5}$ | $\mathbf{100.0_{0.0}}$ |
| *Average* | 79.62 | 79.70 | 82.40 | 82.45 | 82.37 | *83.08* | **83.32** |

with the subsequent optimize and generate steps, although the gain is less than the original BRIDGE which utilizes more examples via the larger train set support.

Table 8: Comparison between BRIDGE with (one step of demonstration optimization only) against *Retrieval*, *Diversity* and Reinforced ICL baselines using `gemini-1.5-pro-001`. Note that the BRIDGE (1O) and Reinforced ICL results are taken from Table 1.

| Tasks | Diversity | | Retrieval | | | Reinf. | BRIDGE |
|---|---|---|---|---|---|---|---|
| Details / hyperparams | $k=10$ | $k=25$ | $k=10$ | $k=25$ | All | ICL | 1O |
| causal_judgement | $66.7_{1.6}$ | $66.3_{2.4}$ | $63.0_{1.5}$ | $67.7_{2.4}$ | $66.7_{2.5}$ | $66.3_{4.8}$ | $68.3_{1.5}$ |
| date_understanding | $93.2_{1.3}$ | $93.0_{2.7}$ | $87.0_{3.5}$ | $93.3_{1.5}$ | $93.0_{1.9}$ | $88.8_{2.5}$ | $92.2_{1.5}$ |
| disambiguation_qa | $72.2_{3.0}$ | $77.8_{0.8}$ | $76.5_{0.9}$ | $71.2_{0.8}$ | $77.5_{1.1}$ | $76.8_{2.4}$ | $71.8_{2.4}$ |
| dyck_languages | $54.0_{15.7}$ | $38.5_{2.6}$ | $39.5_{4.4}$ | $33.2_{3.1}$ | $47.8_{5.2}$ | $55.5_{3.6}$ | $49.2_{2.7}$ |
| formal_fallacies | $85.5_{1.5}$ | $85.0_{1.9}$ | $88.5_{0.5}$ | $88.2_{3.0}$ | $84.2_{1.9}$ | $86.2_{1.1}$ | $86.0_{2.1}$ |
| geometric_shapes | $71.2_{4.4}$ | $69.3_{1.6}$ | $69.8_{2.8}$ | $68.5_{4.2}$ | $79.2_{3.3}$ | $80.2_{2.8}$ | $78.5_{2.1}$ |
| hyperbaton | $95.0_{1.2}$ | $92.2_{2.5}$ | $96.5_{1.1}$ | $97.2_{1.3}$ | $95.2_{1.9}$ | $90.2_{1.1}$ | $96.5_{0.9}$ |
| logical_deduction (7) | $65.8_{3.0}$ | $67.5_{4.4}$ | $69.2_{4.4}$ | $66.3_{2.9}$ | $67.3_{2.4}$ | $65.8_{3.5}$ | $70.2_{1.5}$ |
| movie_recommendation | $67.3_{2.6}$ | $65.0_{2.5}$ | $68.5_{3.4}$ | $68.0_{1.4}$ | $67.3_{3.3}$ | $65.2_{1.6}$ | $67.0_{1.2}$ |
| multistep_arithmetic_two | $92.8_{1.3}$ | $96.2_{0.4}$ | $95.5_{0.9}$ | $94.8_{1.6}$ | $94.3_{1.9}$ | $96.5_{0.5}$ | $96.2_{0.8}$ |
| object_counting | $95.8_{1.1}$ | $95.2_{0.8}$ | $97.2_{2.4}$ | $95.2_{1.9}$ | $91.2_{2.2}$ | $95.5_{0.9}$ | $96.2_{0.4}$ |
| ruin_names | $87.8_{1.3}$ | $89.8_{1.3}$ | $87.8_{0.8}$ | $91.5_{2.1}$ | $90.5_{2.2}$ | $89.8_{1.9}$ | $90.8_{1.1}$ |
| salient_translation_error_detection | $68.5_{2.3}$ | $69.5_{2.1}$ | $68.2_{3.3}$ | $58.2_{2.8}$ | $61.0_{2.1}$ | $69.0_{1.6}$ | $68.8_{0.8}$ |
| snarks | $94.8_{2.3}$ | $96.2_{1.2}$ | $94.4_{1.7}$ | $97.6_{1.2}$ | $95.5_{1.2}$ | $92.7_{3.2}$ | $93.4_{3.0}$ |
| sports_understanding | $95.0_{1.2}$ | $95.8_{1.1}$ | $95.5_{0.9}$ | $95.8_{0.8}$ | $95.0_{1.9}$ | $93.0_{1.4}$ | $92.8_{1.9}$ |
| tracking_shuffled_objects (7) | $55.8_{4.5}$ | $56.8_{5.5}$ | $60.2_{4.3}$ | $67.8_{9.7}$ | $60.2_{2.4}$ | $62.3_{4.2}$ | $95.8_{0.4}$ |
| *Average* | 78.83 | 78.38 | 78.59 | 78.41 | 79.12 | 79.61 | 81.61 |

Table 9: Average test accuracy on BBH tasks using `gemini-1.5-pro-001` by combining BRIDGE with different variants of the heuristic demonstration selection methods. Bold text in this table shows the best algorithm variant at each round of BRIDGE.

| Method | 1O | 1G | 2O | 2G | 3O |
|---|---|---|---|---|---|
| BRIDGE-DIVERSITY ($k=10$) | 77.10 | 79.47 | 78.58 | 81.89 | 79.50 |
| BRIDGE-DIVERSITY ($k=25$) | 78.15 | 80.86 | 78.74 | 80.63 | 79.68 |
| BRIDGE-NEAREST ($k=10$) | 79.07 | 81.80 | 81.40 | 81.35 | 80.39 |
| BRIDGE-NEAREST ($k=25$) | 78.36 | 79.49 | 80.16 | 81.09 | 80.10 |
| BRIDGE-NEAREST (All) | **79.65** | **82.91** | **82.01** | **83.20** | **84.14** |

Table 10: Task-specific test accuracy on BBH tasks using `gemini-1.5-pro-001` with BRIDGE-NEAREST (All) (best method from Table 9).

| Task | 1o | 1G | 2o | 2G | 3o |
|---|---|---|---|---|---|
| causal_judgement | $73.0_{1.1}$ | $62.3_{1.5}$ | $64.7_{0.7}$ | $65.7_{2.2}$ | $63.3_{2.7}$ |
| date_understanding | $94.3_{1.3}$ | $92.0_{1.6}$ | $95.0_{1.4}$ | $92.2_{2.3}$ | $92.8_{0.4}$ |
| disambiguation_qa | $76.8_{0.4}$ | $75.8_{5.0}$ | $72.0_{1.0}$ | $82.0_{2.7}$ | $82.8_{0.8}$ |
| dyck_languages | $58.8_{2.3}$ | $75.0_{4.3}$ | $75.0_{3.3}$ | $78.5_{3.0}$ | $82.0_{1.2}$ |
| formal_fallacies | $84.2_{0.8}$ | $88.5_{1.7}$ | $90.5_{0.9}$ | $89.5_{1.8}$ | $90.0_{0.7}$ |
| geometric_shapes | $75.8_{2.5}$ | $86.2_{3.3}$ | $79.8_{0.8}$ | $84.0_{2.1}$ | $84.5_{1.1}$ |
| hyperbaton | $96.0_{0.7}$ | $93.8_{2.3}$ | $97.0_{0.0}$ | $92.5_{3.2}$ | $98.8_{0.4}$ |
| logical_deduction_seven_objects | $65.8_{3.7}$ | $73.8_{2.3}$ | $68.0_{3.7}$ | $70.0_{1.9}$ | $71.2_{1.8}$ |
| movie_recommendation | $67.0_{1.2}$ | $69.5_{1.7}$ | $63.2_{1.1}$ | $70.0_{2.5}$ | $73.8_{0.8}$ |
| multistep_arithmetic_two | $92.5_{0.5}$ | $97.0_{1.2}$ | $96.7_{0.8}$ | $97.5_{0.9}$ | $94.0_{0.0}$ |
| object_counting | $91.8_{1.5}$ | $95.0_{1.2}$ | $97.0_{0.7}$ | $96.5_{1.7}$ | $100.0_{0.0}$ |
| ruin_names | $88.8_{0.4}$ | $92.0_{0.7}$ | $88.5_{2.1}$ | $89.2_{0.8}$ | $88.2_{1.1}$ |
| salient_translation_error_detection | $63.2_{1.5}$ | $70.0_{1.6}$ | $70.2_{0.4}$ | $70.0_{1.2}$ | $70.5_{0.5}$ |
| snarks | $95.8_{1.7}$ | $94.8_{1.2}$ | $93.7_{1.2}$ | $96.5_{1.2}$ | $95.8_{0.0}$ |
| sports_understanding | $94.0_{0.7}$ | $96.5_{1.5}$ | $93.8_{0.4}$ | $95.5_{1.5}$ | $94.2_{0.4}$ |
| tracking_shuffled_objects_seven_objects | $56.8_{1.6}$ | $64.5_{0.9}$ | $67.0_{1.0}$ | $61.5_{4.4}$ | $64.2_{1.6}$ |
| *Average* | 79.65 | 82.91 | 82.01 | 83.20 | **84.14** |

## C.2 NUMBER OF EXAMPLES

We show the number of examples used for each experiment corresponding to Table 1 in Table 11.

Table 11: Number of examples for each experiment corresponding to Table 1 (`gemini-1.5-pro-001` on BBH tasks). Note that the "All" columns always use all 75 examples provided.

| Tasks | Reinf. ICL | Iter. Reinf. | | BRIDGE-BO | | | | |
|---|---|---|---|---|---|---|---|---|
| # Iterations | 1 | 2 | 3 | 1o | 1G | 2o | 2G | 3o |
| causal_judgement | 36 | 40 | 43 | 11 | 43 | 4 | 39 | 39 |
| date_understanding | 61 | 67 | 72 | 57 | 73 | 44 | 73 | 57 |
| disambiguation_qa | 42 | 66 | 69 | 28 | 61 | 60 | 68 | 65 |
| dyck_languages | 15 | 40 | 52 | 9 | 45 | 42 | 59 | 20 |
| formal_fallacies | 60 | 69 | 69 | 2 | 63 | 30 | 67 | 57 |
| geometric_shapes | 42 | 59 | 68 | 40 | 59 | 19 | 71 | 70 |
| hyperbaton | 70 | 75 | 75 | 4 | 75 | 69 | 75 | 59 |
| logical_deduction_seven_objects | 46 | 60 | 62 | 11 | 54 | 51 | 64 | 61 |
| movie_recommendation | 42 | 53 | 54 | 39 | 49 | 36 | 51 | 41 |
| multistep_arithmetic_two | 65 | 74 | 74 | 38 | 74 | 28 | 72 | 38 |
| object_counting | 65 | 75 | 75 | 60 | 75 | 48 | 75 | 14 |
| ruin_names | 58 | 70 | 71 | 51 | 70 | 69 | 69 | 21 |
| salient_translation_error_detection | 44 | 59 | 60 | 13 | 58 | 7 | 59 | 41 |
| snarks | 47 | 50 | 51 | 19 | 49 | 5 | 48 | 39 |
| sports_understanding | 64 | 75 | 75 | 52 | 75 | 74 | 74 | 68 |
| tracking_shuffled_objects_seven_objects | 58 | 60 | 53 | 2 | 75 | 1 | 75 | 22 |
| *Average* | 50.94 | 62.00 | 63.94 | 27.25 | 62.38 | 36.69 | 64.94 | 44.50 |

## C.3 ADDITIONAL VISUALIZATIONS

In this section, we show analysis similar to Fig. 4 on tasks not represented in the figure of the main text.

## C.4 USING BRIDGE FOR LOW-RESOURCE TRANSLATION

While we have primarily considered the reinforced ICL setup suitable for reasoning and general problem-solving tasks, it is worth noting that the BRIDGE framework may also generalize to other practical settings that benefit from many-shot ICL with some modification on the "optimize" and the "generate" steps. In this section, we conduct a preliminary analysis on the applicability of BRIDGE in the context of machine translation (MT) for low-resource languages.

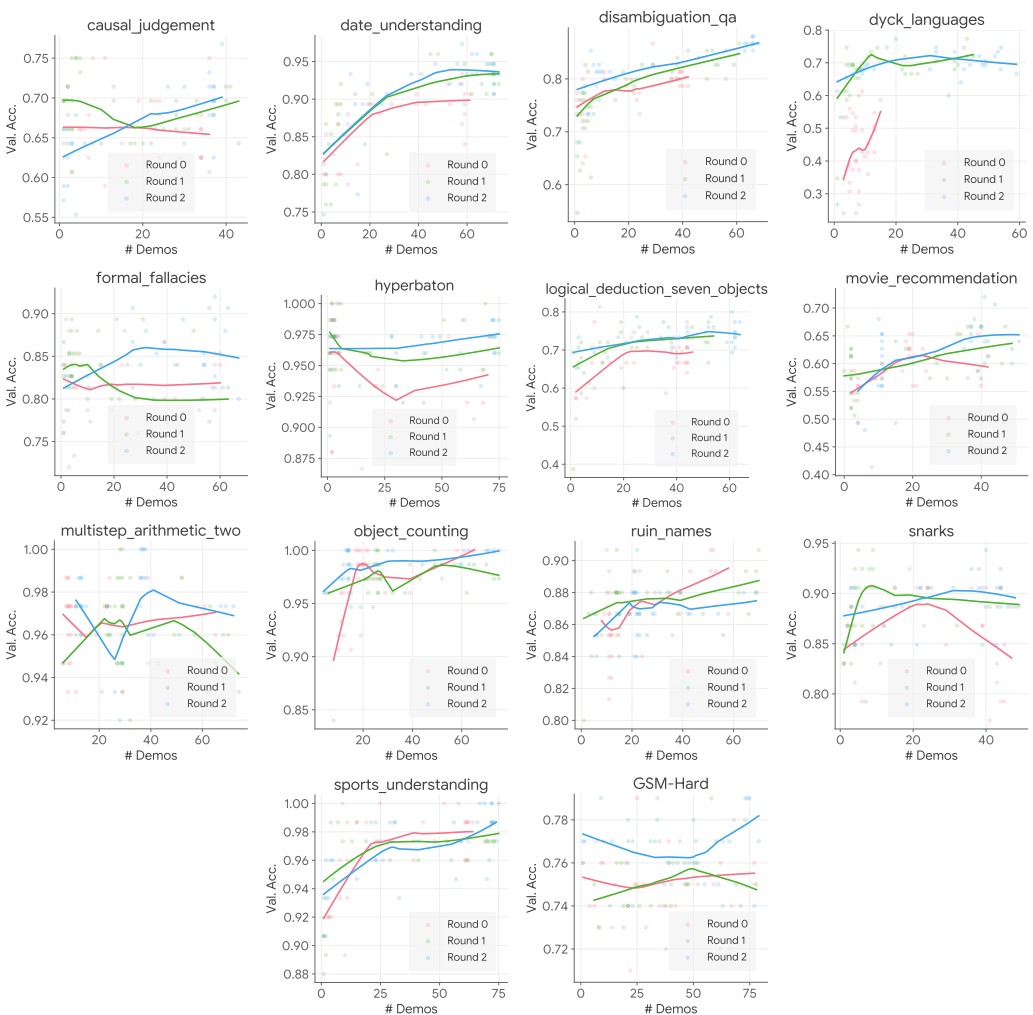

Figure 5: Additional visualization of the task performance at different rounds. Note that in most datasets, additional rounds of BRIDGE led to performance improvement, and some of the exceptions (e.g., multi_arithmetic_two) are possibly caused by visualization artefacts of the extremely small performance variation as shown by the small y-axis ranges.

As noted in Agarwal et al. (2024) and Reid et al. (2024), low-resource machine translation (MT) is one of the task types where many-shot in-context learning (ICL) has demonstrated remarkable performance. In these tasks, there is often a nearly monotonic improvement in translation quality as more source-target language pairs are incorporated into the context – as a notable exception to our observations in Sec. 2 that primarily involve reasoning tasks, in low resource MT, we often observe "more is better" given the information-dense nature of translation tasks – indeed, for translation tasks, barring glaring human errors in the annotation process, the provided data is generally assumed to be of high quality and problems like false positive in model-generated reasoning paths in reasoning tasks are generally negligible for tasks like low resource MT with high quality annotated data. However, in low-resource languages, the model's inherent knowledge is often weak or non-existent due to the lack of exposure to target languages during pre-training or fine-tuning, which can lead to a bottleneck in *data availability* especially for extremely low-resource languages, where **1)** the model lacks zero-shot translation abilities due to insufficient exposure to target languages, and **2)** the scarcity of annotated data becomes a critical limiting factor – to address these, previous works often attempt to augment ground-truth translation data with *model-synthesized* translations (Han et al., 2021; Patel et al., 2022).

In this section, along this line of work, we investigate the applicability of BRIDGE as a method to iteratively improve the *model-synthesized* translation so that they can act as more effective augmentations to the scarce ground-truth data. Specifically, we assume the following in our setup:

- Availability of some ground-truth source-target sentence pairs – this pair will both act as the *train set* from which ground-truth examples are generated and also as the validation set for machine-generated translations.

- Abundant *source* language text – this is almost always true. For example, if we are interested in translating from English to a low-resource language, it is extremely easy to obtain abundant text in English whereas the difficulty is to obtain the corresponding tranlsaiton in the target language.

- LLM for "pseudo-labelling" – we assume the availability of a (strong) LLM that can be queried to generate synthesized data.

---

**Algorithm 3** BRIDGE with pseudo-labelling.

1: **Input**: train set $\mathcal{D}$, unlabeled set with source language sentence, $\mathcal{U}$, number of iteration rounds $K \in \mathbb{N}$ (*outer-loop*), evaluation budget for BO per iteration $n_{\text{eval}}$ (*inner-loop*), *Generator model* used to synthesize examples $\mathcal{M}_g$.
2: **Output**: Optimized set of model-synthesized examples $\mathcal{E}^*$.
3: Partition $\mathcal{D}$ into two disjoint sets $\mathcal{D}_t$ and $\mathcal{D}_v$ via random sampling.
4: [Generate] Generate the pool of initial examples $\mathcal{E}_0$ by predicting $\mathcal{M}_g$ on the **unlabeled set**, using the entire **train** set $\mathcal{D}$ as the demonstrations in the context: $\mathcal{E}_0 \leftarrow \mathcal{M}_g(\mathcal{U}|\mathcal{D})$.
5: **for** $k \in \{1, ..., K\}$ (**Outer** loop) **do**
6:     [Optimize] Run Bayesian optimization (calling subroutine Algorithm 2 on the $\mathcal{D}_v$ to obtain $\mathbf{e}_k^* \leftarrow$ BayesOpt($n_{\text{eval}}=n_{\text{eval}}, \mathcal{E}=\mathcal{E}_k$).
7:     [Generate] **Re-generate** examples $\mathcal{E}_k$ by re-predicting the LLM on the **unlabeled** set, but with the optimized examples $\mathbf{e}_k^*$ from the previous step and $\mathcal{D}_t$ as demonstrations; the {inputs, output}-pairs are concatenated to form the new set of examples $\mathcal{E}_k$ for the next [Optimize] step.
8: **end for**
9: **return** Optimized example set $\mathcal{E}^*$ after $K$ rounds.

---

To approach the problem, we propose to retain the high-level framework of BRIDGE but modify the "optimize" and "generate" steps to accommodate the low-resource MT setup. With reference to Algorithm 3 where we have marked the key differences in blue, the main difference lies in the "generate" step: instead of generating examples with model-generated reasoning paths in the case presented in the main text, here we synthesized examples on the *unlabeled* set $\mathcal{U}$ that we assumed to be available. Since we no longer have access to the ground-truth translation of the sentences in $\mathcal{U}$, we optimize for the optimal subset $\mathbf{e}^*$ by evaluating different combinations of the synthesized examples on the partition of the labeled dataset $\mathcal{E}_v$.

To test BRIDGE on the MT setup, we consider the English-Bemba translation task in the Flores dataset (Guzmán et al., 2019) that was also considered in Agarwal et al. (2024). We assume the access to 100 labeled examples as $\mathcal{D}$ and 50 unlabeled examples $\mathcal{U}$, and hold out another 400 samples as the test set. We use Gemini Flash as the target model and Gemini Pro as the *generator* model in Algorithm 3, and we show the result in Table 12. Overall, we observe that running iterative optimization also improves performance on this task, both exemplified by improvement on the test and validation chrf score, although it seems that additional optimization round in this case led to a small performance degradation. While a more comprehensive evaluation is required, we believe the preliminary result is promising for future effort on this direction.

## C.5 EXPERIMENTS ON ADDITIONAL MODELS

In this section, we report BBH results on two additional models: Mistral Large (`mistral-large-2407`) (Table 13), Mistral Nemo (Table 14) (`mistral-nemo-instruct-2407`) (Jiang et al., 2023) and Claude 3.5 Sonnet (Anthropic, 2024) (Table 15). For both models, we use the versions served on Google Cloud Vertex AI platform. We find that while the base capabilities of the tested models differ slightly (e.g., Claude 3.5 Sonnet has a higher accuracy across the board), the high-level findings primarily derived from

Table 12: Test chrf score of `gemini-1.5-flash-001`. "Gold-only" refers to the result obtained by only using the 100 labeled examples in the context; "All" refers to the result with 100 labeled examples + 50 initially generated examples from `gemini-1.5-pro-001`. Refers to captions of Table 1 for additional explanations.

| Tasks | Gold-only | All | BRIDGE-MT | | | | |
|---|---|---|---|---|---|---|---|
| # Iterations | - | 0 | 1O | 1G | 2O | 2G | 3O |
| en_bem | 37.78 | 38.46 | 38.33 | 39.11 | **39.30** | 38.90 | 39.29 |

Table 13: Test accuracy of Mistral Large (`mistral-large-2407`) on BBH tasks. Refer to captions of Table 1 for detailed explanations.

| Tasks | Reinf. ICL | Iterative Reinf. | | BRIDGE (Ours) | | | | |
|---|---|---|---|---|---|---|---|---|
| # Iterations | 0 | 1 | 2 | 1O | 1G | 2O | 2G | 3O |
| causal_judgement | 69.3 | 66.7 | 72.0 | 68.0 | 65.3 | 69.3 | 64.0 | **73.3** |
| date_understanding | 92.0 | 92.0 | **96.0** | 93.0 | 94.0 | 95.0 | 92.0 | **96.0** |
| disambiguation_qa | 82.0 | 82.0 | 79.0 | 81.0 | **87.0** | **87.0** | 84.0 | 86.0 |
| dyck_language | 56.0 | 62.0 | 56.0 | 70.0 | 59.0 | 70.0 | 63.0 | **71.0** |
| formal_fallacies | **90.0** | 82.0 | 86.0 | 89.0 | 89.0 | **90.0** | 83.0 | 85.0 |
| geometric_shapes | 87.0 | 80.0 | 93.0 | 88.0 | 85.0 | **95.0** | 71.0 | 94.0 |
| hyperbaton | 99.0 | 96.0 | **100.0** | **100.0** | 98.0 | **100.0** | **100.0** | 99.0 |
| logical_deduction (7) | 81.0 | 85.0 | 76.0 | 82.0 | 88.0 | 90.0 | 86.0 | **92.0** |
| movie_recommendation | 74.0 | 71.0 | 74.0 | 77.0 | 66.0 | 78.0 | **80.0** | 79.0 |
| multistep_arithmetic_two | 88.0 | 92.0 | **93.0** | 91.0 | 89.0 | 88.0 | 86.0 | **93.0** |
| object_counting | 99.0 | 99.0 | 99.0 | 98.0 | 98.0 | 98.0 | **100.0** | 98.0 |
| ruin_names | 88.0 | 90.0 | **92.0** | 86.0 | **89.0** | 87.0 | 89.0 | **89.0** |
| salient_translation_error_detection | 66.0 | 68.0 | 70.0 | 78.0 | 69.0 | **75.0** | 72.0 | 73.0 |
| snarks | 95.8 | 95.8 | 97.2 | 94.4 | 95.8 | 95.8 | **95.8** | 93.1 |
| sports_understanding | 94.0 | 97.0 | **98.0** | 93.0 | 95.0 | 96.0 | 97.0 | 96.0 |
| tracking_shuffled_objects (7) | 96.0 | 68.0 | **100.0** | **100.0** | 73.0 | **100.0** | 57.0 | **100.0** |
| *Average* | 84.82 | 83.22 | 87.08 | 86.65 | 83.70 | 88.07 | 82.80 | **88.52** |

Table 14: Test accuracy of Mistral NeMo (`mistral-nemo`) on BBH tasks. Refer to captions of Table 1 for detailed explanations.

| Tasks | Reinf. ICL | Iterative Reinf. | | BRIDGE (Ours) | | | | |
|---|---|---|---|---|---|---|---|---|
| # Iterations | 0 | 1 | 2 | 1O | 1G | 2O | 2G | 3O |
| causal_judgement | 53.3 | **65.3** | 62.7 | 60.0 | 58.7 | 62.7 | 64.0 | 64.0 |
| date_understanding | 66.0 | 71.0 | 68.0 | 69.0 | 69.0 | **78.0** | 70.0 | 75.0 |
| disambiguation_qa | 58.0 | 60.0 | 64.0 | 63.0 | 60.0 | 61.0 | 66.0 | **72.0** |
| dyck_languages | 17.0 | 21.0 | 22.0 | 18.0 | 27.0 | 26.0 | 22.0 | **30.0** |
| formal_fallacies | **64.0** | 55.0 | 53.0 | 63.0 | 59.0 | 52.0 | 51.0 | 59.0 |
| geometric_shapes | 65.0 | 65.0 | 69.0 | **72.0** | **72.0** | 60.0 | 69.0 | 68.0 |
| hyperbaton | 77.0 | 72.0 | 65.0 | 80.0 | 81.0 | 83.0 | 75.0 | **86.0** |
| logical_deduction_seven_objects | 47.0 | 54.0 | 53.0 | 45.0 | 49.0 | **62.0** | 44.0 | 51.0 |
| movie_recommendation | 59.0 | 45.0 | 54.0 | 68.0 | 61.0 | 63.0 | 64.0 | **70.0** |
| multistep_arithmetic_two | 36.0 | 50.0 | 20.0 | 47.0 | 20.0 | 66.0 | 12.0 | **77.0** |
| object_counting | 81.0 | 81.0 | 82.0 | 83.0 | 79.0 | 85.0 | 75.0 | **87.0** |
| ruin_names | 69.0 | 60.0 | 57.0 | **76.0** | 57.0 | 72.0 | 57.0 | 70.0 |
| salient_translation_error_detection | 47.0 | 47.0 | 45.0 | **59.0** | 49.0 | 53.0 | 49.0 | 48.0 |
| snarks | 69.4 | 76.4 | 79.2 | 72.2 | 75.0 | 72.2 | 73.6 | **77.8** |
| sports_understanding | 86.0 | 75.0 | 69.0 | 91.0 | 72.0 | 91.0 | 74.0 | **93.0** |
| tracking_shuffled_objects_seven_objects | 70.0 | 69.0 | 70.0 | 91.0 | 88.0 | **94.0** | 81.0 | 93.0 |
| *Average* | 60.30 | 60.42 | 58.30 | 66.08 | 61.04 | 67.56 | 59.16 | **70.05** |

Gemini results in the main text largely hold. On Claude 3.5 Sonnet, we observe an almost identical high-level trend to Gemini, where each round of BRIDGE incrementally improves performance up to 2G. On the other hand, while Mistral models seemingly benefit less from scaling demonstrations especially in the smaller Mistral NeMo (e.g., sometimes the generate step leads to drops in performance) directly, the improved quality of the generated demonstrations still enables successive optimize step to improve on the preceding round, demonstrating the effectiveness of BRIDGE even when the model does not benefit from scaling examples directly.

## C.6 TRANSFERRING LEARNED DEMONSTRATIONS FROM GSM-HARD TO GSM-8K

In this section, we investigate whether the BRIDGE-discovered demonstrations can transfer across related but distinct datasets. Specifically, we investigate the extent to which the demonstrations found on GSM-Hard (Table 2) generalize to the original GSM-8K and we show the result in Ta-

Table 15: Test accuracy of Claude 3.5 Sonnet (`claude-3-5-sonnet@20240620`) on BBH tasks. Refer to captions of Table 1 for detailed explanations.

| Tasks | Reinf. ICL | Iterative Reinf. | | BRIDGE (Ours) | | | | |
|---|---|---|---|---|---|---|---|---|
| # Iterations | 0 | 1 | 2 | 1O | 1G | 2O | 2G | 3O |
| causal_judgement | 64.0 | 68.0 | 65.3 | 62.7 | 69.3 | **73.3** | 70.7 | 65.3 |
| date_understanding | 94.0 | 95.0 | **96.0** | **97.0** | 94.0 | 95.0 | 96.0 | 95.0 |
| disambiguation_qa | 73.0 | 82.0 | 79.0 | 81.0 | **87.0** | **87.0** | 84.0 | 86.0 |
| dyck_language | 68.0 | 68.0 | 65.0 | 74.0 | 85.0 | 90.0 | **92.0** | 87.0 |
| formal_fallacies | 93.0 | 94.0 | 97.0 | 96.0 | 95.0 | **98.0** | 96.0 | 95.0 |
| geometric_shapes | 92.0 | 94.0 | **98.0** | 88.0 | 90.0 | 85.0 | 96.0 | 89.0 |
| hyperbaton | **100.0** | **100.0** | **100.0** | **100.0** | **100.0** | **100.0** | **100.0** | **100.0** |
| logical_deduction (7) | 92.0 | 96.0 | 96.0 | 89.0 | 95.0 | **97.0** | 91.0 | 93.0 |
| movie_recommendation | 87.0 | 90.0 | 92.0 | 89.0 | 90.0 | 88.0 | **93.0** | 90.0 |
| multistep_arithmetic_two | 99.0 | 99.0 | 99.0 | 99.0 | 99.0 | 99.0 | **100.0** | **100.0** |
| object_counting | **100.0** | **100.0** | **100.0** | **100.0** | **100.0** | **100.0** | **100.0** | **100.0** |
| ruin_names | 93.0 | 93.0 | **94.0** | 91.0 | **94.0** | **94.0** | 92.0 | **94.0** |
| salient_translation_error_detection | 71.0 | 71.0 | **73.0** | 71.0 | 72.0 | **73.0** | **73.0** | **73.0** |
| snarks | 97.2 | 97.2 | 97.2 | 95.8 | 95.8 | **98.6** | **98.6** | 97.2 |
| sports_understanding | 92.0 | 91.0 | **94.0** | 93.0 | **94.0** | **94.0** | 93.0 | 91.0 |
| tracking_shuffled_objects (7) | **100.0** | **100.0** | **100.0** | **100.0** | **100.0** | **100.0** | **100.0** | **100.0** |
| *Average* | 88.45 | 89.89 | 90.35 | 89.16 | 91.26 | 92.00 | **92.20** | 90.97 |

ble 16, where we compare the performance of the demonstrations directly transferred from GSM-Hard at different stages of BRIDGE against directly optimizing on GSM-8K. We find that whereas the demonstrations generated from (iterative) reinforced ICL led to small deterioration of GSM-8K performance, we found the transferred demonstrations from BRIDGE led to small improvement even though the Gemini 1.5 Pro performance on GSM-8K has been rather saturated. While optimizing directly on GSM-8K unsurprisingly led to the highest performance given that there is no distribution shift, we also find that the GSM-Hard demonstrations exhibit considerable generalizability.

Table 16: Comparison of the transferred BRIDGE-generated demonstrations on GSM-Hard vs. directly running BRIDGE on GSM-8K. Runs with performance deteriorations w.r.t. the 0-shot results are marked in red in the table.

| Tasks | 0-shot | Reinf. ICL | Iterative Reinf. | | BRIDGE (Ours) | | | | |
|---|---|---|---|---|---|---|---|---|---|
| # Iterations | - | 0 | 1 | 2 | 1O | 1G | 2O | 2G | 3O |
| Direct | 91.92 | 93.81 | 93.06 | 92.68 | 93.81 | 93.18 | **94.70** | 94.19 | 93.94 |
| Transferred | - | 90.66 | 91.79 | 91.16 | **93.81** | 92.55 | **93.81** | 93.18 | 91.16 |

# D  COMPUTATIONAL COST ANALYSIS

In this section, we provide a computational cost analysis of BRIDGE. In general, since BRIDGE consists of multiple rounds of "Optimize" and "Generate" steps, here we analyze each step in detail.

- **Optimize**: The cost of the "optimize" step depends on the budget allocated ($n_{eval}$ in Line 5 of Algorithm 2), which is user-configurable. If we opt for iterative optimization (such as using Bayesian optimization in the main section of the paper, or random search in App. C.1), each "optimize" step thus entails $n_{eval}$ LLM inferences on the validation set. As shown in the App. C.1, it is also possible to use non-iterative method based on retrieval or embedding diversity, in which case each "optimize" step entails a single round of LLM inferences on the validation set (or the train set, if we use the dataset for both training and validation).

- **Generate**: The "generate" step always involves a single round of LLM inferences on the train set where we simply use the optimized examples from the "optimize" step above as demonstrations and run inference again on the train set.

