# OpenReview forum: "From Few to Many: Self-Improving Many-Shot Reasoners Through Iterative Optimization and Generation"
_ICLR.cc/2025/Conference — ICLR 2025 Poster_

### Official Review · Reviewer_qv5C · 2024-10-30

**Soundness:** 4
**Presentation:** 2
**Contribution:** 3
**Rating:** 6
**Confidence:** 3

**Summary:**

This paper presents a study on the performance of many-shot in-context learning on NLP tasks that require reasoning process (the reinforced ICL setup). The authors first define a importantce scoring function of input examples based on estimated input gradient, and observe that 1) many-shot ICL could be driven by a few high-performing examples, 2) scaling demonstration examples could still be beneficial with improved quality of reasoning. Motivated by these observation, the authors propose a method called BRIDGE to improve many-shot ICL in an iterative manner. Experiments of Gemini across a diverse set of tasks show the effectiveness of BRIDGE.

**Strengths:**

1. Overall, this paper is well writtern and clear to follow.
2. The proposed method is well-motivated through experiment results (Figure 1 and Figure 2). The observation that many-shot ICL could be driven by few high-performing examples is interesting.
3. The proposed method shows decent performance compared to the baseline.
4. The method is evaluated on various  challenging datasets

**Weaknesses:**

1. The analysis is limited to reinforced ICL setting that only considers reasoning-heavy tasks. It's not clear what many-shot ICL would perform on simpler tasks that do not heavily relies on reasoning. Also, could BRIDGE anyway be helpful on simpler tasks?
2. The experiments are only evaluated with Gemini. What would be the performance of proposed method on open-source methods like Llama-3.1? More importantly, for open-source model, would there exists a better way of importance attribution? Also, for base model where the reasoning ability is limited, would the proposed method also be helpful?
3. Lack of the analysis on the computation cost. What's the computational cost of optimize and generate step of BRIDGE, respectively?
4. The definition of importance score function is well-justified. In section 2 Line 153, the importance of each example is estimated by the averate performance of randomly sampled demonstrations on the validation set. The variance would be huge considering different choices of subsets and orders. Have the author performed analysis on the varaince of importance score?
5. Importance attribution is not well justified. In section 2 Line 153, it's not clear how importance score is actually estimated and why it would make sense.
6. The description of "optimzie" step is not clear (Line 261). The text in page 6 is generally unclear to me. For example, what does "TCM" stand for in algorithm 2 step 6? How does the problem become a "bi-objective" problem and how the optimize step is connected to "Bayes optmization"?

Minor issue: ")" is missing in Line 149

**Questions:**

Please refer to the "weakness".

---

> ### Author Response · Authors · 2024-11-22
> **Response (Part 1/3)**
>
> We thank the reviewer for their insightful and positive comments! Please see below for our detailed response, and we hope the reviewer could consider increasing the rating if they feel their concerns have been sufficiently addressed.
>
> > The analysis is limited to reinforced ICL setting that only considers reasoning-heavy tasks. It's not clear what many-shot ICL would perform on simpler tasks that do not heavily relies on reasoning. Also, could BRIDGE anyway be helpful on simpler tasks?
>
> We primarily focused on the reasoning-heavy tasks as they are broadly deemed to be a class of problems that is demanding even for the frontier models (e.g., the SoTA performance of BIRD dataset that we study in Table 4 is around 70% at the time of writing, even with complicated agentic designs and state-of-the-art LLMs). That the crucial intermediate outputs have to be model-generated has also made the problem challenging, but at the same time these facts also mean that 1) the potential room of improvement is ample and 2) any performance bottlenecks can simply be overcome by switching to a stronger model. In fact, while [1], the seminal work in modern many-shot ICL has covered a wide range of tasks, one of their primary contributions is to exactly address the same setup in this paper, hence the proposal of reinforced ICL and unsupervised ICL. Several recent works and popular prompt optimization frameworks [2-4] have also largely focused on this setup, and modern LLMs largely report performances on reasoning-heavy benchmarks (BBH and MATH datasets we considered are common choices) as a demonstration of their general caliber, and we believe the emphasis by these prominent prior works has underscored the importance of this class of problems, which motivated our focus in the paper.
>
> On the other hand, we do agree that studying the utility of many-shot ICL for other problems can be an important direction. While we acknowledge that the present work has primarily focused on the reinforced ICL setting (Line 535 in conclusion), we have also included a study on extending BRIDGE to information-heavy tasks like low-resource machine translation (MT) in Appendix C.4 where we keep the overall BRIDGE pipeline of iterative “optimize” and “generate” intact but modify, where appropriate, the exact implementation of these steps. While the section is a preliminary study, we hope that it shows the potential promise of BRIDGE in an even wider range of applications, which will be a key next step following up to this work.
>
> > The experiments are only evaluated with Gemini. What would be the performance of proposed method on open-source methods like Llama-3.1?
>
> We thank the reviewer for their suggestions and we agree that it is important to ensure the findings are not model-specific. As such, we have added results on Mistral large (mistral-large-2407), which is an open-sourced model as the reviewer suggested, and also Claude 3.5 Sonnet. We refer the reviewer to the App. C5 of the updated manuscript.
>
> > More importantly, for open-source model, would there exists a better way of importance attribution?
>
> We agree that there should exist at least more efficient methods for importance attribution when we lift the restrictions on model access (although we have some clarifications regarding this part – please see our response below). For example, currently we assume black-box, textual output-only access which is highly restrictive but is also the most generally applicable and model-agnostic – we rely only on textual outputs which are provided by all models, instead of any model-specific outputs or features. Currently the importance is currently estimated via repeated sampling, but for a white-box model where we can access the internal parameters and/or gradients, importance can be more efficiently done with, for example, input gradient / saliency attribution or some variant of Shapley value attribution [5] (which only requires output log-probability rather than gradients). We agree that this can be an important next step for this work, but we believe those are standalone improvements separate from the contributions in this paper.
>
> We have incorporated a discussion of this in the Conclusion of the updated manuscript.

---

> ### Author Response · Authors · 2024-11-22
> **Response (Part 2/3)**
>
> > for base model where the reasoning ability is limited, would the proposed method also be helpful?
>
> Similar to Reinforced ICL and other bootstrapping methods (e.g. [2,4,7]), BRIDGE by default initializes with zero-shot CoT to obtain the initial example sets, so the model indeed needs some reasoning ability to begin with (this is also true for the other methods mentioned above). However, even for a weaker model, as long as there is some in-context learning ability, we believe there are various straightforward ways to reduce the reliance on the models’ initial reasoning capabilities. For example, instead of starting with zero-shot CoT, we can initialize with few-shot CoT with a few examples either annotated by humans (e.g., this has been used for highly challenging tasks like BIRD in Table 4) or a stronger model (i.e., similar to distillation in DSPy).
>
> > Lack of the analysis on the computation cost. What's the computational cost of optimize and generate step of BRIDGE, respectively?
>
> The cost of the “optimize” step depends on the budget allocated ($n_{\mathrm{eval}}$ in Line 5 of Algorithm 2), which is user-configurable. If we opt for iterative optimization (such as using Bayesian optimization in the main section of the paper, or random search in App. C.1), each “optimize” step thus entails $n_{\mathrm{eval}}$ LLM inferences on the validation set. As shown in the App. C1 (added materials during rebuttal), it is also possible to use non-iterative method based on retrieval or embedding diversity, in which case each “optimize” step entails a single round of LLM inferences on the validation set (or the train set, if we use the dataset for both training and validation). In general, as expected with any optimization method, there is a cost-performance tradeoff; running an iterative algorithm multiple times incurs a higher one-off cost but generally performs better – this has also been shown in related works like [3].
>
> The “generate” step always involves a single round of LLM inferences on the train set where we simply use the optimized examples from the “optimize” step above as demonstrations and run inference again on the train set.
>
> We agree with the reviewer that understanding the computational cost is an important aspect and we’ve added the discussions above to the paper (App. D).
>
> > The definition of importance score function is well-justified. In section 2 Line 153, the importance of each example is estimated by the averate performance of randomly sampled demonstrations on the validation set. The variance would be huge considering different choices of subsets and orders. Have the author performed analysis on the varaince of importance score?
>
> We thank the reviewer for their feedback. While sampling-based importance approximation inevitably involves some kind of randomness and variance, one key point we’d like to note is that it is meant to be a motivating experiment proving the point that “It does not always take “many shots” to achieve many-shot performance” (caption of Fig 1) and that “many-shot performance can be driven by few high-performing examples” (Page 3) only – we do not have to perform this step every time we run BRIDGE. Instead, the key insight we’d like to show is that 1) **there exists** some way such that we can approach the many-shot results with much fewer examples, whereas on the other hand 2) **there also exists** another way such that even including many examples does not improve performance – these motivate the proposition that we should not simply scale up the number of examples and should instead pay attention to what examples we include if different selection strategies lead to drastically different outcomes. The importance estimation and ranking procedure is simply an exemplary procedure to practically construct both selection methods in one go (by traversing the examples in ascending and descending orders) in an interpretable manner. Furthermore, given the fact that this procedure leads to well-separated lines in Fig 1 which correspond to the two aforementioned criteria, we argue that the importance score has fulfilled its practical purpose and that its noise should at least not dominate over the signal (as otherwise we will not be able to obtain well-separated lines).
>
> > Importance attribution is not well justified. In section 2 Line 153, it's not clear how importance score is actually estimated and why it would make sense.
>
> We compute the importance by estimating the input gradient of the target metric (e.g., validation accuracy) w.r.t. each example – input gradient attribution is a well-used technique for importance/saliency estimation. Due to page limit, we had to leave the exact details on how to approximate the importance score in Appendix A. We’d like to refer the reviewer to Appendix A for detailed explanations.

---

> ### Author Response · Authors · 2024-11-22
> **Response (Part 3/3)**
>
> > The description of "optimzie" step is not clear (Line 261). The text in page 6 is generally unclear to me. For example, what does "TCM" stand for in algorithm 2 step 6? How does the problem become a "bi-objective" problem and how the optimize step is connected to "Bayes optmization"?
>
> The “Optimize” step, as described in Line 265, is motivated by the findings in Sec 2: given that “using all examples as demonstrations is sub-optimal”, the natural question is how do we actually pick the demonstrations that is indeed optimal, and “optimize” step is designed to achieve this. Using the binary vector representation of example sets mentioned in Line 162 onward, the objective is to find some $\mathbf{e}^* \in [0, 1]^m$ to be used as demonstrations for the next “generate” step. Practically, we use a Bayesian optimization for this optimization problem – we opted for Bayesian optimization because it is sample-efficient and zeroth-order (i.e., no gradient information required), although it is also possible to use an alternative optimizer – for example, in App C.1 we also used random search (BRIDGE-RS) where instead of using Bayesian optimization (BO), we randomly sample m sets of examples and pick the best based on validation performance. Besides explicit iterative optimization, it is also possible to use retrieval based on embedding similarity or heuristics like diversity (App. C1) for this purpose; practically, however, we find using BO performed the best, in consistency with prior works like [4, 6] which also used some formulation of BO.
>
> The “bi-objective” is a practical design choice, and means that instead of only optimizing for $\mathbf{e}^* \in \{0, 1\}^m$ that maximizes the validation performance, we also encourage some kind of sparsity as regularization. One straightforward approach is to simply treat sparsity as some penalty term, but the problem is that we do not know how much weight this term should be. The bi-objective formulation is to bypass this and instead of setting the penalty weight as a hyperparameter, we essentially aim to both 1) maximizes validation performance and 2) maximizes sparsity (as measured by the number of examples in the selected set) – it is “bi-objective” since there are two objectives here. This part has also been explained in Line 300 to 310, and we’ve modified this part of the paper to make it even clearer.
>
> “TCM” is a typo and we thank the reviewer for pointing it out. It should be “TCH”, which, as explained in Line 316, is a canonical way to scalarize the two optimization objectives above. This has been corrected in the updated version of the manuscript.
>
> References
>
> [1] Agarwal, R., Singh, A., Zhang, L. M., Bohnet, B., Rosias, L., Chan, S., ... & Larochelle, H. (2024). Many-shot in-context learning. NeurIPS.
>
> [2] Khattab, O., Singhvi, A., Maheshwari, P., Zhang, Z., Santhanam, K., Vardhamanan, S., ... & Potts, C. (2024). Dspy: Compiling declarative language model calls into self-improving pipelines. ICLR.
>
> [3] Wan, X., Sun, R., Nakhost, H., & Arik, S. O. (2024). Teach better or show smarter? on instructions and exemplars in automatic prompt optimization. NeurIPS.
>
> [4] Opsahl-Ong, K., Ryan, M. J., Purtell, J., Broman, D., Potts, C., Zaharia, M., & Khattab, O. (2024). Optimizing Instructions and Demonstrations for Multi-Stage Language Model Programs. arXiv preprint arXiv:2406.11695.
>
> [5] Enouen, J., Nakhost, H., Ebrahimi, S., Arik, S. O., Liu, Y., & Pfister, T. (2023). TextGenSHAP: Scalable Post-hoc Explanations in Text Generation with Long Documents. arXiv preprint arXiv:2312.01279.
>
> [6] Chen, L., Chen, J., Goldstein, T., Huang, H., & Zhou, T. (2023). Instructzero: Efficient instruction optimization for black-box large language models. arXiv preprint arXiv:2306.03082.
>
> [7] Wan, X., Sun, R., Dai, H., Arik, S. O., & Pfister, T. (2023). Better zero-shot reasoning with self-adaptive prompting. ACL.

---

> > ### Comment · Reviewer_qv5C · 2024-11-24
> > **Acknowledgement**
> >
> > I'd like to thank the authors for the effort of putting the responses. It improves my understanding of the paper and addressed my concerns. I think is paper would be valuable to the current research community: 1) the problem is important, 2) the observation is interesting, 3) the method is well motivated and effective.

---

> > > ### Author Response · Authors · 2024-11-26
> > >
> > > Dear Reviewer qv5C,
> > >
> > > We thank the reviewer for their positive and encouraging feedback, and we are glad that the concerns have been addressed. We are happy to discuss further if there are any further points to address to further strengthen the paper and contribute to a higher score.

---

### Official Review · Reviewer_YxzH · 2024-11-02

**Soundness:** 2
**Presentation:** 3
**Contribution:** 2
**Rating:** 8
**Confidence:** 3

**Summary:**

The paper studies the many-shot in-context learning using Gemini family of models on BigBenchHard (BBH). Specifically, the authors demonstrate that identifying a subset of influential examples reinforced with model-generated reasoning can significantly outperform naively using all available examples. They also propose BRIDGE, a simple approach that instantiates this finding to iteratively seeking for the subset of examples and refining model-generated reasoning for these examples. BRIDGE is evaluated on the subset of BBH tasks and the substantial gains over the considered baselines are demonstrated.

**Strengths:**

* Paper studies many-shot in-context learning, the important novel prompting technique that is used to significantly improve LLMs' predictions. Given that, this technique was made available only recently with the rise of long-context LLMs, studying its behaviour is important topic that can be relevant to broad community.

* The proposed approach is simple and algorithmically applicable to both close-source and open-source models.

**Weaknesses:**

* The important baseline might be missing to ensure the strength of the claims, please see Questions section.

* I would also ask for including similar analysis for other LLMs beside close-source ones like Gemini. At least on the subset of the considered tasks, to see whether the conclusions generalize to other models.

**Questions:**

Can you please provide the baseline when a model conditions on all available examples but with all examples using model generated reasoning? This baseline can decouple the importance of subselecting a subset of examples and influence of having model-generated reasoning, and strengthen your claims if, indeed, you need both rather than only model-generated reasoning. The recent works [1, 2] can be used to ensure that a model is constrained to generate reasoning given the input and the ground truth answer.

[1] Hu et al. Amortizing intractable inference in large language models. ICLR 2024

[2] Zhao et al. Probabilistic Inference in Language Models via Twisted Sequential Monte Carlo. ICML 2024

---

> ### Author Response · Authors · 2024-11-22
> **Response**
>
> We thank the reviewer for their constructive feedback! To our understanding, the main concerns of the reviewer lies in 1) results on additional LLMs and 2) adding an infilling baseline – we address both in our response. We hope that the reviewer could consider increasing the score if they feel the concerns have been sufficiently addressed.
>
> > I would also ask for including similar analysis for other LLMs beside close-source ones like Gemini. At least on the subset of the considered tasks, to see whether the conclusions generalize to other models.
>
> We thank the reviewer for their suggestions and we agree that it is important to ensure the findings are not model-specific. As such, we have added results on Mistral large (mistral-large-2407), which is an open-sourced model, and also Claude 3.5 Sonnet. Please see below for the aggregated results on BBH tasks and Appendix C.5 for detailed per-task breakdown.
>
> Aggregated BBH results in 4 different LLMs
> |                      Model                     | Reinf ICL | Iterative Reinf. |       | BRIDGE (Ours) |       |           |           |           |
> |:----------------------------------------------:|----------:|-----------------:|------:|--------------:|------:|----------:|----------:|----------:|
> |                  # Iterations                  |         0 |                1 |     2 |            1o |    1g |        2o |        2g |        3o |
> |       Gemini 1.5 Pro (gemini-1.5-pro-001)      |     79.61 |            81.61 | 82.37 |         82.11 | 84.61 |     85.77 | **87.13** |   _86.33_ |
> |     Gemini 1.5 Flash (gemini-1.5-flash-001)    |     80.25 |            81.91 | 80.72 |         81.61 | 82.79 | **83.79** |   _83.77_ |     83.25 |
> |       Mistral Large (mistral-large@2407)       |     84.82 |            83.22 | 87.08 |         86.65 | 83.70 |   _88.07_ |     82.80 | **88.52** |
> | Claude 3.5 Sonnet (claude-3-5-sonnet@20240620) |     88.45 |            89.89 | 90.35 |         89.16 | 91.26 | _92.00_ |   **92.20** |     90.97 |
>
>
> > The important baseline might be missing to ensure the strength of the claims, please see Questions section.
>
> > Can you please provide the baseline when a model conditions on all available examples but with all examples using model generated reasoning? This baseline can decouple the importance of subselecting a subset of examples and influence of having model-generated reasoning, and strengthen your claims if, indeed, you need both rather than only model-generated reasoning. The recent works [1, 2] can be used to ensure that a model is constrained to generate reasoning given the input and the ground truth answer.
>
> We thank the reviewer for their insightful suggestion. First, we’d like to note that we already provide some baselines that 1) use all (correctly predicted) examples and 2) use model-generated reasoning (“reinf ICL” uses all examples (input, intermediate output and targets) that the model predicted correctly whereas “All CoT” use all examples, regardless of the prediction correctness), although they differ from the reviewer’s suggestions (which we refer to as “infill” hereafter) that the final labels are not revealed to the model but are used for post-hoc filtering (for reinforced ICL). Having said that, we agree that the “infilling” can indeed be a sensible baseline and thank the reviewer for bringing it up. We have added the suggested baseline as “All - infill” in Tables 1 and 3 using the the prompt adapted from Appendix C of [1] the reviewer suggested, and we found that on Gemini 1.5 Pro, it performs roughly on par with Reinforced ICL (i.e., {input, intermediate output, target} of all correctly predicted train samples, but the targets are not revealed to the model). On Gemini 1.5 Flash, it performs slightly worse than Reinforced ICL but in both models they performed stronger than the “All - Direct” and “All - CoT” baselines, but BRIDGE outperformed all these baselines by a large margin. We hope the additional results address the reviewer’s concerns.

---

> > ### Comment · Reviewer_YxzH · 2024-11-25
> >
> > I thank the authors for their clarifications and for adding the new baseline. However, I am still not convinced since I believe the current infilling baseline is underperforming to convincingly show that it is necessary to use BRIDGE (as the combination of generating model reasoning and filtering out "bad" examples). Indeed, authors adopted prompting baseline from [1] rather than the principled method from [1] to generate reasoning for each of the examples. Below, I provide the table with the baselines that are currently used in the paper to compare with BRIDGE.
> >
> > |                             | All examples are used | w/ reasoning                                | GT answer  |
> > |-----------------------------|-----------------------|---------------------------------------------|------------|
> > | Direct                      | yes                   | no                                          | yes        |
> > | All CoT                     | yes                   | might not correspond to ground truth answer | not always |
> > | Infill (prompting baseline) | yes                   | might not correspond to ground truth answer | yes        |
> > | Reinf. ICL                  | no                    | yes                                         | yes        |
> > | Reinf. ICL (iterative)                 | no                    | yes                                         | yes        |
> >
> > All CoT baseline generates reasoning and the final answer for all the examples. As Table 1 shows, this baseline underperforms Direct method that does not employ any reasoning. Given that model is not perfect, my guess is that it happens due to many incorrect answers by the model after generating reasoning. Once such examples are filtered out (Reinf. ICL), we can see that using model-generated reasoning actually helps, however now not all the available examples are utilized. As we can further see from the subsequent Reinf. ICL iterations, the more examples are included (Table 10 in the Appendix) and the performance becomes better.
> >
> > Given these observations, my concern is that once we can employ *all the examples* and the model generated reasoning will be corresponded to the ground truth Y for the given input X, then the improvements brought by BRIDGE will be minor.
> >
> > In the current form, the newly added infilling baseline is somewhat similar to All CoT baseline. Indeed, compared to All CoT that does not constrain the model to provide the ground truth answer after generating the corresponding reasoning chain, Infill prompting only very softly constrain the model to generate reasoning associated with the pair (input prompt, gt answer). In [1] authors show that such infilling by prompting heavily (2x worse in Table 2 of [1]) underperforms principled approach to infill. Even, with such soft constraint the performance increased considerably, i.e., almost +5% on average compared to All CoT.
> >
> > I believe that to accurately show that BRIDGE will still bring improvements upon the infilling baseline, authors should implement the principled approach to sample the reasoning chain for each training example. In particular, let p(XZY) be the likelihood defined by a model, where (for BBH tasks according to authors' Appendix)
> >
> > X =
> > ```
> > You will be given a question. Think step by step before giving a final answer to this question. Show your final answer {{ TASK_SPECIFIC_CONSTRAINTS }} between <answer> and <\answer>
> >
> > {{ EXAMPLES }}
> > ==
> >
> > {{ QUESTION }}
> > ```
> > Z =
> > ```
> > {{ reasoning) }}
> > ```
> >
> > Y =
> >  ```
> > <answer>{{ GT_ANSWER }}<\answer>
> > ```
> > and where XZY means concatenation of strings, and the model reads the input tokens following left-to-right order.
> >
> > Then, given pairs of (X, ground truth Y), one can employ the methodology developed by [1] to sample reasoning tokens $p(Z | X, Y) \propto p(XZY)$, effectively ensuring that the model generated reasoning
> > * is conditioned on the ground truth answers => generated reasoning should lead to the corresponding Y with high probability
> > * the overall prompt is *the same* that is used by all other approaches, including BRIDGE
> >
> > After sampling, the result is (X, Z, Y) triplets that can be used as ICL examples.
> >
> > [1] Hu et al. Amortizing intractable inference in large language models. ICLR 2024

---

> > > ### Author Response · Authors · 2024-11-26
> > >
> > > We thank the reviewer for their feedback. Before responding to the new points, we’d first like to note that, to our understanding, the original concern is about *“decouple the importance of subselecting a subset of examples and influence of having model-generated reasoning”*. Besides the new infilling baseline, **the “Iterative Reinforced ICL” baseline is identical to BRIDGE except BRIDGE additionally selects a subset of correct examples** rather than using all correct examples in Iterative Reinforced ICL). We believe **the fact that BRIDGE outperforms Iterative Reinforced ICL already proved the importance of sample selection**. While the reviewer correctly mentioned that *"performance becomes better with subsequent Reinf ICL iterations"*, it's true only *for some models/tasks* (e.g., it does *not* help in Tables 2&3), whereas BRIDGE consistently improves with a larger margin; we argue this already shows the strength of selection.
> > >
> > > > ... I believe the current infilling baseline is underperforming to convincingly show that it is necessary to use BRIDGE ... authors adopted prompting baseline from [1] rather than the principled method from [1] ...
> > >
> > > First, the reason why we ran the prompting-based infilling in [1] instead of its main proposal (GFlowNet tuning) is that **the latter requires white-box access for tuning on a customized objective; our method requires black-box access only (we mentioned this in page 3, Line 159)** compatible with virtually any model, including those with only prediction API-based access like Gemini/Claude which also have state-of-the-art (long-context) performance. Under this setup, only prompting-based infilling is a fair comparison because the tuning-based methods do not work without internal model access: **In fact, even the authors of [1] themselves only experimented the GFlowNet-based tuning on open-sourced models like GPT-J rather than proprietary, frontier models**. While we thank the reviewer for the table of comparison (we even believe it is a great idea to incorporate it into the paper), we argue that at least to our knowledge, in the black box setup, “reinforced ICL” and “prompting-based infilling” baselines are closest to satisfying all of “All examples are used” (reinforced ICL uses all correctly predicted samples; infilling uses all available samples), “w/ reasoning” (both use model-generated reasoning) and “GT answer”. We’d be happy to discuss more if the reviewer has better suggestions **that are compatible with our setup**.
> > >
> > > > my concern is that once we can employ all the examples ... then the improvements brought by BRIDGE will be minor.
> > > > Even, with such soft constraint the performance increased considerably, i.e., almost +5% on average compared to All CoT.
> > >
> > > Regarding the points above, firstly we note that while the reviewer is correct that infilling led to +5% on average to All CoT (which is a weak baseline since it does not use reasoning at all), it does not even outperform reinforced ICL, which uses a fraction of all examples as demonstrations – BRIDGE still outperformed both baselines by a large margin, so we do not think that the new result itself gives strong indication that “the improvement brought by BRIDGE will be minor”.
> > >
> > > While we acknowledge that it is possible to generate better rationales, through GFlowNet-style finetuning or otherwise, there are several clarifications we’d like to make.
> > >
> > > - **They focus on different setups**: like we mentioned earlier, we argue it is often either impossible (e.g., black-box models) or expensive (since finetuning is required) to run on some models, and we consider our approach, which is a black-box method to enhance demonstration quality, to be more generally applicable. Please note that we are not trying to discredit tuning-based approaches; we simply mean that they are suitable for different setups and the values for both should be recognized – we do agree that a method combining them effectively, such as a method bringing test-time methods to training, can be a promising direction.
> > >
> > > - We urge the reviewer also not to overlook the overall pipeline of BRIDGE, which also contains an iterative process to bootstrap demonstrations to leverage long-context capabilities of frontier models; **the importance of these do not diminish even if we have better rationales to start with**, with demonstration selection (ours) or GFlowNet-style tuning, unless they are absolutely perfect with no room for improvement  – we believe this is not the case regardless of the method used.
> > >
> > > > the overall prompt is the same that is used by all other approaches, including BRIDGE
> > >
> > > We’d note that **there is no prompt inconsistency**. The prompt template the reviewer referred to is only used for rationale generation; for inference, identical prompt formats are used.
> > >
> > > We thank the reviewer again and we welcome further engagements. We hope the reviewer could consider increasing their score if they feel the concerns are sufficiently addressed.

---

> > > > ### Comment · Reviewer_YxzH · 2024-11-26
> > > >
> > > > I thank the authors for their clarifications.
> > > >
> > > > I am still not convinced with the authors explanation. I am aware that the suggested approach [1] considers white-box setting.  I acknowledge the wider applicability of BRIDGE, however to see if it's the only benefit of BRIDGE I requested the baseline. Thats why in my initial review I suggested **using open-source** models to allow running meaningful baseline allowing validating the efficacy of BRIDGE at combining refining rationales and sub selecting demonstrations. I believe that the paper will greatly strengthen from showing proper baseline that employs all the examples with consistently generated rationales.
> > > >
> > > > Given lack of time, setting up and running this baseline seem to be infeasible, I suggest the following proxy experiment: Table 10 shows the number of examples used for each of the methods in Table 1. Can authors run BRIDGE considering the training sets of Reinf. ICL for each task rather than the real full training set, i.e., 36 training examples for causal_judgement and so on?
> > > >
> > > > > there is no prompt inconsistency
> > > >
> > > > There is exactly the prompt inconsistency that might lead to the baseline underperformance. The prompt used to generate rationales for the infilling baseline *is different* from the prompt used for the inference. Thus, the rationales generated with the corresponding prompt does not necessarily lead to the ground truth answer for the corresponding input. You can simply check the accuracy by inserting the generated rationales and continuing generation to obtain the final answer.

---

> ### Author Response · Authors · 2024-11-26
> **Could you clarify the difference between the proxy experiment vs. iterative reinf. ICL?**
>
> We thank the reviewer for their prompt response, and we will shortly respond to the comment fully, but before that, we'd like a quick clarification regarding the proxy experiment requested:
>
> *How is the requested experiment different from the "**iterative** reinf ICL" baseline?*
>
> For the benefit of the reviewer, we paste the description from the caption of Table 1 (lines 383-385) below (with emphasis added):
> > "Iterative Reinf.” refers to the iterative variant of reinforced many-shot ICL where we directly **use all the generated correct examples** from the previous round as demonstrations for the next round **without the optimize step**.
>
> Note that the "optimize" step performs example selection. Iterative reinf ICL dispenses of this step and only performs expansion iteratively with the full train set from Reinf ICL. *This baseline is present in all experiments we have performed (e.g., Table 1-3)*.
>
> We thank the reviewer again for providing actionable items, and we look forward to their clarification.

---

> > ### Comment · Reviewer_YxzH · 2024-11-26
> >
> > If I understand correctly, to construct the **support set**, i.e., set of demonstrations that will be used to evaluate Reinf. ICL, authors have run All CoT and filtered out the examples where the generated answer after model generated rationale was incorrect. Subsequently, these "good" examples are used as ICL support set to evaluate the performance on the test set, i.e., CoT prompting [1]. Iterative Reinf. ICL just refines the **support set** by generating rationales and answers for each example of the **support set** with randomly sampled demonstrations from the rest of the **support set** (again CoT prompting). So these methods, try to refine the support to ultimately obtain the entire training set with all examples containing valid rationales, meaning they lead to ground truth answer.
> >
> > The requested experiment suggests to consider the support set of Reinf. ICL, i.e., after filtering out the examples where the generated answer after model generated rationale was incorrect, as the only training set that is available to run BRIDGE. It basically assumes that we already have the training/support set with valid triplets (input, rationale, gt answer) and now we want to see whether iterative refinement along with subset selection of BRIDGE leads to improvements. And that was basically the idea behind the proper infilling baseline as before.
> >
> > [1] Wei et al. Chain-of-Thought Prompting Elicits Reasoning in Large Language Models. NeurIPS 2022

---

> ### Author Response · Authors · 2024-11-28
> **We ran BRIDGE on the proxy experiments the reviewer suggested and still observe improvements.**
>
> We thank the reviewer for their time and patience while we run their requested experiments.
>
> > I acknowledge the wider applicability of BRIDGE, however to see if it's the only benefit of BRIDGE I requested the baseline. Thats why in my initial review I suggested using open-source models to allow running meaningful baseline allowing validating the efficacy of BRIDGE at combining refining rationales and sub selecting demonstrations.
>
> We thank the reviewer for acknowledging the value of BRIDGE. Firstly, we would like to emphasize that "wider applicability" is an *important* advantage over tuning methods given that at least as of now, many SoTA models are proprietary (including those shown with strong many-shot abilities, e.g., Gemini & Claude) where there is no white-box access, and in our opinion the ability to run on these models is a crucial advantage. Having said that, we did add experiments in open-sourced models (Mistral large and Mistral Nemo (12B)) in the rebuttal.
>
> > Prompt consistency.
>
> There was an initial misunderstanding regarding the meaning of the term, and we thank the reviewer for clarifying what they meant.
>
> > Additional baseline.
>
> We highly appreciate the reviewer for clarifying the request and providing an actionable item and a critical baseline. We run experiments on gemini-1.5-pro-001 using the suggested setting (we termed these experiments "iterative reinf. (restricted)" and "BRIDGE (restricted)", which refer to Iterative Reinf ICL and BRIDGE with the **train support set restricted to the subset of train set where the model predicted correctly initially**. This can be practically done by filtering out the train samples where the model predicted incorrectly initially to ensure we only iterate and refine on the subset where the model predicted correctly after the initial round (i.e., the candidate set generated by running the vanilla Reinf ICL). We refer the reviewer to the results below (we also added a row showing *Average rank* to ensure the average is not unduly biased by outlier subtasks).
>
> |  | Iterative Reinf. (restricted) |  | BRIDGE (restricted) |  |  |  |  |
> |---|---|---|---|---|---|---|---|
> | # Iteration | 1 | 2 | 1o | 1g | 2o | 2g | 3o |
> | causal_judgement | **69.7 (1.1)** | 65.0 (1.5) | 67.7 (2.7) | 65.0 (1.1) | 66.0 (1.5) | 67.0 (1.1) | 65.0 (2.0) |
> | date_understanding | 92.5 (1.5) | **94.0 (1.0)** | 92.5 (1.7) | 93.0 (1.6) | 92.5 (1.1) | 93.5 (1.7) | 89.3 (1.5) |
> | disambiguation_qa | 74.0 (0.7) | 75.8 (3.0) | 70.5 (2.7) | 72.3 (1.5) | 75.5 (3.2) | 71.8 (2.4) | **76.5 (3.8)** |
> | dyck_languages | 59.0 (6.5) | 53.0 (2.9) | 55.0 (5.2) | 52.3 (5.4) | 56.5 (3.4) | 57.5 (1.7) | **60.3 (3.8)** |
> | formal_fallacies | 86.8 (3.3) | 90.5 (2.2) | 85.3 (1.5) | **90.5 (1.1)** | 83.0 (3.2) | 83.5 (0.9) | 85.5 (2.3) |
> | geometric_shapes | 75.3 (1.8) | 78.3 (3.1) | 75.0 (2.6) | 80.5 (4.5) | 81.3 (4.5) | **85.0 (2.6)** | 80.0 (2.6) |
> | hyperbaton | 85.3 (4.0) | 84.5 (3.4) | 94.0 (1.6) | 95.8 (0.8) | 91.8 (1.3) | 93.0 (2.6) | **97.0 (0.7)** |
> | logical_deduction (7) | 67.5 (1.8) | 69.0 (2.5) | 69.5 (2.7) | **71.5 (3.2)** | 66.8 (2.1) | 70.5 (2.3) | 70.8 (2.2) |
> | movie_recommendation | 64.8 (2.7) | 63.3 (2.2) | **68.3 (3.3)** | 62.0 (4.1) | 63.0 (1.2) | 63.3 (1.3) | 61.8 (2.2) |
> | multistep_arithmetic_two | 95.0 (1.2) | 95.5 (0.5) | **97.8 (0.8)** | 95.3 (0.8) | 95.5 (1.8) | 95.8 (1.3) | 95.8 (1.5) |
> | object_counting | 94.5 (2.9) | 94.5 (2.1) | 96.0 (1.6) | 94.8 (0.8) | 94.3 (1.3) | 96.0 (0.7) | **96.3 (1.1)** |
> | ruin_names | 87.3 (1.3) | 88.8 (0.8) | **91.5 (0.9)** | 89.5 (1.8) | 88.5 (2.1) | 90.0 (0.0) | 89.8 (1.8) |
> | salient_translation | 68.0 (2.1) | 67.3 (1.5) | 70.0 (2.2) | 69.8 (3.3) | 72.8 (2.4) | 73.5 (2.7) | **75.8 (2.2)** |
> | snarks | 93.8 (1.2) | 93.8 (2.1) | 93.8 (2.3) | 95.5 (1.2) | 95.1 (0.7) | **95.5 (0.6)** | 94.8 (2.1) |
> | sports_understanding | 94.0 (1.4) | 95.0 (1.0) | 92.8 (1.1) | 95.5 (1.5) | **97.0 (0.0)** | 96.0 (0.7) | 94.8 (0.8) |
> | tracking_shuffled_objects (7)  | 66.8 (3.1) | 67.3 (1.9) | 99.0 (0.0) | 96.3 (1.5) | 98.5 (0.5) | 97.5 (1.5) | **100.0 (0.0)** |
> | Avg acc | 79.62 | 79.70 | 82.40 | 82.45 | 82.37 | _83.08_ | **83.32** |
> | Avg rank | 4.81 | 4.44 | 3.75 | 3.81 | 4.25 | **2.88** | _3.06_ |
>
> On a high level, this provides further evidence on the importance of selection: Iterative Reinf ICL (restricted) without the "optimize" step actually did *not* improve over standard Reinf ICL (avg acc = 79.6%). BRIDGE (restricted), however, still meaningfully improves with the subsequent optimize and generate steps, although the gain is less than the original BRIDGE which utilizes more examples via the larger train set support.
>
> We hope the additional requested baseline should at least provide some evidence demonstrating the value of the proposed algorithm and we plan to will add the results to the paper. We thank the reviewer again for their valuable suggestions and we hope that in light of the updated results the reviewer could consider adjust their rating.

---

> > ### Comment · Reviewer_YxzH · 2024-11-28
> >
> > I thank the authors for the conducted experiment. I don't have any concerns now and I strongly suggest authors to include the strong baseline from either [1] or [2] for the latest large open-source models to obtain model generated reasoning for the entire training set, further strengthening the work. I adjusted my rating accordingly.
> >
> > [1] Hu et al. Amortizing intractable inference in large language models. ICLR 2024
> >
> > [2] Zhao et al. Probabilistic Inference in Language Models via Twisted Sequential Monte Carlo. ICML 2024

---

> ### Author Response · Authors · 2024-11-29
> **Thank you for engaging with us during the rebuttal.**
>
> We'd like to thank the reviewer for their constructive feedback and engaging with us during the discussion period -- we believe that the reviewer's suggestions have strengthened our paper, and we are happy that our responses have addressed the reviewer's concerns.
>
> We agree that the tuning-based baselines [1-2] suggested by the reviewer are highly relevant -- we will definitely incorporate the discussions of them alongside the new experimental results obtained in the final version of the paper when we get the chance to revise it again. We will endeavor to investigate these methods more in-depth both for a comparison and for possible combinations of tuning/training-based and inference-time-based methods, which, as we previously mentioned, an important future direction in our opinion.

---

### Official Review · Reviewer_DZro · 2024-11-03

**Soundness:** 3
**Presentation:** 4
**Contribution:** 3
**Rating:** 6
**Confidence:** 4

**Summary:**

This paper studies the problem of how to improve many-shot in-context learning (ICL). The first observation is that the quality of ICL examples can be more important than the quantity -- a carefully selected subset of in-context learning examples can outperform the whole set.

Based on this observation, this paper naturally moves to the question of how to improve the quality of ICL examples. The key idea is to iteratively select helpful examples from both the seed examples and those generated by the model itself, similar in spirit to STaR but applied to ICL instead. Specifically, in each iteration, the model is prompted to generate both reasoning steps and the answer. Only those with correct answers are kept. Then a Bayesian Optimizer is used to select the best ICL examples in this iteration, which are used as ICL examples for generation in the next iteration.

Results on BBH show that the proposed method largely outperforms baseline methods. Small improvements are achieved on MATH and GSM8K. A large improvement is observed on the BIRD-Challenging subset.

**Strengths:**

- Many-shot ICL is an important problem from both the capability and safety angles.
- This paper shows that even for many-shot ICL, good examples are still necessary.
- The proposed method is a natural extension of the STaR framework to ICL. Given the hardness of optimizing ICL example selection, the proposed optimizer looks like a good contribution.
- Experimental results on BBH look strong, and small but non-trivial improvement are shown on math (MATH and GSM8K) and text-to-sql (BIRD) datasets.

**Weaknesses:**

- The experiments are done only with Gemini models on text tasks. It would be interesting to see the performance on GPT-4o / o1 models and multimodal tasks.
- Variance is not reported for MATH, GSM8K, and BIRD. Given the relatively small improvement on these datasets, it's unclear if the gain is significant.
- As the method optimize the performance on a dataset/task, it would be interesting to see the performance under distributional shifts. For example, does the gain on GSM8K transfer to GSM-symbolic?

**Questions:**

I don't fully understand the claim that BRIDGE is tailored to many-shot ICL. I feel it also applies to few-shot ICL?

---

> ### Author Response · Authors · 2024-11-22
> **Response (Part 1/2)**
>
> We thank the reviewer for their positive and insightful comments. We’d like to refer the reviewer both to the overall response and the detailed, point-by-point reply below, which we believe has thoroughly addressed their concerns. We hope that the reviewer could consider increasing the rating in light of our response.
>
> > The experiments are done only with Gemini models on text tasks. It would be interesting to see the performance on GPT-4o / o1 models and multimodal tasks.
>
> We thank the reviewer for their suggestions and we agree that it is important to ensure the findings are not model-specific. As such, we have added results on Mistral large (mistral-large-2407) and Claude 3.5 Sonnet. Please see below for the aggregated results on BBH tasks and Appendix C.5 for detailed per-task breakdown.
>
>
> Aggregated BBH results in 4 different LLMs
> |                      Model                     | Reinf ICL | Iterative Reinf. |       | BRIDGE (Ours) |       |           |           |           |
> |:----------------------------------------------:|----------:|-----------------:|------:|--------------:|------:|----------:|----------:|----------:|
> |                  # Iterations                  |         0 |                1 |     2 |            1o |    1g |        2o |        2g |        3o |
> |       Gemini 1.5 Pro (gemini-1.5-pro-001)      |     79.61 |            81.61 | 82.37 |         82.11 | 84.61 |     85.77 | **87.13** |   _86.33_ |
> |     Gemini 1.5 Flash (gemini-1.5-flash-001)    |     80.25 |            81.91 | 80.72 |         81.61 | 82.79 | **83.79** |   _83.77_ |     83.25 |
> |       Mistral Large (mistral-large@2407)       |     84.82 |            83.22 | 87.08 |         86.65 | 83.70 |   _88.07_ |     82.80 | **88.52** |
> | Claude 3.5 Sonnet (claude-3-5-sonnet@20240620) |     88.45 |            89.89 | 90.35 |         89.16 | 91.26 | _92.00_ |   **92.20** |     90.97 |
>
> In the future, we will endeavor to include experimental results on more models and more tasks  (such as the GPT-4 family of models and/or multimodal tasks as the reviewer suggested). We hope the reviewer could sympathize with us that running more models and multimodal tasks require significant computational resources which is not feasible during the rebuttal period, that the existing results (along with the new results added in the rebuttal), which we believe are already more expansive than many existing and contemporary works, are already sufficient in establishing the value of this work.
>
> > Variance is not reported for MATH, GSM8K, and BIRD. Given the relatively small improvement on these datasets, it's unclear if the gain is significant.
>
> We’d like to refer the reviewer to the updated manuscript where we updated the results with standard deviation for MATH and GSM-Hard. On BIRD, the finetuning baseline requires more expensive resources for repeated experiments beyond the time and computational constraints during the rebuttal period. However, to show that the improvement is unlikely to be a case of random fluctuations, we note that the CHASE dev set can be divided into 11 distinct databases with very different accuracy levels, and we can compare ”CHASE+BRIDGE, Round 2” (62.0%) and “CHASE PROMPT” (60.1%)  head-to-head in each of the databases – we can see that BRIDGE performs better or equal in 9/11 databases (Wilcoxon signed-rank test p-value = 0.066).
>
> | Accuracy                | BRIDGE (best) | Base      |
> |-------------------------|---------------|-----------|
> | california_schools      | **46.7%**     | 43.8%     |
> | financial               | 54.7%         | **58.5%** |
> | toxicology              | **53.8%**     | **53.8%** |
> | card_games              | **57.1%**     | 52.9%     |
> | codebase_community      | **67.7%**     | 66.1%     |
> | superhero               | **86.8%**     | 82.2%     |
> | formula_1               | **62.1%**     | 55.7%     |
> | european_football_2     | **69.8%**     | 67.4%     |
> | thrombosis_prediction   | 48.5%         | **49.1%** |
> | student_club            | **74.7%**     | **74.7%** |
> | debit_card_specializing | **50.0%**     | 48.4%     |
> | _Overall acc_           | **62.0%**     | 60.1%     |

---

> > ### Author Response · Authors · 2024-11-22
> > **Response (Part 2/2)**
> >
> > > As the method optimize the performance on a dataset/task, it would be interesting to see the performance under distributional shifts. For example, does the gain on GSM8K transfer to GSM-symbolic?
> >
> > We thank the reviewer for their suggestion. To the best of our knowledge, the GSM-symbolic dataset is not yet publicly available and we’d also like to clarify that the experiments we conducted are on GSM-**Hard**, instead of GSM-**8K** (which Gemini 1.5 Pro already achieves 90%+ zero-shot accuracy) – GSM-Hard is obtained by replacing the numbers in GSM-8K with larger and unusual ones, significantly increasing the calculation difficulty, a procedure that actually shares some similarities with how GSM-symbolic is curated.
> >
> > To concretely answer the reviewer’s question, we added an experiment where we transfer the best examples learned by BRIDGE on GSM-Hard to GSM-8K and compared the transferred results (i.e., with distributional shift) to the results where we searched on GSM-8K (i.e., without distributional shift) in App. C6 of the updated manuscript. We can observe that while searching directly unsurprisingly performs better, there is still strong transferability from the transferred results.
> >
> > > I don't fully understand the claim that BRIDGE is tailored to many-shot ICL. I feel it also applies to few-shot ICL?
> >
> > There is no universally accepted criterion between “few-shot” and “many-shot” to our knowledge, and we roughly define “few-shot” as using no more than ~10 examples whereas “many-shot” is any number beyond that. With this definition, we do agree that BRIDGE is *not limited to* many-shot ICL, as the reviewer aptly pointed out. For example, on mistral-large-2407 where the gain from scaling examples (which is also related to the way the model is trained) is not as significant as Gemini and Claude, BRIDGE can still lead to improvements over the “optimize” step in the different rounds by improving the quality of the model-generated reasoning. The reason why we emphasized on the aspect of “many-shot” is the “generate” step of BRIDGE where we iteratively improve the examples: this can be seen in Fig 2 and 4 – note the lines corresponding to the iterated examples both 1) end later, which suggest more examples are available, and 2) dominate over the original examples in all parts of the pareto front, which suggest improvement in quality. This ensures that scaling examples in the context (i.e., many-shot) reaches higher peak performance and plateaus later (Fig 4) when the model indeed benefits from many-shot ICL (e.g., Gemini and Claude models), addressing a key pain point about many-shot ICL that the relation between performance and number of examples is highly variable and task-specific. This can also be seen in Table 1, where the best overall performance of Gemini 1.5 Pro is achieved in the scaling stage (2G). Furthermore, the demonstration selection is a combinatorial optimization problem whose difficulty in general scales exponentially if we have a higher number of candidates (i.e., many-shot); we believe the difference between a naive selection strategy (e.g., random search or heuristics) and an advanced one will also be larger in the many-shot regime as a result.

---

> > > ### Author Response · Authors · 2024-11-30
> > > **We look forward to your feedback**
> > >
> > > Dear reviewer DZro,
> > >
> > > We'd like to thank the reviewer again for their constructive and feedback review. To our understanding, the reviewer was primarily concerned about the availability of results beyond Gemini models, variance of performance on non-BBH tasks and the transferability of the resulting demonstrations -- we believe that we have addressed each of the concerns thoroughly. As we are approaching the end of the discussion period, we'd be extremely grateful if the reviewer could take a look at our response and the revised paper, and we will be more than happy to address any remaining / further questions the reviewer may have. Otherwise, we hope the reviewer would also consider adjusting the rating if they feel their concerns have been addressed satisfactorily.
> > >
> > > Best,
> > >
> > > Authors

---

### Official Review · Reviewer_cswN · 2024-11-04

**Soundness:** 2
**Presentation:** 3
**Contribution:** 2
**Rating:** 5
**Confidence:** 4

**Summary:**

This paper addresses the many-shot in-context learning (ICL) paradigm. In contrast to few-shot learning, many-shot ICL leverages a larger number of demonstration examples to better exploit long-context capabilities. Initially, the paper demonstrates that selecting influential demonstration examples remains valuable, even in long-context settings. Building on this observation, the authors propose an algorithm that alternates between two key steps: an optimization step, which employs Bayesian optimization to identify influential example sets, and a generation step, which reuses these sets to automatically extend reasoning paths, thereby facilitating the transition back to the many-shot regime.

**Strengths:**

- The proposed framework integrates example selection with LLM-generated examples to better align with real-world applications.

- The paper validates this approach through extensive experiments conducted across a wide range of tasks.

**Weaknesses:**

The validation of the findings appears overly simplistic. For instance, as model capabilities scale, the order and number of demonstration examples may have diminishing importance in affecting in-context learning (ICL) performance. The authors should reconsider their conclusions by re-evaluating their results on less advanced LLMs, providing a more comprehensive understanding of the interplay between long-context utilization and demonstration selection. Without this analysis, the conclusion may be over-claimed to specific models.

Lack of baselines. In the experiments, the authors did not compare their approach with existing learning-based strategies and learn-free ones for demonstration selection.

The writing of the paper requires further refinement. Sections 3 and 4 are challenging to read and follow, despite the method itself being relatively straightforward. For instance, the description of the different variants could be more appropriately placed in the Experimental Setting section for better clarity.

**Questions:**

Q1 In reference to Figure 1, does bottom_k refer to selecting examples that are less similar to the test case? Why does bottom_k achieve performance comparable to top_k as the number of demonstrations increases? Additionally, why does bottom_k outperform the all-examples approach?

---

> ### Author Response · Authors · 2024-11-22
> **Response (Part 1/3)**
>
> We thank the reviewer for their detailed feedback, and we’d like to refer them to both our revised manuscript which contains details of additional experiments requested and our detailed, point-by-point response below. We hope that the reviewer could consider increasing the rating if they feel their concerns have been sufficiently addressed.
>
> > The validation of the findings appears overly simplistic. For instance, as model capabilities scale, the order and number of demonstration examples may have diminishing importance in affecting in-context learning (ICL) performance. The authors should reconsider their conclusions by re-evaluating their results on less advanced LLMs, providing a more comprehensive understanding of the interplay between long-context utilization and demonstration selection. Without this analysis, the conclusion may be over-claimed to specific models.
>
> We thank the reviewer for their comment. While we acknowledge the value of improving performance on less advanced LLMs, in this work we mainly experimented on the SoTA models as we are more interested in pushing the frontier of model capabilities (i.e., focusing on the issues where there headroom for improvement is ample even for the most advanced models) rather than focusing on problems that can be easily addressed by switching to a stronger model – specifically to the reviewer’s comment, we believe that while SoTA models are expected to be considerably more robust to the ordering of the examples (this can be seen by the relatively small standard deviation in Tables 1 & 2), selecting and iteratively improving demonstrations still have a significant performance impact judging from the margin of improvements even on the SoTA models. Furthermore, Many-shot ICL, especially on more challenging task categories (e.g., reasoning) typically demands stronger models, and seminal works [1-2] have also focused on frontier models like Gemini 1.5, Claude 3.5 and models of the GPT-4 family.
>
> We do value the reviewer’s suggestion on ensuring that the findings are not model-specific, and as such we have added experiments on Mistral (mistral-large-2407) and Claude (claude-3.5-sonnet) models. Please see below for the aggregated results on BBH tasks and Appendix C.5 for detailed per-task breakdown.
>
> Aggregated BBH results in 4 different LLMs
> |                      Model                     | Reinf ICL | Iterative Reinf. |       | BRIDGE (Ours) |       |           |           |           |
> |:----------------------------------------------:|----------:|-----------------:|------:|--------------:|------:|----------:|----------:|----------:|
> |                  # Iterations                  |         0 |                1 |     2 |            1o |    1g |        2o |        2g |        3o |
> |       Gemini 1.5 Pro (gemini-1.5-pro-001)      |     79.61 |            81.61 | 82.37 |         82.11 | 84.61 |     85.77 | **87.13** |   _86.33_ |
> |     Gemini 1.5 Flash (gemini-1.5-flash-001)    |     80.25 |            81.91 | 80.72 |         81.61 | 82.79 | **83.79** |   _83.77_ |     83.25 |
> |       Mistral Large (mistral-large@2407)       |     84.82 |            83.22 | 87.08 |         86.65 | 83.70 |   _88.07_ |     82.80 | **88.52** |
> | Claude 3.5 Sonnet (claude-3-5-sonnet@20240620) |     88.45 |            89.89 | 90.35 |         89.16 | 91.26 | _92.00_ |   **92.20** |     90.97 |
>
> > Lack of baselines. In the experiments, the authors did not compare their approach with existing learning-based strategies and learn-free ones for demonstration selection.
>
> We appreciate the reviewer’s comment and following their suggestions, we have added two baselines based on diversity and retrieval based on embedding similarity in App. C1 of the updated draft. In a nutshell, if we simply would like to compare the demonstration selection component of BRIDGE outperforms the heuristic selection with the best tested hyperparameter (see table below for a summary). Since we BRIDGE is not only about demo selection, we can also combine these heuristics with the BRIDGE pipeline orthogonally, and the best performance achieved is 84.14% (Table 8), but before we elaborate on the new results, there are a few clarifications we’d like to make.
>
> Average test accuracy on BBH using gemini-1.5-pro-001
> |                      | Avg BBH Acc. |
> |----------------------|--------------|
> | Diversity, k=10      | 78.83        |
> | Diversity, k=25      | 78.38        |
> | Retrieval, k=10      | 78.59        |
> | Retrieval, k=25      | 78.41        |
> | Retrieval, k=All     | 79.12        |
> | Reinf ICL            | 79.61        |
> | Random Search (DSPy) | 80.31        |
> | BayesOpt (BRIDGE-1o) | **81.61**    |

---

> ### Author Response · Authors · 2024-11-22
> **Response (Part 2/3)**
>
> First, the setup of BRIDGE (i.e., *many-shot*) significantly differs from the typical premises of demonstration selection (i.e., *few-shot*). Demonstration selection has often been assumed to be *indispensable* because the context window has been historically a limiting factor where it was impossible or extremely expensive to scale the number of demonstrations; even when such scaling was possible, prior works on earlier models even on simpler, discriminative tasks have shown mixed results [3], and these facts have necessitated some sort of selection. However, as mentioned in the introduction, frontier models are much less subjected to these issues. For example, with context limit well into the millions, previous works showed that Gemini is both *compatible with* and *benefits from* more examples, which suggests that demonstration selection is no longer a necessity – this is reflected in the fact that several prior works in many-shot ICL [1-2] do not employ any advanced selection techniques beyond random sampling. Unlike these previous observations, Our first insight precisely shows that **demonstration selection is still beneficial even if it is no longer necessary** (Page 3) and only removes information from the context (comparing to using all demonstrations, demonstration selection picks a subset) – we believe this is already a significant contribution in the many-shot setup and is contrary to the intuition that “more is better”.
>
> Second, **BRIDGE is not just about demonstration selection**: while we proposed a Bayesian optimization framework for efficient demonstration selection, it is only one component of the BRIDGE pipeline – the “generate” step and the iterative procedure, both of which are absent in typical demonstration selection algorithm, are crucial in ensuring we can still take advantage of the long-context capabilities of frontier models (a selection-only algorithm would reduce the problem back to few-shot and won’t be able to take advantage of the long context window available) and continually improve performance, as we’ve shown in Tables 1-3. In fact, the baselines mentioned by the reviewer (e.g., diversity and similarity) are orthogonal to the BRIDGE framework by swapping the “optimize” component to these methods, which we will describe below.
>
> With these points in mind, we’d like to refer the reviewer to the added results in the App. C.1 where we replace the demonstration selection component with retrieval based on cosine similarity (similar to [4]) and diversity (similar to [5]) – we can observe that these variants underperform the version proposed in the paper: one key reason is that as we’ve shown in Sec 2, not all samples, even those with correct reasoning, are necessarily useful as demonstrations. Whereas concretely evaluating on the validation set would allow us to distinguish and prune out unuseful demos, heuristics like similarity and diversity cannot distinguish that since the selection is purely based on similarity / embedding diversity but is not linked to validation performance. Another learning/cross-validation-based baseline which uses random search that is similar to the selection procedure in the popular DSPy [6] are already included in Table 5 of the appendix. It can be seen that BRIDGE-BO outperformed all these learning-based & learning-free variants. Finally, while it is certainly possible to use an alternative demonstration selection method beyond what we experimented on, as we mentioned above, any such method is orthogonal to the overall BRIDGE framework, and **a stronger selection method will only enhance, not diminish, the overall performance of BRIDGE**.
>
> > The writing of the paper requires further refinement. Sections 3 and 4 are challenging to read and follow, despite the method itself being relatively straightforward. For instance, the description of the different variants could be more appropriately placed in the Experimental Setting section for better clarity.
>
> We thank the reviewer for their suggestions and we have added some additional descriptions to “Experimental Setting” to clarify this further, as suggested. However, we note that there are no “different variants” of the algorithm; Tables 1 and 3 show the progression of one single run of BRIDGE at different points in time as we mentioned in the caption of Table 1. We also used different color codings for the “optimize” and “generate” step to make it more clear, and made miscellaneous edits in Sec 3 and 4 to improve clarity.

---

> ### Author Response · Authors · 2024-11-22
> **Response (Part 3/3)**
>
> > Q1 In reference to Figure 1, does bottom_k refer to selecting examples that are less similar to the test case? Why does bottom_k achieve performance comparable to top_k as the number of demonstrations increases? Additionally, why does bottom_k outperform the all-examples approach?
>
> Top/bottom-K does not refer to similarity to test cases (we indeed added a baseline based on embedding similarity between demos and test cases following the reviewer’s suggestion, but that is different and distinct from Fig 1). The meaning of top_k and bottom_k is explained in the paragraph beginning on Page 4. Specifically, we ran an attribution method to learn the importance of each example. Then, we sorted the examples based on their imputed importance score; top-k means if we select the top-k examples based on their importance score, whereas bottom-k means if we select the bottom-k examples based on the score. Top-k and bottom-k are bound to converge when we set k -> N (where N is the total number of examples) because when k=N, top-k and bottom-k refer to the identical set of examples. We’ve updated the caption of Fig 1 to make it even clearer.
>
> Regarding the second part of the question, there are two aspects that we’d like to clarify: firstly, **there is no guarantee that using all examples will lead to optimal performance**, which is one of our key findings and, in fact, is a key motivation for the “optimize” step. A key argument in the paper is that even for long-context models where providing many examples in the context is possible, it is often suboptimal, and a judicious selection of examples can often outperform all examples. Secondly, we find “all examples,” which we assume is what the reviewer refers to, only using {input, output} pairs in the context is a rather weak baseline (e.g., ~10% worse from Table 1) as it does not contain the intermediate outputs, i.e., no reasoning  (the same baseline is also included as the first column (“All - Direct”) in Table 1 and 3, and the fact that even bottom-K outperforms it suggests that incorporating intermediate, model-generated rationales is key for strong performance. We also provide baselines utilizing examples with reasoning (“All - CoT”, “All - Infill” and “Reinf. ICL”) in Tables 1 and 3; they are noticeably stronger than “All -Direct”, but BRIDGE still outperforms them.
>
> References:
>
> [1] Agarwal, R., Singh, A., Zhang, L. M., Bohnet, B., Rosias, L., Chan, S., ... & Larochelle, H. (2024). Many-shot in-context learning. NeurIPS.
>
> [2] Y. Jiang, J. Irvin, J. H. Wang, M. A. Chaudhry, J. H. Chen, and A. Y. Ng (2024), “Many-Shot In-Context Learning in Multimodal Foundation Models,” arXiv preprint: arXiv:2405.09798.
>
> [3] Hao, Y., Sun, Y., Dong, L., Han, Z., Gu, Y., & Wei, F. (2022). Structured prompting: Scaling in-context learning to 1,000 examples. arXiv preprint arXiv:2212.06713.
>
> [4] Liu, J., Shen, D., Zhang, Y., Dolan, B., Carin, L., & Chen, W. (2021). What Makes Good In-Context Examples for GPT-$3 $?. arXiv preprint arXiv:2101.06804.
>
> [5] Zhang, Z., Zhang, A., Li, M., & Smola, A. (2023). Automatic chain of thought prompting in large language models. ICLR.
>
> [6] Khattab, O., Singhvi, A., Maheshwari, P., Zhang, Z., Santhanam, K., Vardhamanan, S., ... & Potts, C. (2024). Dspy: Compiling declarative language model calls into self-improving pipelines. ICLR.

---

> > ### Comment · Reviewer_cswN · 2024-11-26
> > **Thanks for the response**
> >
> > Thanks very much for the update. However, I may not be able to agree with the authors that SoTA models matter more. Noted that one of the reasons we seek to many-shot setting is that we do not need to fully fine-tune the model during deployment [Agarwal, NeurIPS'24], such setting is more practical in the context of SLM/less advanced LLM, and the authors did not address my concern regarding this issue. Thus, I am not able to further increase my score.

---

> ### Author Response · Authors · 2024-11-26
> **We didn't say "SoTA models matter more"; we are unsure why "not fully finetune the model" means "many-shot is practical to less powerful models".**
>
> As we mentioned originally in the reply, we did not say “SoTA models matter more” — we said improving smaller models is valuable, but we focused on SoTA models in this paper (although we do add a new set of experiments on **mistral-nemo (12B)**, which is a small model by modern standards). We believe it is a consensus from the community that both are important.
>
> Second, we are unsure why the desire to “not fully finetune the model” leads to the conclusion that “many-shot is more practical to less powerful models.” On the contrary, besides the reasons we stated above (e.g., the benefits in scaling from many-shot are often more obvious in long-context SoTA models), finetuning large frontier models can also be extremely costly or impossible for end-users (proprietary models), which means many-shot can be a **more** desirable alternative than in less powerful models where finetuning is easier and cheaper. In fact, **the paper the reviewer cited itself [Agarwal, NeurIP’24] only considers the SoTA models** (mostly Gemini 1.5, some Claude 3.5 and GPT-4-turbo). We’d be grateful if the reviewer could clarify further.

---

> ### Author Response · Authors · 2024-11-28
> **New results on Mistral-Nemo (12B) show that large / SoTA models are not prerequisite to performance benefits from BRIDGE**
>
> We thank the reviewer again for their time and effort. Complementary to our previous comment, while the work primarily focused on SoTA models (we provided a justification why in previous comments), we value the reviewer's suggestion on the applicability of BRIDGE on weaker and/or smaller models. We add a new experiment where we run BRIDGE on Mistral-Nemo  **12B**, along with Mistral-Large results we've already added in the updated paper. We refer the reviewer to the results below (and we'll add the details to the paper as well). We chose mistral-nemo because 1) it is a small and accessible model, yet has state-of-the-art performance amongst models of similar sizes; and 2) it supports a reasonably long context (128k), which makes it suitable in our setup.
>
> | **mistral-nemo-12b**  | Reinf ICL | Iterative Reinf. |  | BRIDGE |  |  |  |  |
> |---|---|---|---|---|---|---|---|---|
> |  | 0 | 1 | 2 | 1o | 1g | 2o | 2g | 3o |
> | causal_judgement | 53.3 | **65.3** | 62.7 | 60.0 | 58.7 | 62.7 | 64.0 | 64.0 |
> | date_understanding | 66.0 | 71.0 | 68.0 | 69.0 | 69.0 | **78.0** | 70.0 | 75.0 |
> | disambiguation_qa | 58.0 | 60.0 | 64.0 | 63.0 | 60.0 | 61.0 | 66.0 | **72.0** |
> | dyck_languages | 17.0 | 21.0 | 22.0 | 18.0 | 27.0 | 26.0 | 22.0 | **30.0** |
> | formal_fallacies | **64.0** | 55.0 | 53.0 | 63.0 | 59.0 | 52.0 | 51.0 | 59.0 |
> | geometric_shapes | 65.0 | 65.0 | 69.0 | **72.0** | **72.0** | 60.0 | 69.0 | 68.0 |
> | hyperbaton | 77.0 | 72.0 | 65.0 | 80.0 | 81.0 | 83.0 | 75.0 | **86.0** |
> | logical_deduction_seven_objects | 47.0 | 54.0 | 53.0 | 45.0 | 49.0 | **62.0** | 44.0 | 51.0 |
> | movie_recommendation | 59.0 | 45.0 | 54.0 | 68.0 | 61.0 | 63.0 | 64.0 | **70.0** |
> | multistep_arithmetic_two | 36.0 | 50.0 | 20.0 | 47.0 | 20.0 | 66.0 | 12.0 | **77.0** |
> | object_counting | 81.0 | 81.0 | 82.0 | 83.0 | 79.0 | **85.0** | 75.0 | 87.0 |
> | ruin_names | 69.0 | 60.0 | 57.0 | **76.0** | 57.0 | 72.0 | 57.0 | 70.0 |
> | salient_translation_error_detection | 47.0 | 47.0 | 45.0 | **59.0** | 49.0 | 53.0 | 49.0 | 48.0 |
> | snarks | 69.4 | 76.4 | 79.2 | 72.2 | 75.0 | 72.2 | 73.6 | **77.8** |
> | sports_understanding | 86.0 | 75.0 | 69.0 | 91.0 | 72.0 | 91.0 | 74.0 | **93.0** |
> | tracking_shuffled_objects_seven_objects | 70.0 | 69.0 | 70.0 | 91.0 | 88.0 | **94.0** | 81.0 | 93.0 |
> | Avg Acc (%) | 60.30 | 60.42 | 58.30 | 66.08 | 61.04 | _67.56_ | 59.16 | **70.05** |
>
> Overall, we find that BRIDGE continues to lead to large improvement margins (+10% on average compared to the base Reinf ICL), although similar to mistral-large, the model seems to benefit less from more examples compared to proprietary models with strong many-shot abilities (e.g., Gemini and Claude), but this is more related to how the models are trained to take advantage of the examples. Nonetheless, the *generate* step leads to better candidates that act as a pool for the subsequent "optimize" step, which is highly effective in this case. We believe this convincingly show that BRIDGE can still be highly effective in the context of smaller models / models that are weaker in many-shot ICL, which we believe was the crux of the reviewer concern -- we hope the reviewer could consider adjusting their score if they feel their concerns have been addressed adequately.

---

> > ### Author Response · Authors · 2024-11-30
> > **We look forward to your feedback**
> >
> > Dear Reviewer cswN,
> >
> > We thank the reviewer again for their feedback. To our understanding, the biggest concern the reviewer had was about the applicability of the method on smaller/less advanced models. We believe we have addressed this concern by 1) providing further clarification on why we primarily focused on SoTA models and 2) **added a new set of results on a 12B model to demonstrate that the performance gain is robust to model capabilities / sizes**. As we are approaching the end of the discussion period, we'd kindly request the reviewer to take a look at our response and new results and consider adjusting the score if they feel their concerns have been resolved.
> >
> > Best,
> >
> > Authors

---

> ### Author Response · Authors · 2024-12-03
>
> Dear Reviewer cswN,
>
> We have provided additional experiments and explanations to address your valuable feedback.
>
> Could you please kindly take a look, and let us know if you have further concerns so that we can provide further clarifications if needed?
>
> Thank you so much!
>
> Best,
>
> Authors

---

> ### Comment · Reviewer_cswN · 2024-12-03
> **Thank you**
>
> thanks very much for the addtional results by using Mistral-Nemo (12B). Most of my concerns have been addressed, I am happy to increase the score, but I still expect to see the performance of SLM (e.g., 7B) (maybe with some discussion?), even the results are not desired.

---

> ### Author Response · Authors · 2024-12-04
> **We conducted some preliminary experiments on a 7B model as requested.**
>
> We thank the reviewer for engaging with the discussions and we are glad that most of the concerns have been addressed and they are happy with increasing the score. We also greatly value the reviewer's suggestions on SLMs and will definitely incorporate some discussion (such as the one towards the end of our last reply) into the paper when we get the chance to revise it again.
>
> In response to the request of running experiments on even smaller models, we have conducted a preliminary set of experiments a **7B** model (specifically, Mistral-7B-Instruct-v0.3):
>
> |Mistral-7B-Instruct-v0.3| Reinf ICL | Iter. Reinf. |  | BRIDGE |  |  |  |  |
> |---|---|---|---|---|---|---|---|---|
> |  | 0 | 1 | 2 | 1o | 1g | 2o | 2g | 3o |
> | tracking_shuffled_objects (7) | 10.0 | 13.0 | 17.0 | 28.0 | 19.0 | _35.0_ | 26.0 | **36.0** |
> | disambiguation_qa | _74.0_ | 69.0 | 72.0 | 68.0 | **77.0** | 65.0 | 60.0 | 70.0 |
> | geometric_shapes | 34.0 | 28.0 | **46.0** | 37.0 | 31.0 | 35.0 | 33.0 | _43.0_ |
> | object_counting | 41.0 | 53.0 | _57.0_ | 53.0 | 55.0 | 51.0 | 41.0 | **59.0** |
>
> We hope the reviewer could sympathize with us that given the time constraint (the last chance to reply is by AoE time today (12/3); we were only aware of the reviewer's request a few hours ago since OpenReview does not actively notify participants of edits to existing comments), it was infeasible to complete a full set of experiments on all tasks on time and we have to use a smaller search budget for the optimize step (we used $n_{\mathrm{eval}} = 8$ in the table above), although we endeavor to have results that are compatible with the rest of the paper by the camera-ready deadline. Having said that, we believe that the preliminary results still show promise of the proposed method. Additionally, we believe that the since the 12B model should behave more similarly to typical SLMs than to the SoTA models, the results in our previous response should also at least imply some evidence on the effectiveness of our method on even smaller models.

---

### Author Response · Authors · 2024-11-22
**Overall response**

Overall Response
We thank all reviewers for their feedback and suggestions! In the common response below, we’d like to provide in-depth answers to some of the common questions, and we also summarize the key changes in the revised manuscript.

### Experiments on Additional LLMs
We appreciate the reviewers’ suggestions on experiments on additional LLMs, and we agree that it is an important point. As such, we have added results on Mistral large (mistral-large-2407) and Claude 3.5 Sonnet. While **we refer the reviewer to App. C5 of the updated manuscript for detailed results**, we’d also like to summarize the key results below (the Gemini results are taken from Tables 1 and 3 in the paper). We find that in all LLMs BRIDGE led to improvements, which suggest that the gains in the paper are not model-specific.


Table S1: Aggregated BBH results in 4 different LLMs
|                      Model                     | Reinf ICL | Iterative Reinf. |       | BRIDGE (Ours) |       |           |           |           |
|:----------------------------------------------:|----------:|-----------------:|------:|--------------:|------:|----------:|----------:|----------:|
|                  # Iterations                  |         0 |                1 |     2 |            1o |    1g |        2o |        2g |        3o |
|       Gemini 1.5 Pro (gemini-1.5-pro-001)      |     79.61 |            81.61 | 82.37 |         82.11 | 84.61 |     85.77 | **87.13** |   _86.33_ |
|     Gemini 1.5 Flash (gemini-1.5-flash-001)    |     80.25 |            81.91 | 80.72 |         81.61 | 82.79 | **83.79** |   _83.77_ |     83.25 |
|       Mistral Large (mistral-large@2407)       |     84.82 |            83.22 | 87.08 |         86.65 | 83.70 |   _88.07_ |     82.80 | **88.52** |
| Claude 3.5 Sonnet (claude-3-5-sonnet@20240620) |     88.45 |            89.89 | 90.35 |         89.16 | 91.26 | _92.00_ |   **92.20** |     90.97 |


### Key Changes
Here we summarize the key changes in the latest revision. We highlight all changes and additions in $\textcolor{magenta}{\text{magenta}}$ for the convenience of the reviewers, and below we list all key changes.

- Added standard deviation for MATH and GSM-Hard results in Table 2.
- Added the infilling baseline utilizing all samples + model-generated reasoning given inputs and ground-truth labels in Table 1 & 3, and the implementation details are in Appendix B.2.
- Added results for Mistral Large and Claude 3.5 Sonnet in Appendix C.5
- Added results comparing BRIDGE against additional demonstration selection methods in Appendix C.1.
- Added transfer learning experiments (transferring from GSM-Hard to GSM-8K) in Appendix C.6.
- Corrected typo in Algorithm 2 .
- Updated captions of Fig 1 to better clarify the meaning of “top-K” and “bottom-K”.
- Added the cost analysis to Appendix D.
- Miscellaneous adjustments to Sec 3 for better clarity.
- Updated Conclusion to reflect the future direction of more efficient methods with white-box model access.

---

### Meta-Review · Area_Chair_WD9R · 2024-12-17

**Metareview:**

This paper investigates the many-shot in-context learning (ICL) paradigm, emphasizing that the quality of selected demonstration examples can significantly influence performance even when leveraging long-context capabilities. To address this, the authors propose BRIDGE, an iterative algorithm that combines Bayesian optimization for influential example selection with model-generated reasoning examples. The method alternates between optimization and generation steps to enhance the quality of demonstrations and reasoning paths. The effectiveness of BRIDGE is demonstrated through experiments on various tasks, particularly on reasoning-heavy benchmarks like BigBenchHard (BBH), MATH, GSM8K, and BIRD.

Strengths

+ The paper addresses the critical problem of enhancing many-shot ICL, a relatively new but impactful area due to the rise of long-context models.
+ The proposed framework integrates example selection and model-generated reasoning, making it applicable to both closed- and open-source models.
+ BRIDGE is well-motivated by the observation that a subset of influential examples can drive performance improvements, even in a many-shot regime.
+ Extensive experiments on reasoning-heavy tasks (e.g., BBH, MATH, GSM8K, and BIRD) validate the proposed method, with strong improvements on BBH and smaller yet non-trivial gains on other datasets.
+ The study provides novel insights into the importance of example selection over mere quantity, highlighting its practical implications for ICL.

Weaknesses

+ Experiments are limited to the Gemini family of models, with no evaluation on open-source models like LLaMA or other state-of-the-art LLMs (e.g., GPT-4o), limiting the generalizability of the findings.
+ Important baselines, such as using all examples with model-generated reasoning, are missing, which weakens claims about the necessity of example selection and iterative refinement.
+ Variance in experimental results, particularly for tasks like MATH and GSM8K, is not reported, making it unclear whether the observed gains are statistically significant.
+ The computational cost of the optimization and generation steps in BRIDGE is not analyzed, which could limit the method's practicality.
+ The paper lacks clarity in certain sections, particularly the explanation of the "optimize" step, Bayesian optimization setup, and the estimation of importance scores.
+ The analysis is restricted to reasoning-heavy tasks, leaving open questions about BRIDGE's applicability to simpler tasks or models with limited reasoning abilities.

Most concerns have been addressed by the authors during the rebuttal period.

**Additional Comments On Reviewer Discussion:**

This paper ended with mostly positive reviews, with only one borderline reject, which I believe would have been raised to borderline accept if the reviewer is able to see the authors’ latest response. Specifically, the reviewer has asked for additional results on Mistral-Nemo (12B) and SLM, which the authors provided late in the discussion period. Another reviewer, YxzH, asked for additional results on strong baselines, which the authors provided and led the an increased rating to 8. Overall, I feel the authors did a good job in clarifying their contributions and encouraging them to incorporate the new results and discussions into the camera ready version.

---

### Decision · Program_Chairs · 2025-01-22

Accept (Poster)